# *Breaking Multi-Task Curse*:
# Reward-Weighted Evolution for Black-Box Many-Task Optimization

Yanchi Li[1]   Jiao Liu[2]   Wenyin Gong[1]   Qiong Gu[3]   Yue Zhao[4]   Yew-Soon Ong[2,5]

## Abstract

Evolutionary multi-tasking accelerates black-box optimization via knowledge transfer but falters in scenarios involving many low-similarity tasks. We identify this scalability barrier as the *Multi-Task Curse*, driven by evaluation budget dispersion and negative transfer. To overcome this, we propose MES-RET (*M*any-task *E*volution *S*trategy with *R*eward-weighted *E*valuation and *T*ransfer), which combats budget dispersion via a reward-weighted evaluation scheme that guarantees superior expected improvement, while simultaneously mitigating negative transfer through a robust reward-weighted aggregation of mean and covariance statistics, ensuring a safe fallback to independent evolution. Furthermore, to handle neural dimensional mismatches in many-task policy search, we introduce a semantic parameter alignment strategy that bridges heterogeneous state-action spaces. Extensive experiments on synthetic benchmarks, real-world engineering problems, and reinforcement learning tasks demonstrate that MES-RET consistently outperforms state-of-the-art methods, notably enabling skill transfer across morphologically distinct policies.

## 1. Introduction

Recently, many studies have explored the use of black-box optimizers for training neural networks, particularly in scenarios where gradient-based methods face instability issues. For instance, in reinforcement learning (RL) control tasks,

[1]School of Computer Science, China University of Geosciences, Wuhan, China [2]College of Computing and Data Science, Nanyang Technological University, Singapore [3]School of Computer Engineering, Hubei University of Arts and Science, Xiangyang, China [4]School of Electronic Engineering, Xidian University, Xi'an, China [5]Centre for Frontier AI Research (CFAR), Agency for Science, Technology and Research, Singapore. Correspondence to: Wenyin Gong <wygong@cug.edu.cn>.

*Proceedings of the 43$^{rd}$ International Conference on Machine Learning*, Seoul, South Korea. PMLR 306, 2026. Copyright 2026 by the author(s).

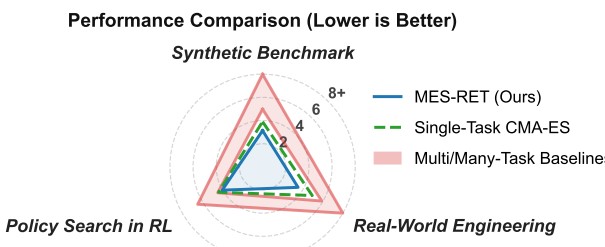

**Performance Comparison (Lower is Better)**

*Synthetic Benchmark*

— MES-RET (Ours)
-- Single-Task CMA-ES
Multi/Many-Task Baselines

*Policy Search in RL*          *Real-World Engineering*

*Figure 1. Breaking Multi-Task Curse.* The red zone marks the minimum to maximum Friedman ranking range where multi/many-task methods fail to beat the single-task baseline (green dashed line) in three low-similarity many-task optimization scenarios. Our proposed MES-RET (blue) consistently breaks this curse, outperforming all baselines.

gradient-based approaches often suffer from unstable training dynamics due to sparse rewards and high variance in policy gradients (Li et al., 2024a). Similarly, in physics-informed neural networks, the complex loss landscapes can lead to training instabilities (Wong et al., 2025), and in image analysis, neural models are often highly vulnerable to adversarial attacks (Dong et al., 2024). In such challenging scenarios, evolutionary algorithms (EAs) (Michalewicz, 1996; Hansen & Ostermeier, 2001), as a class of effective black-box optimization methods, have demonstrated their ability to provide stable and robust solutions (Salimans et al., 2017; Wong et al., 2021). The strengths of EAs lie in their global search capability, gradient-free nature, and inherent parallelizability, which benefit from simulating natural evolutionary processes (Miikkulainen, 2025). For example, their global search ability enables them to achieve comparable performance to large-scale networks while using relatively small neural architectures (Chalumeau et al., 2023; Zhang et al., 2023). Despite these advantages, the sample inefficiency of EAs remains a critical bottleneck, often requiring millions of interactions to converge, which limits their broader adoption in resource-constrained artificial intelligence applications.

Evolutionary multi-tasking (EMT) (Gupta et al., 2016; Wei et al., 2022) has emerged as a promising paradigm to mitigate this inefficiency. By optimizing multiple related tasks simultaneously, EMT exploits latent similarities among tasks, such as correlations in objective function landscapes (Bali et al., 2020; Li et al., 2024e), to facilitate knowl-

edge transfer, demonstrating significant potential in classic combinatorial optimization scenarios (Guo et al., 2025a;b). Intuitively, this paradigm incorporates a multitask interaction term that acts as a structural prior regarding parameter similarity (Bai et al., 2022). In an ideal scenario, such a mechanism allows a single evaluation of a candidate solution to contribute to the evolution of multiple tasks, thereby maximizing information utilization and enhancing overall convergence speed.

However, standard EMT methods often falter when scaling to many-task scenarios involving dozens of tasks with low similarity, a challenge we identify as the *Multi-Task Curse*. This curse manifests primarily in two aspects: *i) Evaluation Budget Dispersion*: With a fixed total evaluation budget, increasing the number of tasks reduces the resources available for each, making it critical to allocate evaluations dynamically to tasks with the highest potential for improvement rather than uniformly. *ii) Negative Transfer*: When tasks possess divergent objective landscapes, varying dimensionalities, or distinct physical dynamics (in RL) (Gupta et al., 2022; Zhang et al., 2024; Towers et al., 2024), blind knowledge transfer based on spurious correlations can mislead the search, deteriorating performance below that of independent solvers. According to our experimental observations (see Figure 1), due to these challenges, existing EMT methods often struggle to outperform well-established single-task baselines like CMA-ES (Hansen & Ostermeier, 2001) in low-similarity many-task settings.

In this paper, we propose MES-RET (*M*any-task *E*volution *S*trategy with *R*eward-weighted *E*valuation and *T*ransfer) to systematically break the *Multi-Task Curse*. Distinguished by its scalability and robustness, MES-RET effectively solves dozens of diverse tasks within a single evolutionary process without relying on expensive learning or domain-specific feature extraction. To counter *Evaluation Budget Dispersion*, it employs a potential-driven allocation strategy. By dynamically biasing resources, the method satisfies the critical positive correlation condition required to maximize global gains. To mitigate *Negative Transfer*, we introduce a protective transfer mechanism that functions as a safety valve. It ensures that external guidance is competitively filtered, allowing the algorithm to seamlessly revert to independent evolution whenever cross-task correlations prove maladaptive. The main contributions are as follows:

1. We propose *Reward-Weighted Evaluation*, an evaluation budget allocation scheme that synthesizes convergence and diversity signals to quantify task potential. It is designed to satisfy the sufficient condition for superior expected improvement over uniform strategies.

2. We introduce *Reward-Weighted Transfer*, a safe knowledge transfer technique. By transferring aggregated Gaussian statistics via weak guidance injection, it fil-

ters task-specific noise and ensures robustness against negative transfer in low-similarity regimes.

3. We design *Semantic Policy Parameter Alignment* to bridge dimensional mismatches in scalable many-task RL. This strategy structurally decouples policy parameters into joint-search and task-specific subspaces, serving as a prerequisite for knowledge transfer across heterogeneous state-action spaces.

Extensive experiments on synthetic benchmarks, real-world engineering problems, and many-task policy search scenarios confirm that MES-RET consistently outperforms state-of-the-art methods in terms of optimization quality and convergence efficiency across both many-task optimization and policy search scenarios.

**Conflict of Interest Disclosure**    The authors have no relevant financial or non-financial interests to disclose.

## 2. Background

### 2.1. Multi- & Many-Task Optimization

Multi-task optimization refers to the simultaneous optimization of multiple tasks, where each task is defined by a objective function $f_k : \mathbb{R}^{n_k} \to \mathbb{R}$, with $k = 1, 2, \ldots, K$ indexing the tasks. The objective is to find a set of optimal solutions $\{\mathbf{x}_k^* \in \mathbb{R}^{n_k} \mid k = 1, 2, \ldots, K\}$ that achieve the best objective values $f_k(\mathbf{x}_k^*)$ for their respective tasks. As the number of tasks increases, multi-task scales into many-task optimization, which involves optimizing many tasks simultaneously (Liaw & Ting, 2019). This scaling introduces significant challenges in computational resource allocation and requires robust mechanisms to mitigate negative transfer among potentially diverse tasks.

MFEA (Gupta et al., 2016) is the pioneering EMT method in solving black-box multi-task optimization problems, which introduces a multifactorial framework to optimize multiple tasks in a single population with implicit inter-task variable swapping. It has been extended to various EMT variants, such as MFEA-II (Bali et al., 2020) and AT-MFEA (Xue et al., 2022). In addition to multifactorial frameworks, EMT methods have also been developed based on multi-population or multi-distribution approaches, such as EMaTO-MKT (Liang et al., 2022), SBCMAES (Liaw & Ting, 2019), and MTES-KG (Li et al., 2024e). These methods typically involve maintaining multiple populations or distributions for multiple tasks, where each population or distribution is optimized independently but can share information through explicit knowledge transfer techniques.

### 2.2. Evolution Strategies

ES is a class of black-box optimization methods that evolve a population distribution, commonly represented as a mul-

tivariate Gaussian distribution $\mathcal{N}(\boldsymbol{m}, \boldsymbol{C})$, where $\boldsymbol{m} \in \mathbb{R}^n$ denotes the mean vector and $\boldsymbol{C} \in \mathbb{R}^{n \times n}$ denotes the covariance matrix. The optimization process involves iteratively sampling solutions from this distribution, evaluating their objective values, and updating the distribution parameters.

The most widely used ES variant is the covariance matrix adaptation ES (CMA-ES) (Hansen & Ostermeier, 2001), which iteratively adapts the mean $\boldsymbol{m}$, covariance matrix $\boldsymbol{C}$, and step-size $\sigma$ to guide the search. At each generation, $\lambda$ candidates are sampled via $\boldsymbol{x}_i \sim \mathcal{N}(\boldsymbol{m}, \sigma^2 \boldsymbol{C})$. To exploit correlations between consecutive steps, CMA-ES maintains two evolution paths, $\boldsymbol{p}_c$ and $\boldsymbol{p}_\sigma$, which accumulate search directions for the covariance and step-size updates, respectively. The update rules are:

$$
\begin{aligned}
\boldsymbol{m} &\leftarrow \boldsymbol{m} + \sigma \boldsymbol{y}_w, \quad \text{with } \boldsymbol{y}_w = \sum_{i=1}^{\mu} w_i \frac{\boldsymbol{x}_{i:\lambda} - \boldsymbol{m}}{\sigma} \\
\boldsymbol{p}_c &\leftarrow (1 - c_c) \boldsymbol{p}_c + \sqrt{c_c(2 - c_c) \mu_{\text{eff}}} \sqrt{\boldsymbol{C}} \boldsymbol{y}_w \\
\boldsymbol{C} &\leftarrow (1 - c_1 - c_\mu) \boldsymbol{C} + c_1 \boldsymbol{p}_c \boldsymbol{p}_c^\top + c_\mu \sum_{i=1}^{\mu} w_i \boldsymbol{y}_i \boldsymbol{y}_i^\top \\
\sigma &\leftarrow \sigma \cdot \exp\left( \frac{c_\sigma}{d_\sigma} \left( \frac{\|\boldsymbol{p}_\sigma\|}{\mathbb{E}[\|\mathcal{N}(\boldsymbol{0}, \boldsymbol{I})\|]} - 1 \right) \right),
\end{aligned}
\tag{1}
$$

where $\boldsymbol{y}_i$ represents the normalized displacement of sampled solutions, $\boldsymbol{y}_w$ is the weighted mean of $\mu$ selected elites utilizing recombination weights $w_i$, and $\boldsymbol{p}_\sigma$ is updated similarly to $\boldsymbol{p}_c$ but with isotropic scaling. The constants $c_c, c_1, c_\mu, c_\sigma, d_\sigma$ are learning rates and damping factors. This adaptive mechanism allows CMA-ES to approximate second-order curvature and achieve robust convergence (Gissler, 2024). Another notable ES variant is natural ES (NES) (Wierstra et al., 2014), which approximates the natural gradient for updating distribution parameters. In this work, we focus on CMA-ES as the backbone optimizer due to its robustness and proven effectiveness in black-box optimization tasks.

### 2.3. Evolutionary Reinforcement Learning

RL is typically formulated as a Markov decision process $(\mathcal{S}, \mathcal{A}, P, r, \gamma)$ (Sutton & Barto, 2018; Mnih et al., 2015), where the goal is to learn a policy $\pi_{\boldsymbol{\theta}}$ that maximizes the expected cumulative reward:

$$
J(\boldsymbol{\theta}) = \mathbb{E}_{\tau \sim \pi_{\boldsymbol{\theta}}} \left[ \sum_{t=0}^{T} \gamma^t r(s_t, a_t) \right].
\tag{2}
$$

In deep RL, policies are parameterized by neural networks. Consequently, variations in the dimensions of state space $\mathcal{S}$ and action space $\mathcal{A}$ across tasks lead to incompatible parameter spaces $\boldsymbol{\theta}$, creating significant barriers for direct inter-task knowledge transfer.

Evolutionary RL refers to the application of evolutionary methods for RL tasks (Li et al., 2024a; Miikkulainen, 2025).

It has been shown that ES can search policies through neuroevolution in RL tasks by directly optimizing the policy parameters (Salimans et al., 2017; Lange et al., 2023). In addition to purely evolutionary approaches, several methods integrate evolutionary methods with policy gradient-based methods. Representative examples include CERL (Khadka et al., 2019), ERL-Re$^2$ (Hao et al., 2023), and EvoRainbow (Li et al., 2024b), which combine policy gradient techniques (Schulman et al., 2017; Mnih et al., 2016) with population-based search to leverage the strengths of both paradigms. In this work, for the sake of simplicity and generality, we focus on ES for black-box policy search in evolutionary RL, aiming to directly evolve policy parameters without relying on policy gradients.

### 2.4. Formalization of the *Multi-Task Curse*

To rigorously formalize the negative transfer and the *Multi-Task Curse* ($\mathcal{I}_{MT} < \mathcal{I}_{ST}$) identified in Section 1, we analyze the Taylor expansion of the expected improvement. In CMA-ES, optimization operates on the Gaussian-smoothed objective $J_k(\boldsymbol{m}_k, \boldsymbol{C}_k) = \mathbb{E}_{\boldsymbol{x} \sim \mathcal{N}}[f_k(\boldsymbol{x})]$. We quantify the curse by decomposing the difference in total expected improvement between multi-task ($\mathcal{I}_{MT}$) and single-task ($\mathcal{I}_{ST}$) optimization:

$$
\mathcal{I}_{MT} - \mathcal{I}_{ST} \approx \underbrace{\sum_{k=1}^{K} \frac{\partial \mathbb{E}[\Delta_k]}{\partial b_k} \left( b_k - \frac{B}{K} \right)}_{\text{Resource Dilution}}
$$
$$
+ \underbrace{\sum_{k=1}^{K} \left( \langle \nabla_{\boldsymbol{m}} J_k, \delta \boldsymbol{m}_k \rangle + \text{Tr}(\nabla_{\boldsymbol{C}} J_k \delta \boldsymbol{C}_k) \right)}_{\text{Negative Transfer}},
$$
$$
\tag{3}
$$

where $b_k$ is the allocated evaluation budget for task $k$, $B$ is the total budget, and $\delta \boldsymbol{m}_k, \delta \boldsymbol{C}_k$ are the transferred parameter updates. Using the properties of ES natural gradients ($\nabla_{\boldsymbol{m}} J_k \approx \nabla f_k$ and $\nabla_{\boldsymbol{C}} J_k \approx \frac{1}{2} H_k$), the mathematical inevitability of the curse under naive multi-tasking decomposes into three factors:

1. *Resource Dilution*: Uniform budget allocation forces the first term to be $\leq 0$ as tasks scale.

2. *Mean Shift*: Conflicting transferred updates ($\delta \boldsymbol{m}_k$) oppose the true local gradients ($\langle \nabla f_k, \delta \boldsymbol{m}_k \rangle < 0$).

3. *Covariance Distortion*: Maladaptive variance ($\delta \boldsymbol{C}_k$) inflates the condition number of the search distribution, driving $\text{Tr}(H_k \delta \boldsymbol{C}_k) < 0$.

As will be detailed in Section 3, the proposed MES-RET breaks this curse systematically. Its reward-weighted evaluation optimizes the budget allocation against dilution. The semantic alignment pools high-quality shared statistics to

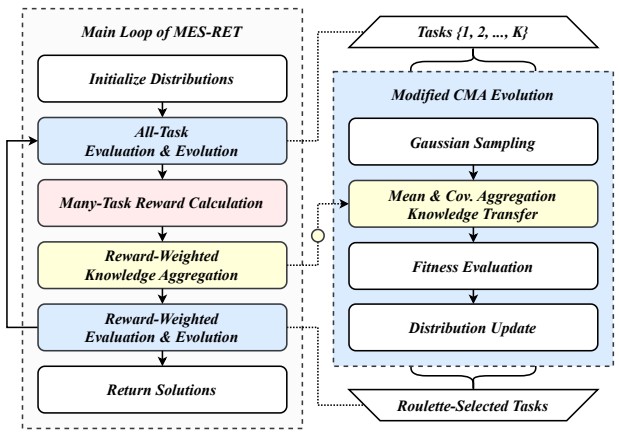

*Figure 2.* Workflow of the proposed MES-RET.

maximize the transfer upper bound. Crucially, the Weak Guidance Injection establishes a strict lower bound: by restricting external knowledge to discrete evaluation, it safely filters conflicting gradients and distorted covariances via rank-based truncation. Even if mediocre conflicting knowledge barely survives truncation, its non-linear update weight $w_i$ decays exponentially toward 0, ensuring a robust fallback to independent evolution.

## 3. Proposed Method: MES-RET

Figure 2 illustrates the architecture of MES-RET, which structurally decouples the main control loop (left panel) from a shared evolutionary engine (right panel). The process initiates by establishing Gaussian distributions $\mathcal{N}(\boldsymbol{m}_k, \boldsymbol{C}_k)$ for all $K$ tasks, where $\boldsymbol{m}_k \in \mathbb{R}^{n_k}$ and $\boldsymbol{C}_k \in \mathbb{R}^{n_k \times n_k}$ denote the mean vector and covariance matrix, respectively.

**Modified CMA Evolution.** Central to the framework is the unified evolution module (right panel), which augments standard CMA-ES with a knowledge transfer phase. Unlike isolated evolution, this module interjects a *Mean & Covariance Aggregation* step immediately after Gaussian sampling. Here, external guidance derived from global statistics is injected into the candidate solutions before they undergo fitness evaluation and distribution updates.

**Iterative Optimization Process.** The main loop coordinates the learning process through a four-stage sequence per generation. First, in the *All-Task Evolution* phase, the modified evolution module is executed for every task $\{1, \ldots, K\}$ to ensure comprehensive performance assessment. Second, the system computes *Many-Task Rewards* based on the observed performance gains and distribution diversity. Third, these rewards drive the *Knowledge Aggregation* step, updating the global statistics that fuel the transfer phase in the subsequent phase. Finally, the *Reward-Weighted Evaluation* phase allocates an additional computational budget to a sub-

---

**Algorithm 1** Reward-Weighted Evaluation

1: **for** $k = 1$ to $K$ **do**
2:     Evolve task $k$: $\mathcal{N}_k \leftarrow$ CMA-Evolution($\mathcal{N}_k, \lambda$)
3: **end for**
4: $\{r_k\}_{k=1}^K \leftarrow$ Calculate-Reward($\cdot$){See Sec. 3.2}
5: $\{p_k\}_{k=1}^K \leftarrow \{r_k / \sum_{j=1}^K r_j\}_{k=1}^K$
6: **for** 1 to $K$ **do**
7:     Select task: $t \leftarrow$ Categorical-Sampling($\{p_k\}_{k=1}^K$)
8:     Evolve task $t$: $\mathcal{N}_t \leftarrow$ CMA-Evolution($\mathcal{N}_t, \lambda$)
9: **end for**

---

set of high-potential tasks to accelerate their convergence using the same evolution engine. This cycle repeats until stopping criteria are met, yielding solutions $\{\boldsymbol{x}_k^*\}_{k=1}^K$.

The complete pseudocode of MES-RET is detailed in Appendix C. As analyzed in Appendix D, the time complexity of MES-RET is $\mathcal{O}(K(\lambda n^2 + \lambda \log \lambda))$ (full cov.) or $\mathcal{O}(K(\lambda n + \lambda \log \lambda))$ (separable), where $\lambda$ is the sample size and $n$ is the dimensionality; both are asymptotically equivalent to $K$ independent CMA-ES or sep-CMA-ES runs, ensuring negligible overhead.

### 3.1. Reward-Weighted Evaluation

The core driver of our efficiency is the reward-weighted evaluation module. Instead of a uniform allocation, it operates through a two-phase strategy as shown in Algorithm 1. In each generation, all tasks first undergo one evolution cycle, consuming $\lambda \cdot K$ evaluations to ensure each task receives equal computational resources for initial state estimation. Subsequently, an additional budget of $\lambda \cdot K$ evaluations is allocated using categorical sampling based on normalized reward probabilities derived from the tasks' potential.

To theoretically justify this biased allocation, we provide the following sufficient condition:

**Proposition 3.1** (Effectiveness Condition). *Let $r_k$ be the assigned reward and $\Delta_k$ the expected improvement for task $k$. If the allocation satisfies the positive correlation condition:*

$$K \sum_k r_k \Delta_k \geq (\sum_k r_k)(\sum_k \Delta_k), \tag{4}$$

*then the expected total improvement of the reward-weighted strategy exceeds or equals that of the uniform strategy:* $\mathbb{E}[\Delta_{total}^{weighted}] \geq \mathbb{E}[\Delta_{total}^{uniform}]$.

*Proof.* See Appendix A. □

> **Remark:** Proposition 3.1 establishes *positive correlation* as the critical prerequisite. This theoretical boundary defines our design objective: the reward signal $r_k$ must serve as a robust proxy for the expected improvement $\mathbb{E}[\Delta_k]$.

## 3.2. Many-Task Reward Calculation

Guided by the objective in Proposition 3.1, we propose a dynamic reward mechanism engineered to correlate with $\mathbb{E}[\Delta_k]$ across diverse landscapes. Since potential improvement stems from either rapid convergence or discovering new basins of attraction, we synthesize both aspects.

**Performance Improvement Reward:**  To quantify convergence potential, we compute a scale-invariant metric $\mathcal{I}_k$ based on the relative historical progress for task $k$:

$$\mathcal{I}_k = \frac{f_k^{\text{pre}} - f_k^{\text{cur}}}{|f_k^{\text{init}} - f_k^{\text{pre}}| + \epsilon}, \quad (5)$$

where $f_k^{\text{cur}}$, $f_k^{\text{pre}}$, and $f_k^{\text{init}}$ denote the current, previous, and initial best objective values. The denominator acts as a normalization factor to decouple the improvement magnitude from the specific scale of the objective function. We then normalize these scores and apply a Softmax transformation:

$$r_k^{\text{perf}} = \frac{\exp(\bar{\mathcal{I}}_k)}{\sum_j \exp(\bar{\mathcal{I}}_j)}, \quad \text{where } \bar{\mathcal{I}}_k = \frac{\mathcal{I}_k - \mathcal{I}_{\min}}{\mathcal{I}_{\max} - \mathcal{I}_{\min}}. \quad (6)$$

This accentuates tasks with significant gains while preserving non-zero probabilities for those with minor progress.

**Distribution Diversity Reward.**  To capture the capability of escaping local optima, we measure the search distribution's diversity. Maximizing the covariance magnitude increases Gaussian differential entropy, thereby preventing probability density collapse (Nitanda et al., 2022). We quantify this via the normalized trace:

$$r_k^{\text{div}} = \frac{\sigma_k \cdot \text{trace}(\boldsymbol{C}_k)}{n_k}, \quad (7)$$

where $\sigma_k$ is the step size and $n_k$ is the dimensionality. Search spaces are mapped to $[0, 1]^{n_k}$ (Gupta et al., 2016) to ensure comparability.

**Dynamic Reward Composition.**  To satisfy Eq. (4) throughout the optimization process, the proxy for improvement must adapt. Early stages benefit from rapid convergence, while later stages require sustained diversity to escape local optima. The final reward $r_k$ is selected stochastically based on the evaluation progress rate $\rho = e/E$:

$$r_k = \begin{cases} r_k^{\text{perf}} & \text{if } u \geq \rho, \quad u \sim \mathcal{U}(0, 1), \\ r_k^{\text{div}} & \text{otherwise.} \end{cases} \quad (8)$$

This $\rho$-modulated schedule ensures that the reward signal $r_k$ remains aligned with the true source of potential improvement as the search phase transitions.

**Theoretical Intuition of the Reward Design.**  While a universal theoretical guarantee is mathematically precluded

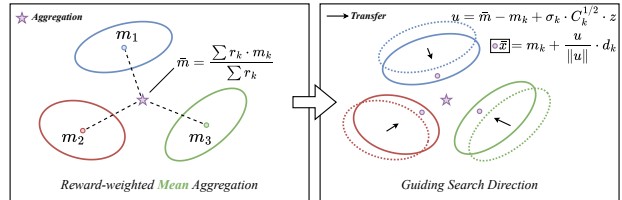

*Figure 3.* Reward-weighted mean aggregation and transfer for guiding search direction.

in highly deceptive landscapes, our formulation satisfies the condition in Proposition 3.1 via a dual-mechanism approach tailored to CMA-ES dynamics. Under the assumption of local smoothness, CMA-ES performs approximate natural gradient descent on the Gaussian-smoothed objective $J_k(\boldsymbol{m}_k, \boldsymbol{C}_k) = \mathbb{E}_{\boldsymbol{x} \sim \mathcal{N}}[f_k(\boldsymbol{x})]$. Because consecutive natural gradients correlate strongly, historical progress ($r_k^{\text{perf}}$) serves as a mathematically rigorous proxy for expected local improvement $\mathbb{E}[\Delta_k]$, guaranteeing that the positive correlation condition holds reliably during early-to-mid optimization stages. Conversely, $r_k^{\text{div}}$ is designed to counteract CMA-ES's tendency for premature covariance collapse in deceptive basins (where $r_k^{\text{perf}} \to 0$). By implicitly lower-bounding the differential entropy of the search distribution via trace($\boldsymbol{C}_k$), it acts as an auxiliary fallback that preserves the structural prerequisites for future improvement.

## 3.3. Reward-Weighted Transfer

Leveraging the reward signal $r_k$ as an indicator of evolutionary quality, we facilitate knowledge transfer by explicitly prioritizing information derived from high-potential tasks. Our strategy is premised on the view that tasks with higher rewards possess superior search dynamics that act as valuable references. We implement this via two reward-weighted mechanisms targeting search direction (mean) and adaptation rate (covariance).

### 3.3.1. GUIDING SEARCH DIRECTION VIA MEAN AGGREGATION

Inspired by the adaptability of pre-trained models, which implies that optimal parameters for potentially related tasks often share common metric features in the solution space, we introduce a reward-weighted mean aggregation mechanism. To guide distributions toward promising regions shared by high-performing tasks (Figure 3), we compute a consolidated mean vector $\bar{\boldsymbol{m}}$:

$$\bar{m}_j = \frac{\sum_{k \in \mathcal{K}_j} r_k \cdot m_{k,j}}{\sum_{k \in \mathcal{K}_j} r_k}, \quad (9)$$

where $\mathcal{K}_j = \{k : n_k \geq j\}$ is the set of tasks containing the $j$-th dimension, and $\bar{\boldsymbol{m}} = [\bar{m}_1, \ldots, \bar{m}_d]^\top$ with $d = \max_k(n_k)$.

We transfer this knowledge by generating $\tau$ external solu-

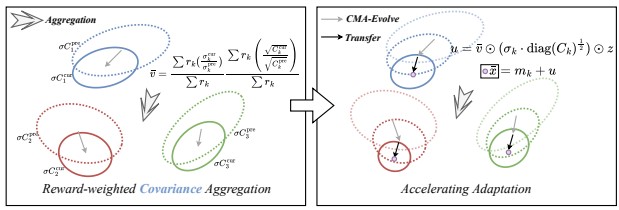

*Figure 4.* Reward-weighted covariance aggregation and transfer for accelerating adaptation.

tions per task. A perturbation vector $u_i$ is first derived to integrate $\bar{m}$:

$$u_i = \bar{m} - m_k + \sigma_k \cdot \sqrt{C_k} \cdot z_i, \quad z_i \sim \mathcal{N}(0, I), \quad (10)$$

where $\bar{m}$ is truncated to match the task's dimensionality. The external solution $\bar{x}_{k,i}$ is then synthesized by scaling $u_i$ to the task's local exploratory scope $d_k$:

$$\bar{x}_{k,i} = m_k + \frac{u_i}{\|u_i\|} \cdot d_k, \quad d_k = \frac{1}{\lambda} \sum_{j=1}^{\lambda} \|x_{k,j} - m_k\|, \quad (11)$$

where $d_k$ represents the average Euclidean distance of the native solutions $\{x_{k,j}\}_{j=1}^{\lambda}$ from the mean $m_k$. This scaling ensures the transferred guidance remains compatible with the task's current landscape characteristics.

### 3.3.2. ACCELERATING ADAPTATION VIA COVARIANCE AGGREGATION

In CMA-ES, the adaptation of step size $\sigma_k$ and covariance matrix $C_k$ is critical for convergence. To transfer successful evolutionary dynamics (*e.g.*, expansion/contraction trends) from fast-converging tasks (Figure 4), we compute an aggregated variation vector $\bar{v}$ based on diagonal elements to decouple transferable scaling trends from task-specific correlations:

$$\bar{v}_j = \underbrace{\frac{\sum_{k=1}^{K} r_k(\sigma_k^{\text{cur}}/\sigma_k^{\text{pre}})}{\sum_{k=1}^{K} r_k}}_{\text{global step size variation}} \cdot \underbrace{\frac{\sum_{k \in \mathcal{K}_j} r_k \left(\sqrt{C_{k,jj}^{\text{cur}}}/\sqrt{C_{k,jj}^{\text{pre}}}\right)}{\sum_{k \in \mathcal{K}_j} r_k}}_{\text{principal diagonal covariance variation}},$$
$$(12)$$

where $\sigma_k^{\text{cur/pre}}$ are the current and previous step sizes, and $C_{k,jj}^{\text{cur/pre}}$ are the diagonal elements of the covariance matrix. $\bar{v} = [\bar{v}_1, \ldots, \bar{v}_d]^\top$ is the aggregated variation vector.

This vector guides the generation of another $\tau$ solutions by scaling the local covariance structure:

$$u_i = \bar{v} \odot (\sigma_k \cdot \sqrt{\text{diag}(C_k)}) \odot z_i, \quad z_i \sim \mathcal{N}(0, I), \quad (13)$$

where $\odot$ denotes element-wise multiplication, $\text{diag}(\cdot)$ denotes diagonal elements, and $\bar{v}$ is truncated to $n_k$ dimensions. The solution is then generated by:

$$\bar{x}_{k,i} = m_k + u_i. \quad (14)$$

This mechanism enables the transfer of distribution variation patterns from rapidly adapting tasks to slower ones without disrupting the target task's specific correlation structure.

### 3.3.3. NEGATIVE TRANSFER MITIGATION

To mitigate the *Multi-Task Curse* under low similarity conditions, we employ a defensive strategy formalized as:

**Definition 3.2** (**Weak Guidance Injection**)**.** We construct evaluation samples by replacing a strictly limited subset of $2\tau$ native solutions with external candidates $\{\bar{x}_i\}_{i=1}^{2\tau}$ derived from aggregated statistics. The injection size is constrained by $\tau \ll \mu$ to ensure minimal interference.

Setting a small $\tau$ ensures that the influence of external solutions remains marginal compared to the native elite set (*e.g.*, $\tau = 1$ is sufficient in practice; see Appendix G). Based on this setup, we establish the following guarantee:

**Proposition 3.3** (**Mitigation of Negative Transfer**)**.** *Let the distribution parameters be updated via the weak guidance injection (Definition 3.2), where a set of external solutions $\mathcal{S}_{trans} = \{\bar{x}_i\}_{i=1}^{2\tau}$ replaces a subset of native candidates. Let $\mathcal{S}_{native}^{\mu}$ denote the set of the top-$\mu$ elite solutions derived purely from the native distribution. If the injected solutions induce inferior performance such that $\min_{\bar{x} \in \mathcal{S}_{trans}} \mathcal{L}(\bar{x}) > \max_{x \in \mathcal{S}_{native}^{\mu}} \mathcal{L}(x)$, then the distribution parameter update $\Delta\theta$ is explicitly decoupled from $\mathcal{S}_{trans}$. Consequently, the algorithm guarantees the mitigation of negative transfer by assigning zero weight to all solutions in $\mathcal{S}_{trans}$.*

*Proof.* See Appendix B. □

> **Remark:** Proposition 3.3 formalizes a *safety valve* mechanism: external solutions undergo competitive ranking and are automatically discarded if maladaptive. This ensures that the algorithm seamlessly reverts to the standard CMA-ES behavior, effectively neutralizing negative transfer risks.

### 3.4. Semantic Policy Parameter Alignment

A primary challenge in applying EMT to RL is the dimensional mismatch caused by diverse state and action spaces. Naive unification often disrupts network semantics, leading to functional misalignment where parameters with distinct roles are improperly mapped. To address this, we propose a semantic policy alignment strategy that partitions the flattened optimization vector $x_k$ (derived from policy parameters $\theta_k$) into *task-specific* and *joint-search* subspaces based on functional roles.

For MLPs in Figure 5, we decompose the parameters into:

- **Joint-Search Subspace ($x^{\text{joint}}$):** Comprises the first-layer bias $b_1$ and all subsequent hidden parameters $\{W_2...W_{n-1}, b_2...b_{n-1}\}$. Notably, $b_1$ is included here as its dimension is determined by the fixed hidden layer size. This forms a semantically consistent backbone that encodes generalized dynamics.

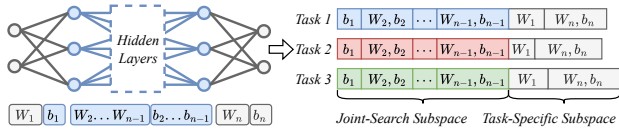

*Figure 5.* Policy parameter alignment for MLPs. The *joint-search* subspace maintains alignment for inter-task transfer, while the *task-specific* subspace accommodates varying input/output dimensions.

- **Task-Specific Subspace ($x^{\text{spec}}$):** Comprises the input weights $W_1$ and output parameters $\{W_n, b_n\}$. Since observation ($n_k^{\text{obs}}$) and action ($n_k^{\text{act}}$) dimensions vary, these layers serve as distinct sensor-motor interfaces.

By establishing a *joint-search* subspace, our approach facilitates knowledge sharing based on semantic consistency, while accommodating the unique evolution of *task-specific* sensor-motor mappings. This structural design offers inherent generalization capabilities for diverse deep learning architectures. Specifically, it allows intermediate representations to function as the shared medium for semantic transfer, leaving the architectural extremities to handle *task-specific* variations. While this framework is adaptable to architectures of arbitrary depth, it currently relies on fixed hidden layer sizes across tasks to maintain valid correspondences in the joint-search subspace.

## 4. Experiments

### 4.1. Experimental Settings

**Tasks.** We evaluate MES-RET on three diverse scenarios: **87 synthetic functions** (Awad et al., 2017), **42 real-world engineering problems** (Kumar et al., 2020), and **18 Gymnasium control tasks** (Towers et al., 2024).

**Baselines.** We compare against 12 representative baseline algorithms, including single-task: **CMA-ES** (Hansen & Ostermeier, 2001), **OpenAI-ES** (Salimans et al., 2017), **L-SHADE** (Tanabe & Fukunaga, 2014), and **CCEF-ECHT** (Li et al., 2024d); multi/many-task: **SBC-MAES** (Liaw & Ting, 2019), **MTES-KG** (Li et al., 2024e), **TNG-SNES** (Li et al., 2024c), **AT-MFEA** (Xue et al., 2022), **EMaTO-MKT** (Liang et al., 2022), and **CEDA-MP** (Zhang et al., 2024); and policy gradient: **PPO** (Schulman et al., 2017), and **A2C** (Mnih et al., 2016) from Stable-Baselines3 (Raffin et al., 2021).

**Variants.** We conduct ablation studies with **w/o-RE** (without reward-weighted evaluation) and **w/o-RT** (without reward-weighted transfer).

**Evaluation Metrics.** Performance is assessed using the **best objective value** for optimization tasks across 30 independent runs and **average cumulative reward** for policy search tasks of 3 random rollouts across 10 independent runs.

*Table 1.* Results on two optimization scenarios (30 independent runs). $+, -, =$ indicate the algorithm is significantly better, worse, or equivalent to the proposed MES-RET. #B $\uparrow$ reports the number of tasks with best results. Rank $\downarrow$ shows the Friedman ranking.

| Synthetic Optimization (87 tasks) | | | | | | Real-World Application (42 tasks) | | | | | |
|---|---|---|---|---|---|---|---|---|---|---|---|
| Algorithm | + | − | = | #B | Rank | Algorithm | + | − | = | #B | Rank |
| **MES-RET** | $\downarrow$ | $\uparrow$ | / | **25** | **3.13** | **MES-RET** | $\downarrow$ | $\uparrow$ | / | **18** | **3.54** |
| w/o-RE | 1 | 17 | 69 | 19 | 3.82 | w/o-RE | 4 | 11 | 27 | 15 | 3.65 |
| w/o-RT | 1 | 7 | 79 | 23 | 3.28 | w/o-RT | 5 | 10 | 27 | 17 | 3.79 |
| CMA-ES | 6 | 21 | 60 | 20 | 3.90 | CMA-ES | 7 | 19 | 16 | 15 | 4.95 |
| SBCMAES | 1 | 83 | 3 | 0 | 9.68 | SBCMAES | 1 | 35 | 6 | 2 | 8.90 |
| MTES-KG | 4 | 43 | 40 | 13 | 4.97 | MTES-KG | 2 | 31 | 9 | 3 | 6.75 |
| TNG-SNES | 7 | 63 | 17 | 7 | 6.70 | TNG-SNES | 0 | 30 | 12 | 5 | 7.05 |
| L-SHADE | 20 | 57 | 10 | 19 | 5.33 | CCEF-ECHT | 9 | 17 | 16 | 17 | 4.23 |
| AT-MFEA | 8 | 72 | 7 | 4 | 7.02 | AT-MFEA | 5 | 29 | 8 | 4 | 6.24 |
| EMaTO-MKT | 5 | 73 | 9 | 4 | 7.17 | CEDA-MP | 4 | 28 | 10 | 4 | 5.90 |

To ensure rigorous comparison, we report the **Wilcoxon rank-sum test** ($\alpha = 0.05$) results for pairwise significance, the **number of best tasks** (#B), and the **Friedman ranking** to summarize global performance.

More detailed settings are provided in Appendix E and F. The full performance results are provided in Appendix I.

### 4.2. Comparison & Ablation on Optimization

Table 1 summarizes the results on optimization tasks. MES-RET achieves significantly better performance than most baselines, with dominant advantages over other multi/many-task optimizers. It achieves the highest number of best-performing tasks (#B = 25 and 18) and lowest Friedman rankings (3.13 and 3.54). Notably, most many-task methods (*e.g.*, SBCMAES, MTES-KG, AT-MFEA) underperform single-task baselines like CMA-ES and L-SHADE, highlighting that MES-RET effectively addresses the *Multi-Task Curse* on low-similarity optimization tasks.

Ablation studies quantify the necessity of each module. The w/o-RE variant exhibits marked deterioration, being significantly outperformed on 17 synthetic and 11 real-world tasks. These statistics substantiate Proposition 3.1 by demonstrating that uniform sampling fails to satisfy the condition required to maximize total improvement. Consequently, the empirical observation $\mathbb{E}[\Delta_{\text{total}}^{\text{uniform}}] < \mathbb{E}[\Delta_{\text{total}}^{\text{weighted}}]$ across benchmarks validates the covariance condition in Eq. (4). In parallel, the degradation in w/o-RT highlights the critical role of reward-weighted transfer in guiding cross-task sharing, where its absence leads to suboptimal convergence.

### 4.3. Comparison & Ablation on Policy Search

Table 2 summarizes the results across 18 Gymnasium tasks. The proposed sep-MES-RET (seperable version of MES-RET) achieves the best overall performance with the lowest ranking (4.03) and the highest count of best-performing tasks (#B = 6, tied with PPO). Compared to the single-task sep-CMA-ES, our method secures 5 significant wins with only 1 loss, effectively mitigating the *Multi-Task Curse*

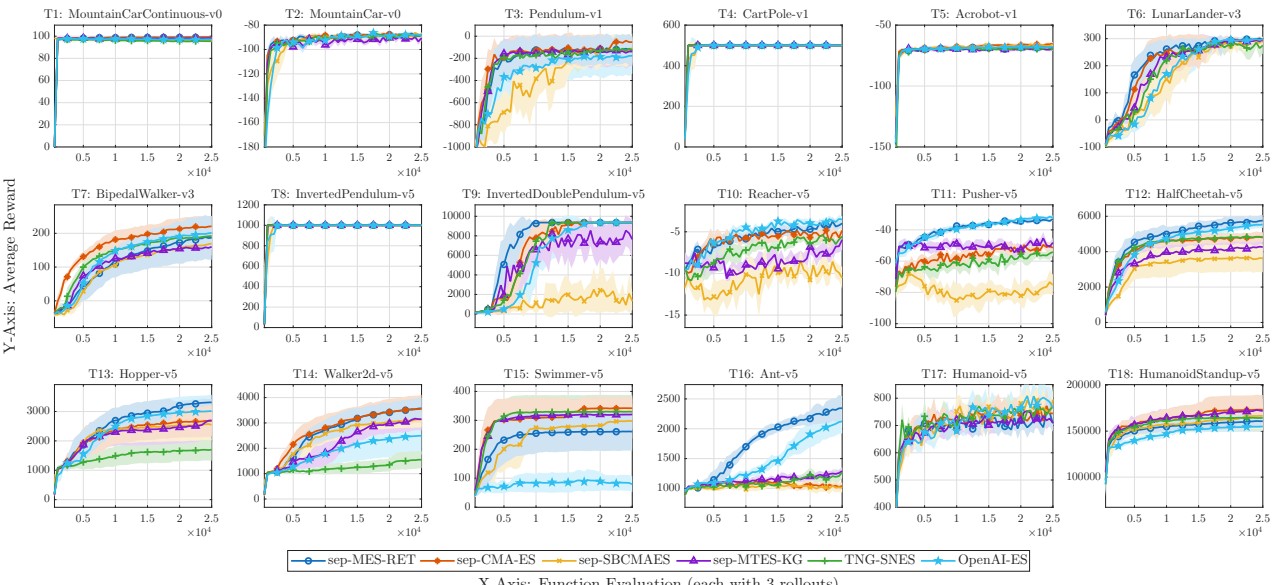

*Figure 6.* Average reward convergence of evolution strategies on 18 Gymnasium tasks. Solid lines show generational mean performance; shaded regions show 95% confidence intervals across 10 independent runs.

*Table 2.* Results on policy search (10 independent runs). $+, -, =$ indicate the algorithm is significantly better, worse, or equivalent to the proposed sep-MES-RET. #B $\uparrow$ reports the number of tasks with best results. Rank $\downarrow$ shows the Friedman ranking.

| Algorithm | $+$ | $-$ | $=$ | #B | Rank |
|---|---|---|---|---|---|
| **sep-MES-RET** | $\downarrow$ | $\uparrow$ | $/$ | 6 | **4.03** |
| w/o-RE | 1 | 3 | 14 | 3 | 5.42 |
| w/o-RT | 3 | 6 | 9 | 4 | 5.08 |
| sep-CMA-ES | 1 | 5 | 12 | 3 | 4.39 |
| sep-SBCMAES | 2 | 7 | 9 | 3 | 6.47 |
| sep-MTES-KG | 0 | 7 | 11 | 2 | 6.31 |
| TNG-SNES | 1 | 5 | 12 | 4 | 4.47 |
| OpenAI-ES | 0 | 4 | 14 | 3 | 4.67 |
| PPO | 3 | 10 | 5 | **6** | 5.75 |
| A2C | 0 | 14 | 4 | 1 | 8.42 |

observed in other multi/many-task baselines. While RL methods like PPO exhibit occasional peaks, they suffer from high variance and significantly more losses (10).

Ablation studies validate the proposed mechanisms. The performance degradation in w/o-RE aligns with the theoretical consistency in Proposition 3.1, confirming that our reward-weighted evaluation is superior to uniform sampling. Furthermore, the substantial rank drop in w/o-RT (5.08) underscores the criticality of the transfer technique, which enables effective cross-task knowledge sharing through semantic policy alignment.

Figure 6 visualizes the convergence curves. sep-MES-RET demonstrates robust convergence, particularly in high-dimensional locomotion tasks (*e.g.*, Ant, HalfCheetah, and Hopper), where it significantly outpaces baselines in both learning speed and final reward. Appendix H provides additional convergence plots with ablation and policy gradient baselines.

## 4.4. Visualization and Analysis of Search Behavior

Our empirical analysis of learned policies yields three key discoveries regarding reward-weighted transfer via semantic policy parameter alignment:

1. *Parameter Correlation:* Figure 8 shows strong correlations in sep-MES-RET's *joint-search subspace* across 18 tasks, while w/o-RT exhibits near-zero correlations. This confirms that sep-MES-RET effectively enables parameter transfer across tasks.

2. *Behavior Transfer:* Figure 7 shows that parameter correlation induces consistent locomotion behaviors across three distinct environments, while w/o-RT develops task-specific inconsistent strategies.

3. *Parameter-Behavior Correspondence:* The semantic policy alignment successfully decouples transferable motion primitives from task-specific adaptations.

## 4.5. Extension to Heterogeneous Architectures

To validate semantic policy parameter alignment across asymmetric and heterogeneous architectures, we conducted an experiment using four continuous control tasks. HalfCheetah and Hopper employ standard MLPs, while Walker2d and Ant utilize a 1D-CNN feature extractor coupled with an MLP policy head.

We implemented a strict topological decoupling: shared parameters (biases of the first fully-connected layer and all parameters of subsequent intermediate MLP layers) are uniformly packed and aligned across tasks. Conversely,

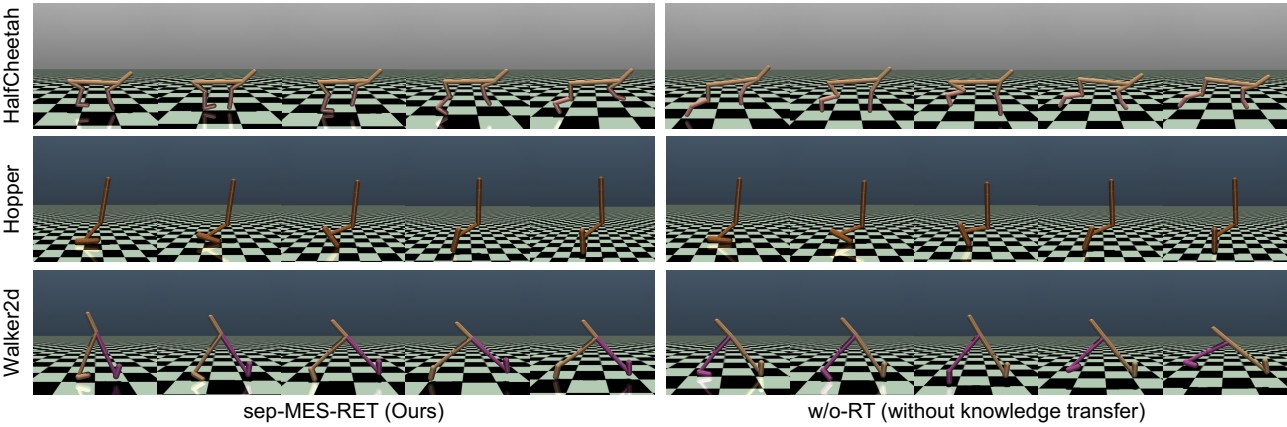

sep-MES-RET (Ours)             w/o-RT (without knowledge transfer)

*Figure 7.* Visualization of the learned locomotion policies by proposed sep-MES-RET (left) and w/o-RT (right) across HalfCheetah, Hopper, and Walker2d environments. sep-MES-RET discovered transferable movement patterns: all three agents learned similar forward propulsion mechanisms using synchronized leg extension and kicking motions. In contrast, w/o-RT, without joint-search subspace based knowledge transfer, developed inconsistent and distinct movement strategies for each environment.

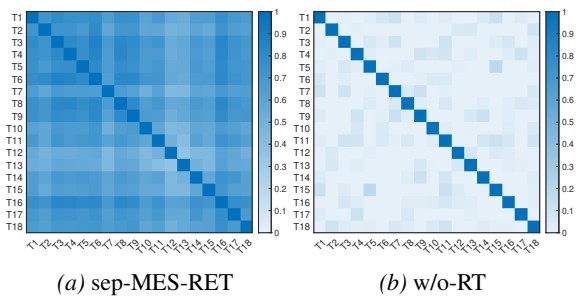

    *(a)* sep-MES-RET          *(b)* w/o-RT

*Figure 8.* Pearson correlation matrix of *joint-search subspace* parameters. sep-MES-RET achieves high cross-task correlations, whereas w/o-RT exhibits near-zero parameter similarity.

task-specific parameters (CNN filters, projection weights, and input/output layers) are isolated.

*Table 3.* Validation on heterogeneous neural architectures.

| Task | Architecture | sep-MES-RET | w/o-RT |
|------|-------------|-------------|--------|
| HalfCheetah-v5 | MLP | **5527.16 ± 525.94** | 4978.99 ± 638.23 |
| Hopper-v5 | MLP | **2913.40 ± 710.51** | 2281.76 ± 878.17 |
| Walker2d-v5 | CNN+MLP | **2845.42 ± 232.21** | 2716.71 ± 703.37 |
| Ant-v5 | CNN+MLP | 1322.88 ± 249.46 | **1397.13 ± 187.65** |

The results (Table 3) show that sep-MES-RET achieves 5527.16 and 2913.40 on HalfCheetah and Hopper, respectively, outperforming the independent w/o-RT baseline (4978.99 and 2281.76). Notably, it also improves Walker2d to 2845.42 (vs. 2716.71 for baseline), demonstrating that latent motor-control semantics can be mapped and positively transferred across highly disparate architectures, provided the terminal decision-making sub-manifolds are rigorously aligned. On Ant, it achieves 1322.88 (vs. 1397.13), indicating a slight deficit but validating that the Weak Guidance Injection successfully detects this semantic misalignment, strictly bounding the negative transfer to prevent catastrophic divergence under severe architectural mismatch.

## 5. Conclusion

This work tackles the *Multi-Task Curse* in evolutionary multi-tasking by proposing MES-RET, which specifically addresses the challenges of evaluation budget dispersion and negative transfer. The proposed reward-weighted evolution strategy dynamically allocates evaluation budgets through a performance-diversity balanced reward mechanism to maximize resource efficiency. Simultaneously, MES-RET facilitates robust knowledge transfer by aggregating mean and covariance statistics across tasks to ensure stable convergence. Extensive experiments demonstrate that MES-RET significantly outperforms state-of-the-art methods across 87 synthetic benchmarks, 42 real-world optimization problems, and 18 policy search environments. Furthermore, the semantic policy alignment strategy effectively resolves dimensional mismatches and functional misalignment, enabling the transfer of locomotion skills across morphologically distinct agents.

We acknowledge that our semantic policy alignment relies on consistent hidden layer dimensions across tasks, limiting its direct applicability to heterogeneous architectures. However, MES-RET serves as a robust black-box many-task optimizer, and its modularity facilitates future integration with policy gradient methods to further enhance sample efficiency in more complex RL tasks.

### Software and Data

The source code of the proposed MES-RET algorithm, along with supplementary files, is available at https://github.com/intLyc/MES-RET

Our implementation relies on the open-source multi-task optimization platform MToP (Li et al., 2026) at https://github.com/intLyc/MTO-Platform

## Acknowledgements

This work was partly supported by the National Natural Science Foundation of China under Grant No. 62576325, the MTI under its AI Centre of Excellence for Manufacturing (AIMfg) under Award No. W25MCMF014, and the A*STAR-NTU Research Joint Lab for Smart and Sustainable Advanced Manufacturing, the Major Research & Development Program of Hubei Province under Grant No. 2025BEB002 and 2023BCA006, the Regional Innovation System Program of Hubei Province under Grant No. 2025EIA022, and the Hubei Superior and Distinctive Discipline Group of "New-Energy Vehicles and Smart Transportation".

## Impact Statement

This paper presents work whose goal is to advance the field of Machine Learning. There are many potential societal consequences of our work, none of which we feel must be specifically highlighted here.

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

## A. Proof of Proposition 3.1 (Effectiveness Condition)

*Proof.* The per-step expected improvement under the uniform allocation strategy, where each task is selected with probability $1/K$, is given by the arithmetic mean of the improvements:

$$\mathbb{E}[\Delta_{\text{step}}^{\text{uniform}}] = \sum_{k=1}^{K} \frac{1}{K} \Delta_k = \frac{1}{K} \sum_{k=1}^{K} \Delta_k. \tag{15}$$

For the reward-weighted strategy, computational resources are allocated according to probabilities $p_k = \frac{r_k}{R_{\text{total}}}$, where $R_{\text{total}} = \sum_{j=1}^{K} r_j$. The per-step expected improvement is thus:

$$\mathbb{E}[\Delta_{\text{step}}^{\text{weighted}}] = \sum_{k=1}^{K} p_k \Delta_k = \sum_{k=1}^{K} \frac{r_k}{R_{\text{total}}} \Delta_k = \frac{1}{R_{\text{total}}} \sum_{k=1}^{K} r_k \Delta_k. \tag{16}$$

Since the total expected improvement over $K$ allocation steps is simply $K$ times the per-step expectation, determining the superiority of the per-step expectation is sufficient to prove the proposition. We seek to demonstrate that $\mathbb{E}[\Delta_{\text{step}}^{\text{weighted}}] \geq \mathbb{E}[\Delta_{\text{step}}^{\text{uniform}}]$. Substituting the expressions derived above, it suffices to show:

$$\frac{1}{R_{\text{total}}} \sum_{k=1}^{K} r_k \Delta_k \geq \frac{1}{K} \sum_{k=1}^{K} \Delta_k. \tag{17}$$

Multiplying both sides by the positive constant $K \cdot R_{\text{total}}$ (noting that $R_{\text{total}} > 0$ and $K > 0$), the inequality becomes:

$$K \sum_{k=1}^{K} r_k \Delta_k \geq R_{\text{total}} \sum_{k=1}^{K} \Delta_k. \tag{18}$$

Substituting $R_{\text{total}} = \sum_{k=1}^{K} r_k$ back into the right-hand side yields:

$$K \sum_{k=1}^{K} r_k \Delta_k \geq \left( \sum_{k=1}^{K} r_k \right) \left( \sum_{k=1}^{K} \Delta_k \right). \tag{19}$$

This final inequality is identical to the assumption stated in Eq. (4). Therefore, provided the non-negative covariance condition holds, the weighted strategy yields a higher expected improvement than the uniform strategy. $\quad\square$

## B. Proof of Proposition 3.3 (Mitigation of Negative Transfer)

*Proof.* The proof relies on the rank-based truncation selection inherent to the CMA-ES update rule and the conservative injection size defined in Definition 3.2.

Consider the modified batch of candidate solutions of size $\lambda$, consisting of $2\tau$ external samples $\mathcal{S}_{\text{trans}}$ and $\lambda - 2\tau$ native samples. The natural gradient updates for the distribution parameters (mean vector $\boldsymbol{m}$ and covariance matrix $\boldsymbol{C}$) are determined by the weighted recombination of the sorted candidates $\boldsymbol{x}_{i:\lambda}$:

$$\boldsymbol{m}_{k+1} = \sum_{i=1}^{\lambda} w_i \boldsymbol{x}_{i:\lambda}, \tag{20}$$

$$\boldsymbol{C}_{k+1} = (1 - c_1 - c_\mu) \boldsymbol{C}_k + c_1 \boldsymbol{p}_c \boldsymbol{p}_c^\top + c_\mu \sum_{i=1}^{\lambda} w_i (\boldsymbol{x}_{i:\lambda} - \boldsymbol{m}_k)(\boldsymbol{x}_{i:\lambda} - \boldsymbol{m}_k)^\top, \tag{21}$$

where $c_1$ and $c_\mu$ are learning rates for the rank-1 and rank-$\mu$ updates, respectively. The recombination weights $w_i$ strictly adhere to the truncation constraints:

$$w_1 \geq \cdots \geq w_\mu > 0, \quad \sum_{j=1}^{\mu} w_j = 1, \quad \text{and} \quad w_i = 0 \text{ for all } i > \mu. \tag{22}$$

The premise $\min_{\bar{\boldsymbol{x}} \in \mathcal{S}_{\text{trans}}} \mathcal{L}(\bar{\boldsymbol{x}}) > \max_{\boldsymbol{x} \in \mathcal{S}_{\text{native}}^{\mu}} \mathcal{L}(\boldsymbol{x})$ implies that every external solution performs strictly worse than the $\mu$-th best native sample. Consequently, all solutions in $\mathcal{S}_{\text{trans}}$ are sorted into ranks strictly greater than $\mu$, i.e., $r(\bar{\boldsymbol{x}}) > \mu$ for all $\bar{\boldsymbol{x}} \in \mathcal{S}_{\text{trans}}$. Furthermore, Definition 3.2 constrains the injection size such that $\tau \ll \mu$. Given that the cardinality of the available native pool is $|\mathcal{S}_{\text{native}}| = \lambda - 2\tau$, we explicitly require the condition $\lambda - 2\tau \geq \mu$ to hold. This ensures that the pool of native samples is mathematically sufficient to fully populate the top-$\mu$ elite set without relying on external candidates.

Due to the truncation selection, the weights associated with the external solutions vanish ($w_{r(\bar{\boldsymbol{x}})} = 0$). Therefore, the contribution of the external set to both parameter updates is explicitly nullified:

$$\sum_{\bar{\boldsymbol{x}} \in \mathcal{S}_{\text{trans}}} w_{r(\bar{\boldsymbol{x}})}(\bar{\boldsymbol{x}} - \boldsymbol{m}_k) = \boldsymbol{0} \quad \text{and} \quad \sum_{\bar{\boldsymbol{x}} \in \mathcal{S}_{\text{trans}}} w_{r(\bar{\boldsymbol{x}})}(\bar{\boldsymbol{x}} - \boldsymbol{m}_k)(\bar{\boldsymbol{x}} - \boldsymbol{m}_k)^\top = \boldsymbol{0}. \tag{23}$$

Thus, the optimization trajectory $(\boldsymbol{m}_k, \boldsymbol{C}_k)$ remains invariant to the inferior external guidance, confirming the intrinsic mitigation of negative transfer. $\square$

## C. Complete Algorithm

Algorithm 2 provides the full implementation of MES-RET, which simultaneously evolves distributions for $K$ different tasks through reward-weighted evaluation and knowledge transfer. The algorithm initializes Gaussian distributions $\mathcal{N}(\boldsymbol{m}_k, \boldsymbol{C}_k)$ with evolution paths for all tasks. Each generation begins by creating $\lambda$ solutions per task using three complementary methods: standard sampling from the task's distribution, mean-aggregation transfer that steers solutions toward a shared global mean $\bar{\boldsymbol{m}}$ while preserving local distribution properties, and covariance-aggregation transfer that adjusts perturbation scales using a shared variation vector $\bar{\boldsymbol{v}}$ to align with cross-task patterns. After evaluating solutions, distributions are updated following standard CMA-ES rules for mean, covariance, and step-size adaptation.

The many-task reward mechanism then calculates per-task rewards $r_k$ that alternate between normalized performance improvements and distribution diversity metrics based on the optimization progress ($e/E$ ratio). These rewards drive two key processes: reward-weighted evaluation allocates extra optimization cycles to promising tasks via categorical sampling, while reward-weighted knowledge transfer updates the shared vectors $\bar{\boldsymbol{m}}$ and $\bar{\boldsymbol{v}}$ by aggregating information across tasks, with higher-reward tasks contributing more significantly. The updated shared vectors are reintegrated during the next generation's sampling phase through direction guidance and adaptation acceleration. This loop repeats until termination, outputting optimized distributions for all tasks.

### C.1. Detailed CMA-Based Evolution Procedure.

To address the background of the core optimizer, we detail the standard Covariance Matrix Adaptation Evolution Strategy (CMA-ES) backbone procedure here, which corresponds to the inner optimization loop in Algorithm 2.

CMA-ES is a state-of-the-art black-box optimizer that maintains a multivariate or separable Gaussian search distribution parameterized by a mean vector $\boldsymbol{m}_k$, a covariance matrix $\boldsymbol{C}_k$, and a global step size $\sigma_k$ for each task $k$. The procedure consists of the following key algorithmic steps per generation:

1. **Sampling and Evaluation**: A population of $\lambda$ candidate solutions is generated by sampling from $\mathcal{N}(\boldsymbol{m}_k, \sigma_k^2 \boldsymbol{C}_k)$. After integrating the external guidance via the proposed transfer mechanisms, each candidate is evaluated on the objective function to obtain its fitness value.

2. **Selection and Recombination (Mean Update)**: The sampled solutions are sorted based on their fitness. The top $\mu$ elite solutions (typically $\mu = \lfloor \lambda/2 \rfloor$) are selected. Their weighted average dictates the new search center, updating the mean $\boldsymbol{m}_k$ towards regions of the search space that yielded better objective values. The recombination weights $\omega_i$ are rank-based and decay logarithmically.

3. **Step-Size Adaptation**: CMA-ES utilizes a conjugate evolution path $\boldsymbol{p}_{\sigma,k}$ to track the trajectory of the mean updates over consecutive generations. If the path is longer than expected under random selection (indicating that consecutive steps are moving in the same direction), the step size $\sigma_k$ is exponentially increased to accelerate convergence. Conversely, if the path is shorter (indicating oscillatory steps), $\sigma_k$ is decreased to facilitate fine-tuning.

4. **Covariance Matrix Adaptation**: A second evolution path $\boldsymbol{p}_{c,k}$ is maintained to capture the anisotropic structural information of the optimization landscape. The covariance matrix $\boldsymbol{C}_k$ is updated using a combination of a rank-1

**Algorithm 2 MES-RET** (Many-Task Evolution Strategy with Reward-Weighted Evaluation and Transfer)

---

**Input**: Tasks number $K$, Sample size $\lambda$, Transfer size $\tau$
**Output**: Distributions $\{\mathcal{N}_k\}_{k=1}^{K}$
    */* Initialization for all tasks */*
1: **for** $k = 1$ to $K$ **do**
2:    Initialize distribution $\mathcal{N}(\boldsymbol{m}_k, \boldsymbol{C}_k)$, step size $\sigma_k$, and evolution paths $\boldsymbol{p}_{\sigma,k} = \mathbf{0}, \boldsymbol{p}_{c,k} = \mathbf{0}$.
3: **end for**
4: Initialize knowledge transfer vectors $\bar{\boldsymbol{m}} = \mathbf{0}, \bar{\boldsymbol{v}} = \mathbf{1}$
5: **while** Termination condition not met **do**
    */* All Tasks Sampling & Evaluation & Update */*
6:   **for** $k = 1$ to $K$ **do**
      */* Gaussian Sampling */*
7:     $\{\boldsymbol{x}_{k,i}\}_{i=1}^{\lambda} \leftarrow \{\boldsymbol{m}_k + \sigma_k \cdot \sqrt{\boldsymbol{C}_k} \cdot \boldsymbol{z}_i\}_{i=1}^{\lambda}, \; \boldsymbol{z}_i \sim \mathcal{N}(\mathbf{0}, \boldsymbol{I})$
      */* Mean-Aggregation Knowledge Transfer */*
8:     $d_k = \frac{1}{\lambda} \sum_{j=1}^{\lambda} \|\boldsymbol{x}_{k,j} - \boldsymbol{m}_k\|$
9:     **for** $i = 1$ to $\tau$ **do**
10:       $\boldsymbol{u}_i = \bar{\boldsymbol{m}} - \boldsymbol{m}_k + \sigma_k \cdot \sqrt{\boldsymbol{C}_k} \cdot \boldsymbol{z}_i, \; \boldsymbol{z}_i \sim \mathcal{N}(\mathbf{0}, \boldsymbol{I})$
11:       $\boldsymbol{x}_{k,i} \leftarrow \boldsymbol{m}_k + \frac{\boldsymbol{u}_i}{\|\boldsymbol{u}_i\|} \cdot d_k$
12:     **end for**
      */* Covariance-Aggregation Knowledge Transfer */*
13:     **for** $i = \tau + 1$ to $2\tau$ **do**
14:       $\boldsymbol{u}_i = \bar{\boldsymbol{v}} \odot (\sigma_k \cdot \sqrt{\mathrm{diag}(\boldsymbol{C}_k)}) \odot \boldsymbol{z}_i, \; \boldsymbol{z}_i \sim \mathcal{N}(\mathbf{0}, \boldsymbol{I})$
15:       $\boldsymbol{x}_{k,i} \leftarrow \boldsymbol{m}_k + \boldsymbol{u}_i$
16:     **end for**
      */* Evaluation and Update */*
17:     $\{f_{k,i}\}_{i=1}^{\lambda} \leftarrow \mathrm{Evaluation}(\{\boldsymbol{x}_{k,i}\}_{i=1}^{\lambda})$
18:     $\{\boldsymbol{y}_{k,i}\}_{i=1}^{\lambda} \leftarrow (\{\boldsymbol{x}_{k,i}\}_{i=1}^{\lambda} - \boldsymbol{m}_k)/\sigma_k$
19:     Sort $\{\boldsymbol{y}_{k,i}\}_{i=1}^{\lambda}$ by objective values $\{f_{k,i}\}_{i=1}^{\lambda}$
20:     $\langle \boldsymbol{y} \rangle_{W,k} \leftarrow \sum_{i=1}^{\mu} \omega_i \boldsymbol{y}_{k,i}$
21:     $\boldsymbol{m}_k \leftarrow \boldsymbol{m}_k + \sigma_k \langle \boldsymbol{y} \rangle_{W,k}$
22:     $\boldsymbol{p}_{\sigma,k} \leftarrow (1 - c_\sigma)\boldsymbol{p}_{\sigma,k} + \sqrt{c_\sigma(2 - c_\sigma)\mu_{\mathrm{eff}}}\, \boldsymbol{C}_k^{-\frac{1}{2}} \langle \boldsymbol{y} \rangle_{W,k}$
23:     $\sigma_k \leftarrow \sigma_k \cdot \exp\left( \frac{c_\sigma}{d_\sigma} \left( \frac{\|\boldsymbol{p}_{\sigma,k}\|}{\mathbb{E}[\|\mathcal{N}(\mathbf{0}, \boldsymbol{I})\|]} - 1 \right) \right)$
24:     $\boldsymbol{p}_{c,k} \leftarrow (1 - c_c)\boldsymbol{p}_{c,k} + h_\sigma \sqrt{c_c(2 - c_c)\mu_{\mathrm{eff}}} \langle \boldsymbol{y} \rangle_{W,k}$
25:     $\boldsymbol{C}_k \leftarrow (1 - c_1 - c_\mu \sum_{i=1}^{\mu} \omega_i)\boldsymbol{C}_k + c_1 \boldsymbol{p}_{c,k} \boldsymbol{p}_{c,k}^{\top} + c_\mu \sum_{i=1}^{\mu} \omega_i \boldsymbol{y}_{k,i} \boldsymbol{y}_{k,i}^{\top}$
26:   **end for**
    */* Many-Task Reward Calculation */*
27:   **if** $u \geq \frac{e}{E}$, where $u \sim \mathcal{U}(0, 1)$ **then**
28:     $\{r_k\}_{k=1}^{K} \leftarrow \mathrm{Softmax}(\{\frac{f_k^{\mathrm{pre}} - f_k^{\mathrm{cur}}}{|f_k^{\mathrm{init}} - f_k^{\mathrm{pre}}| + \epsilon}\}_{k=1}^{K})$ with min-max normalization
29:   **else**
30:     $\{r_k\}_{k=1}^{K} \leftarrow \{\frac{\sigma_k \cdot \mathrm{trace}(\boldsymbol{C}_k)}{n_k}\}_{k=1}^{K}$
31:   **end if**
    */* Reward-Weighted Knowledge Aggregation */*
32:   $\bar{\boldsymbol{m}} \leftarrow \{\frac{\sum_{k \in \mathcal{K}_j} r_k \cdot m_{k,j}}{\sum_{k \in \mathcal{K}_j} r_k}\}_{j=1}^{\max(n_{1:K})}$
33:   $\bar{\boldsymbol{v}} \leftarrow \{\frac{\sum_{k=1}^{K} r_k(\sigma_k^{\mathrm{cur}}/\sigma_k^{\mathrm{pre}})}{\sum_{k=1}^{K} r_k} \cdot \frac{\sum_{k \in \mathcal{K}_j} r_k \left(\sqrt{C_{k,jj}^{\mathrm{cur}}}/\sqrt{C_{k,jj}^{\mathrm{pre}}}\right)}{\sum_{k \in \mathcal{K}_j} r_k}\}_{j=1}^{\max(n_{1:K})}$
    */* Reward-Weighted Evaluation */*
34:   $\{p_k\} \leftarrow \{r_k / \sum_{j=1}^{K} r_j\}_{k=1}^{K}$
35:   **for** 1 to $K$ **do**
36:     Task $t \leftarrow \mathrm{Categorical\text{-}Sampling}(\{p_k\}_{k=1}^{K})$
37:     $\{\boldsymbol{x}_{t,i}\}_{i=1}^{\lambda} \leftarrow \mathrm{Sampling}(\mathcal{N}_t)$ following the same sampling procedure as above
38:     $\{f_{t,i}\}_{i=1}^{\lambda} \leftarrow \mathrm{Evaluation}(\{\boldsymbol{x}_{t,i}\}_{i=1}^{\lambda})$
39:     $\mathcal{N}_t \leftarrow \mathrm{Update}(\mathcal{N}_t, \{\boldsymbol{x}_{t,i}\}_{i=1}^{\lambda}, \{f_{t,i}\}_{i=1}^{\lambda})$ following the same update procedure as above
40:   **end for**
41: **end while**

update (utilizing the evolution path $\boldsymbol{p}_{c,k}$ to exploit historical correlations across generations) and a rank-$\mu$ update (utilizing the variance of the current elite population). This adaptation allows the search distribution to approximate the inverse Hessian matrix of the objective function, enabling efficient navigation of ill-conditioned landscapes.

By integrating this robust CMA-ES procedure as the core backbone, MES-RET inherits its invariance properties and capability to handle complex, non-separable optimization landscapes, while systematically scaling it to many-task scenarios via dynamic resource allocation and reward-weighted knowledge transfer.

## D. Time Complexity Analysis

The time complexity of MES-RET depends on the number of tasks $K$, maximum dimensionality $n$, and sample size $\lambda$. We analyze the costs broken down by evolutionary phases and algorithmic overhead.

**Evolutionary Phases:** The algorithm executes two distinct evolution phases per generation: the *All-Task Evolution* and the *Reward-Weighted Evaluation*.

- *All-Task Evolution:* Executes one evolution step for all $K$ tasks.

- *Reward-Weighted Evaluation:* Allocates an additional budget of $K$ evaluations based on task potential.

Consequently, the algorithm performs a total of $2K$ evolution updates per generation. In the full covariance implementation, the dominant operations per task are multivariate Gaussian sampling and covariance matrix updates, costing $\mathcal{O}(\lambda n^2)$. Solution sorting adds $\mathcal{O}(\lambda \log \lambda)$. Thus, the total complexity for the evolutionary phases is $\mathcal{O}(K(\lambda n^2 + \lambda \log \lambda))$.

**Algorithmic Overhead:** The additional costs arise from the reward-weighted mechanisms:

- *Reward Calculation & Aggregation:* Computing rewards and aggregating global statistics involves vector operations costing $\mathcal{O}(Kn)$.

- *Knowledge Transfer (External Sampling):* Generating $\tau$ external solutions requires transforming standard normal vectors via the covariance matrix ($\sqrt{\boldsymbol{C_k}} \cdot \boldsymbol{z}_i$). For full covariance, this matrix-vector multiplication costs $\mathcal{O}(n^2)$ per sample, leading to a total transfer overhead of $\mathcal{O}(K\tau n^2)$. For separable covariance, this reduces to element-wise operations costing $\mathcal{O}(K\tau n)$.

The total overhead for full covariance is therefore $\mathcal{O}(K\tau n^2 + Kn)$.

**Overall Complexity:** Combining these components, the total per-iteration complexity for full covariance MES-RET is $\mathcal{O}(K\lambda n^2 + K\tau n^2 + K\lambda \log \lambda)$. Since the external injection size is designed to be small ($\tau \ll \lambda$), the term $K\tau n^2$ is dominated by $K\lambda n^2$. Thus, the overall complexity simplifies to $\mathcal{O}(K\lambda n^2 + K\lambda \log \lambda)$. In high-dimensional settings where separable covariance is employed, the complexity becomes $\mathcal{O}(K\lambda n + K\tau n + K\lambda \log \lambda)$, which similarly simplifies to $\mathcal{O}(K\lambda n + K\lambda \log \lambda)$ due to the small $\tau$.

In summary, regarding the dominant computational terms, MES-RET maintains the same asymptotic time complexity class as the standard CMA-ES ($\mathcal{O}(K\lambda n^2)$) and sep-CMA-ES ($\mathcal{O}(K\lambda n)$) while introducing only minor overhead from the reward-weighted mechanisms.

## E. Task Settings

This section provides detailed descriptions of the tasks used in our experiments, including synthetic benchmarks, real-world engineering problems, and Gymnasium reinforcement learning environments.

### E.1. Synthetic Benchmark Tasks

Table 4 summarizes the synthetic benchmark tasks used in our experiments. The tasks are grouped into four categories: unimodal, multimodal, hybrid, and composition functions. Each category includes problems with varying dimensionalities ($n = 10, 30, 50$), following the CEC 2017 single-objective competition (Awad et al., 2017). Each task is defined by a mathematical function that is shifted and rotated to increase the optimization difficulty and reduce inter-task similarity.

*Table 4.* Overview of the 87 synthetic unconstrained optimization tasks. These tasks are derived from 29 base functions with diverse landscapes, each generating 3 tasks of varying dimensions ($n$). Hybrid functions are constructed by combining multiple base functions of different dimensions, while composition functions are formed by linearly combining base functions with different weights. All tasks are shifted and rotated using different vectors and matrices.

| Task | Description | $n$ |
|---|---|---|
| | **Unimodal Functions** | |
| T1, T30, T59 | Shifted and Rotated Bent Cigar | 10, 30, 50 |
| T2, T31, T60 | Shifted and Rotated Zakharov | 10, 30, 50 |
| | **Multimodal Functions** | |
| T3, T32, T61 | Shifted and Rotated Rosenbrock | 10, 30, 50 |
| T4, T33, T62 | Shifted and Rotated Rastrigin | 10, 30, 50 |
| T5, T34, T63 | Shifted and Rotated Expanded Scaffer F6 | 10, 30, 50 |
| T6, T35, T64 | Shifted and Rotated Lunacek Bi-Rastrigin | 10, 30, 50 |
| T7, T36, T65 | Shifted and Rotated Non-Continuous Rastrigin | 10, 30, 50 |
| T8, T37, T66 | Shifted and Rotated Levy | 10, 30, 50 |
| T9, T38, T67 | Shifted and Rotated Schwefel | 10, 30, 50 |
| | **Hybrid Functions:** $F = f_{1,[1,d_1]} + f_{2,[d_1,d_2]} + \ldots$ | |
| T10, T39, T68 | Hy1: Zakharov, Rosenbrock, Rastrigin | 10, 30, 50 |
| T11, T40, T69 | Hy2: Elliptic, Schwefel, Bent Cigar | 10, 30, 50 |
| T12, T41, T70 | Hy3: Bent Cigar, Rosenbrock, Lunacek Bi-Rastrigin | 10, 30, 50 |
| T13, T42, T71 | Hy4: Elliptic, Ackley, Schaffer F6, Rastrigin | 10, 30, 50 |
| T14, T43, T72 | Hy5: Bent Cigar, HGBat, Rastrigin, Rosenbrock | 10, 30, 50 |
| T15, T44, T73 | Hy6: Schaffer F6, HGBat, Rosenbrock, Schwefel | 10, 30, 50 |
| T16, T45, T74 | Hy7: Katsuura, Ackley, Griewank, Schwefel, Rastrigin | 10, 30, 50 |
| T17, T46, T75 | Hy8: Elliptic, Ackley, Rastrigin, HGBat, Discus | 10, 30, 50 |
| T18, T47, T76 | Hy9: Bent Cigar, Rastrigin, Griewank, Weierstrass, Schaffer F6 | 10, 30, 50 |
| T19, T48, T77 | Hy10: Happycat, Katsuura, Ackley, Rastrigin, Schwefel, Schaffer F6 | 10, 30, 50 |
| | **Composition Functions:** $F = w_1 f_1 + w_2 f_2 + \ldots$ | |
| T20, T49, T78 | Co1: Rosenbrock, Elliptic, Rastrigin | 10, 30, 50 |
| T21, T50, T79 | Co2: Rastrigin, Griewank, Schwefel | 10, 30, 50 |
| T22, T51, T80 | Co3: Rosenbrock, Ackley, Schwefel, Rastrigin | 10, 30, 50 |
| T23, T52, T81 | Co4: Ackley, Elliptic, Griewank, Rastrigin | 10, 30, 50 |
| T24, T53, T82 | Co5: Rastrigin, Happycat, Ackley, Discus, Rosenbrock | 10, 30, 50 |
| T25, T54, T83 | Co6: Schaffer F6, Schwefel, Griewank, Rosenbrock, Rastrigin | 10, 30, 50 |
| T26, T55, T84 | Co7: HGBat, Rastrigin, Schwefel, Bent Cigar, Elliptic, Schaffer F6 | 10, 30, 50 |
| T27, T56, T85 | Co8: Ackley, Griewank, Discus, Rosenbrock, HappyCat, Schaffer F6 | 10, 30, 50 |
| T28, T57, T86 | Co9: Hy5, Hy6, Hy7 | 10, 30, 50 |
| T29, T58, T87 | Co10: Hy5, Hy8, Hy9 | 10, 30, 50 |

The number of function evaluations $E$ is set to $K \cdot 50,000$. The sample size $\lambda$ (or population size $N$) is fixed at 100 for algorithms on each task. The benchmark includes $K = 87$ tasks, each defined by a distinct objective function with a unique global optimum. These tasks are designed to evaluate the ability of optimization algorithms to address diverse challenges, including distinct landscapes, varying dimensionalities, and multimodality.

### E.2. Real-World Engineering Tasks

Table 5 summarizes the 42 real-world constrained optimization tasks used in our experiments. These tasks originate from a variety of engineering domains, including chemical processes, process synthesis and design, mechanical engineering, power electronics, and livestock feed ration optimization (Kumar et al., 2020). Each task is formulated as a constrained optimization problem, where the objective is to optimize a set of decision variables subject to both inequality and equality constraints.

The number of function evaluations $E$ is set to $K \cdot 50,000$. The sample size $\lambda$ (or population size $N$) is fixed at 100 for algorithms on each task. The benchmark consists of $K = 42$ tasks, each representing a distinct real-world engineering problem. These tasks are designed to evaluate the capability of optimization algorithms in handling diversified constraints, complex landscapes, and varying dimensionalities.

### E.3. Gymnasium Environments

Table 6 summarizes the 18 reinforcement learning tasks from the Gymnasium benchmark used in our experiments. These tasks are grouped into three categories: classical control, Box2D, and MuJoCo (Towers et al., 2024). Each task is defined by its observation and action spaces, with the action space being either continuous or discrete depending on the environment.

One function evaluation corresponds to three random rollouts in ESs, with each rollout limited to a maximum of $1,000$ steps. The total number of evaluations is set to $E = K \cdot 25,000$, resulting in $K \cdot 75,000$ rollouts. The sample size $\lambda$ (or population size $N$) is fixed at 50 for algorithms on each task.

Policy gradient methods (PPO and A2C) use a stopping criterion of $K \cdot 10,000,000$ environment steps. While this interaction count is lower than the theoretical maximum for ES methods, this threshold was explicitly selected to align the wall-clock runtime (approx. 30 hours) across all baselines. This setting ensures a fair comparison of computational efficiency under fixed hardware constraints, acknowledging that ES methods leverage massive parallelization to compensate for their lower sample efficiency compared to gradient-based approaches. Furthermore, we empirically verified that (Figure 9) this budget is sufficient for PPO and A2C to reliably reach convergence on the tested tasks, ensuring that their performance is not artificially capped by the stopping criterion.

The benchmark includes $K = 18$ tasks, each representing a distinct reinforcement learning environment. These tasks are designed to evaluate the ability of algorithms to address challenges such as different action spaces, high-dimensional policy parameterizations, and diverse reward structures.

Policy networks are implemented as MLPs with two hidden layers. We adopt distinct hidden layer sizes to ensure each optimization paradigm operates in its effective regime: for ES-based methods, each hidden layer contains 16 neurons to mitigate the curse of dimensionality inherent in black-box optimization over large parameter spaces; for policy gradient methods, 64 neurons are used to leverage their capacity for representation learning via backpropagation. All hidden layers use the *Tanh* activation function. The output layer is adapted to the action space: for continuous actions, it produces a scalar for each action dimension; for discrete actions, it outputs logits over the action classes.

### E.4. Experimental Protocol

To ensure statistical reliability, all synthetic and real-world engineering tasks are evaluated over **30 independent runs**. For policy search tasks, we conduct **10 independent runs**, where the performance of each policy is calculated as the average cumulative reward over 3 rollouts. One rollout represents a complete episode with a maximum length of 1,000 steps and a discount factor $\gamma = 1$.

All experiments are conducted on a Mac mini equipped with an Apple M4 Pro chip and 64 GB of RAM. The code is implemented in MATLAB and Python, and rollout evaluations for ESs are parallelized across CPU cores.

*Table 5.* Overview of the 42 real-world constrained optimization tasks. Here, $n$ represents the number of decision variables, $g$ denotes the number of inequality constraints, and $h$ indicates the number of equality constraints.

| Task | Description | $n$ | $g$ | $h$ |
|------|-------------|-----|-----|-----|
| | **Industrial chemical processes** | | | |
| T1 | Heat exchanger network design (case 1) | 9 | 0 | 8 |
| T2 | Heat exchanger network design (case 2) | 11 | 0 | 9 |
| T3 | Optimal operation of Alkylation unit | 7 | 14 | 0 |
| T4 | Reactor network design | 6 | 1 | 4 |
| T5 | Haverly's pooling problem | 9 | 2 | 4 |
| | **Process synthesis and design problems** | | | |
| T6 | Process synthesis problem | 2 | 2 | 0 |
| T7 | Process synthesis and design problem | 3 | 1 | 1 |
| T8 | Process flow sheeting problem | 3 | 3 | 0 |
| T9 | Two-reactor problem | 7 | 4 | 4 |
| T10 | Process synthesis problem | 7 | 9 | 0 |
| T11 | Process design problem | 5 | 3 | 0 |
| T12 | Multi-product batch plant | 10 | 10 | 0 |
| | **Mechanical engineering problem** | | | |
| T13 | Weight minimization of a speed reducer | 7 | 11 | 0 |
| T14 | Optimal design of industrial refrigeration system | 14 | 15 | 0 |
| T15 | Tension/compression spring design (case 1) | 3 | 3 | 0 |
| T16 | Pressure vessel design | 4 | 4 | 0 |
| T17 | Welded beam design | 4 | 5 | 0 |
| T18 | Three-bar truss design problem | 2 | 3 | 0 |
| T19 | Multiple disk clutch brake design problem | 5 | 7 | 0 |
| T20 | Planetary gear train design optimization problem | 9 | 10 | 1 |
| T21 | Step-cone pulley problem | 5 | 8 | 3 |
| T22 | Robot gripper problem | 7 | 7 | 0 |
| T23 | Hydro-static thrust bearing design problem | 4 | 7 | 0 |
| T24 | Four-stage gearbox problem | 22 | 86 | 0 |
| T25 | 10-bar truss design | 10 | 3 | 0 |
| T26 | Rolling element bearing | 10 | 9 | 0 |
| T27 | Gas transmission compressor design | 4 | 1 | 0 |
| T28 | Tension/compression spring design (case 2) | 3 | 8 | 0 |
| T29 | Gear train design Problem | 4 | 1 | 1 |
| T30 | Himmelblau's function | 5 | 6 | 0 |
| T31 | Topology optimization | 30 | 30 | 0 |
| | **Power Electronic Problems** | | | |
| T32 | Wind farm layout problem | 30 | 91 | 0 |
| T33 | Synchronous optimal pulse-width modulation (SOPM) for 3-level invereters | 25 | 24 | 1 |
| T34 | SOPM for 5-level inverters | 25 | 24 | 1 |
| T35 | SOPM for 7-level inverters | 25 | 24 | 1 |
| T36 | SOPM for 9-level inverters | 30 | 29 | 1 |
| T37 | SOPM for 11-level inverters | 30 | 29 | 1 |
| T38 | SOPM for 13-level inverters | 30 | 29 | 1 |
| | **Livestock Feed Ration Optimization** | | | |
| T39 | Beef Cattle (case 1) | 59 | 14 | 1 |
| T40 | Beef Cattle (case 2) | 59 | 14 | 1 |
| T41 | Beef Cattle (case 3) | 59 | 14 | 1 |
| T42 | Beef Cattle (case 4) | 59 | 14 | 1 |

*Table 6.* Overview of the 18 Gymnasium reinforcement learning tasks. The observation and action columns specify the dimensionality and type, where "Con." denotes continuous space and "Dis." denotes discrete space.

| Task | Description | Observation | | Action | |
|------|-------------|-------------|------|--------|------|
| | **Classical Control** | | | | |
| T1 | MountainCarContinuous-v0 | 2 | Con. | 1 | Con. |
| T2 | MountainCar-v0 | 2 | Con. | 3 | Dis. |
| T3 | Pendulum-v1 | 3 | Con. | 1 | Con. |
| T4 | CartPole-v1 | 4 | Con. | 2 | Dis. |
| T5 | Acrobot-v1 | 6 | Con. | 3 | Dis. |
| | **Box2D** | | | | |
| T6 | LunarLander-v3 | 8 | Con. | 4 | Dis. |
| T7 | BipedalWalker-v3 | 24 | Con. | 4 | Con. |
| | **MuJoCo** | | | | |
| T8 | InvertedPendulum-v5 | 4 | Con. | 1 | Con. |
| T9 | InvertedDoublePendulum-v5 | 9 | Con. | 1 | Con. |
| T10 | Reacher-v5 | 10 | Con. | 2 | Con. |
| T11 | Pusher-v5 | 23 | Con. | 7 | Con. |
| T12 | HalfCheetah-v5 | 17 | Con. | 6 | Con. |
| T13 | Hopper-v5 | 11 | Con. | 3 | Con. |
| T14 | Walker2d-v5 | 17 | Con. | 6 | Con. |
| T15 | Swimmer-v5 | 8 | Con. | 2 | Con. |
| T16 | Ant-v5 | 105 | Con. | 8 | Con. |
| T17 | Humanoid-v5 | 348 | Con. | 17 | Con. |
| T18 | HumanoidStandup-v5 | 378 | Con. | 17 | Con. |

*Table 7.* Overview of the baseline algorithms. The first column lists the algorithm names, the second provides a brief description, the third indicates whether the algorithm handles single, multi, or many tasks, and the last three columns (SO, CO, PS) indicate the algorithm's applicability to synthetic optimization, constrained optimization, and policy search tasks, respectively. The Type column specifies the algorithm category.

| Algorithm | Description | Task | SO | CO | PS | Type |
|-----------|-------------|------|----|----|----|------|
| CMA-ES (Hansen & Ostermeier, 2001) | Covariance Matrix Adaptation Evolution Strategy | Single | ✓ | ✓ | ✓ | |
| OpenAI-ES (Salimans et al., 2017) | OpenAI Evolution Strategy with Stochastic Gradient Descent | Single | | | ✓ | Evolution |
| SBCMAES (Liaw & Ting, 2019) | Many-Task CMA-ES based on Symbiosis in Biocoenosis | Many | ✓ | ✓ | ✓ | Strategy |
| MTES-KG (Li et al., 2024e) | Multi-Task CMA-ES with Knowledge-Guided External Sampling | Multi | ✓ | ✓ | ✓ | |
| TNG-SNES (Li et al., 2024c) | Many-Task Natural ES with Task-Averaged Natural Gradient Transfer | Many | ✓ | ✓ | ✓ | |
| L-SHADE (Tanabe & Fukunaga, 2014) | Success History based Adaptive DE with Linear Population Reduction | Single | ✓ | | | Differential |
| CCEF-ECHT (Li et al., 2024d) | Competitive and Cooperative DE for Constrained Optimization | Single | | ✓ | | Evolution |
| CEDA-MP (Zhang et al., 2024) | Constrained Multi-Task DE via Co-Evolution and Domain Adaptation | Multi | | ✓ | | |
| AT-MFEA (Xue et al., 2022) | Affine Transformation Enhanced Heterogeneous Multi-Task GA | Multi | ✓ | ✓ | | Genetic |
| EMaTO-MKT (Liang et al., 2022) | Many-Task GA based on Multi-Source Knowledge Transfer | Many | ✓ | | | Algorithm |
| PPO (Schulman et al., 2017) | Proximal Policy Optimization with Clipped Surrogate Objective | Single | | | ✓ | Policy |
| A2C (Mnih et al., 2016) | Advantage Actor-Critic with Generalized Advantage Estimation | Single | | | ✓ | Gradient |

# F. Baseline Algorithms

Table 7 presents an overview of the baseline algorithms evaluated in our experiments. The algorithms are organized into four categories: evolution strategies, differential evolution, genetic algorithms, and policy gradient methods. For each algorithm, we provide a brief description and indicate its applicability to synthetic optimization (SO), constrained optimization (CO), and policy search (PS) tasks.

Given the high-dimensional nature of the policy search tasks (up to thousands of parameters), calculating the full covariance matrix is computationally prohibitive. Therefore, for all ES-based methods applied to Gymnasium tasks (including MES-RET, CMA-ES, SBCMAES, and MTES-KG), we adopt the separable variant with "sep-" using diagonal covariance matrix adaptation (Ros & Hansen, 2008). Note that all ES-based methods for policy search tasks utilize the proposed semantic policy parameter alignment technique (Section 3.4) for knowledge transfer, ensuring fair comparison.

For constrained optimization tasks (*i.e.*, real-world engineering problems), several algorithms (including MES-RET, CMA-ES, SBCMAES, TNG-SNES, and AT-MFEA) are adapted to handle constraints using the constraint dominance principle. Under this principle, solutions with fewer constraint violations are preferred; if violations are equal, selection is based on objective function values. The hyperparameters of each baseline algorithm are either adopted from the original publications or empirically tuned through preliminary experiments to ensure fair comparison.

For all ES-based method, the initial step size is set to $\sigma_0 = 0.3(U - L)$, where $U$ and $L$ denote the upper and lower bounds of the search space, respectively. For CMA-ES-based methods (MES-RET, CMA-ES, SBCMAES, and MTES-KG), commonly used hyperparameters are configured as follows: parent number $\mu = \lfloor \lambda/2 \rfloor$, effective factor $\mu_{\text{eff}} = 1/\sum_{j=1}^{\mu} \omega_j^2$, evolution path parameters $c_c = \frac{4+\mu_{\text{eff}}/n}{4+n+2\mu_{\text{eff}}/n}$, learning rates for full covariance matrix adaptation $c_1 = \frac{2}{(n+1.3)^2+\mu_{\text{eff}}}$ and $c_\mu = \min(1 - c_1, \ 2 \cdot \frac{\mu_{\text{eff}}-2+1/\mu_{\text{eff}}}{(n+2)^2+\mu_{\text{eff}}})$, learning rates for diagonal covariance matrix adaptation $c_{\text{cov}} = \frac{1}{\mu_{\text{eff}}} \cdot \frac{2}{(n+\sqrt{2})^2} + \left(1 - \frac{1}{\mu_{\text{eff}}}\right) \cdot \min\left(1, \ \frac{2\mu_{\text{eff}}-1}{(n+2)^2+\mu_{\text{eff}}}\right)$, and step size control parameters $c_\sigma = \frac{\mu_{\text{eff}}+2}{n+\mu_{\text{eff}}+5}$, $d_\sigma = 1 + c_\sigma + 2 \cdot \max\left(\sqrt{\frac{\mu_{\text{eff}}-1}{n+1}}, \ 0\right)$.

The algorithm-specific hyperparameters used in our experiments are summarized as follows:

- **MES-RET**: $\tau = 1$.

- **OpenAI-ES**: optimizer using Adam with $lr$ and $\sigma$ exponential decay, $lr = 0.01$, $lr_{\text{decay}} = 0.1$, $sigma_{\text{decay}} = 0.2$, $\beta_1 = 0.9$, $\beta_2 = 0.999$.

- **SBCMAES**: $Benefit = 0.25$, $Harm = 0.5$.

- **MTES-KG**: $\tau_0 = 2$, $\alpha = 0.5$, $adjGap = 50$.

- **TNG-SNES**: $\rho_0 = 0.1$, $\alpha_0 = 0.7$, $gap = 100$, $\eta_\mu = 1$, $\eta_\sigma = \frac{3+\log(n)}{5\sqrt{n}}$.

- **L-SHADE**: $p = 0.2$, $H = 5$, $R = 18$, $A = 2.1$.

- **CCEF-ECHT**: $PR_0 = 0.1$, $\alpha = 0.5$, $\beta = 0.1$, $P_{min} = 0.1$, $RH = 10$.

- **CEDA-MP**: $rmp = 0.15$, $\mu_C = 2$, $\mu_M = 5$.

- **AT-MFEA**: $rmp = 0.3$, $\mu_C = 2$, $\mu_M = 5$.

- **EMaTO-MKT**: $amp_0 = 0.9$, $\sigma = 1$, $K = 10$, $KTN = 5$, $\mu_C = 2$, $\mu_M = 5$.

- **PPO**: $lr = 0.0003$, $n_{\text{steps}} = 1024$, $batch\_size = 64$, $\gamma = 0.99$, $gae_\lambda = 0.95$, $clip\_range = 0.2$, $ent\_coef = 0.01$, $max\_grad\_norm = 0.5$.

- **A2C**: $lr = 0.0007$, $n_{\text{steps}} = 5$, $\gamma = 0.99$, $gae_\lambda = 0.95$, $ent\_coef = 0.01$, $vf\_coef = 0.5$, $max\_grad\_norm = 0.5$.

## G. Parameter Sensitivity Analysis

*Table 8.* Friedman test of parameter $\tau$ sensitivity analysis.

| | **Synthetic** | | **Real-World** | | **Policy Search** | |
|---|---|---|---|---|---|---|
| Parameter | Rank | $p$-value | Rank | $p$-value | Rank | $p$-value |
| $\tau = 1/3$ | 2.84 | 0.1955 | 3.03 | 0.0976 | 3.12 | 0.1791 |
| $\tau = 1/2$ | **2.53** | — | 2.99 | 0.2758 | 3.02 | 0.3704 |
| $\tau = 1$ | 2.90 | 0.1310 | **2.93** | — | 3.09 | 0.2203 |
| $\tau = 2$ | 3.32 | 0.0011 | 3.01 | 0.1880 | **2.81** | — |
| $\tau = 3$ | 3.41 | 0.0003 | 3.04 | 0.0631 | 2.96 | 0.5093 |

Table 8 presents the results of the Friedman test conducted to analyze the sensitivity of the parameter $\tau$ in MES-RET across different task sets. The table includes the average rankings and $p$-values for each value of $\tau$. We specify that for the regime $\tau \in (0, 1)$, $\tau$ is treated as the probability of injecting a single external solution per generation. For $\tau \geq 1$, $\tau$ denotes the deterministic integer count of transferred solutions as defined in the standard algorithm.

The results indicate that $\tau = 1/2$ performs best on synthetic tasks, while $\tau = 1$ is optimal for real-world tasks. For policy search tasks, $\tau = 2$ yields the best performance. Overall, although the optimal $\tau$ value varies slightly across task categories, the performance differences among $\tau \in \{1/2, 1, 2\}$ are generally small and statistically insignificant in most cases. Therefore, in all comparative experiments throughout the paper, we fix $\tau = 1$ as a default setting.

## H. Convergence Curves

Figures 9 and 10 depict the convergence curves of average rewards achieved by the proposed MES-RET method combined with policy gradient methods and its component variants across 18 reinforcement learning tasks. The x-axis denotes training progress, while the y-axis shows the average reward averaged over three random rollouts. Solid lines represent the mean performance of sampled policies per generation, and shaded regions indicate 95% confidence intervals based on 10 independent runs.

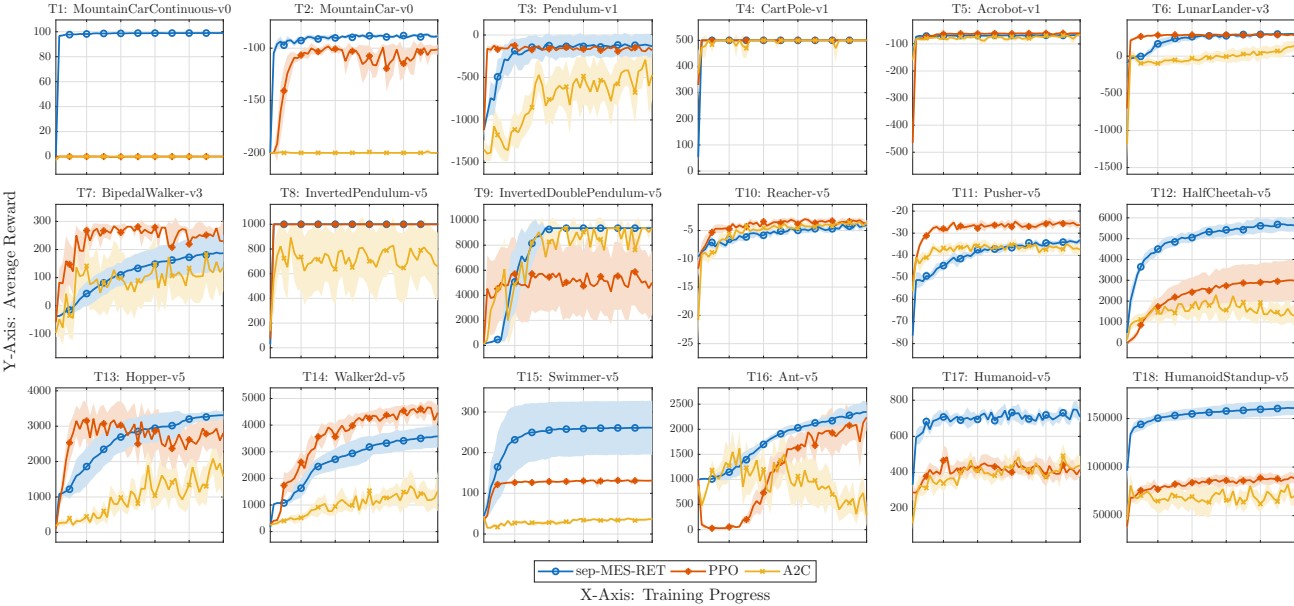

*Figure 9.* Convergence curves of average rewards obtained by the proposed MES-RET with policy gradient method on the 18 reinforcement learning tasks. The x-axis denotes the training progress, while the y-axis shows the average reward return over 3 random rollouts. Solid lines represent the mean performance of the sampled policies per generation, and the shaded areas indicate the 95% confidence intervals computed from 10 independent runs.

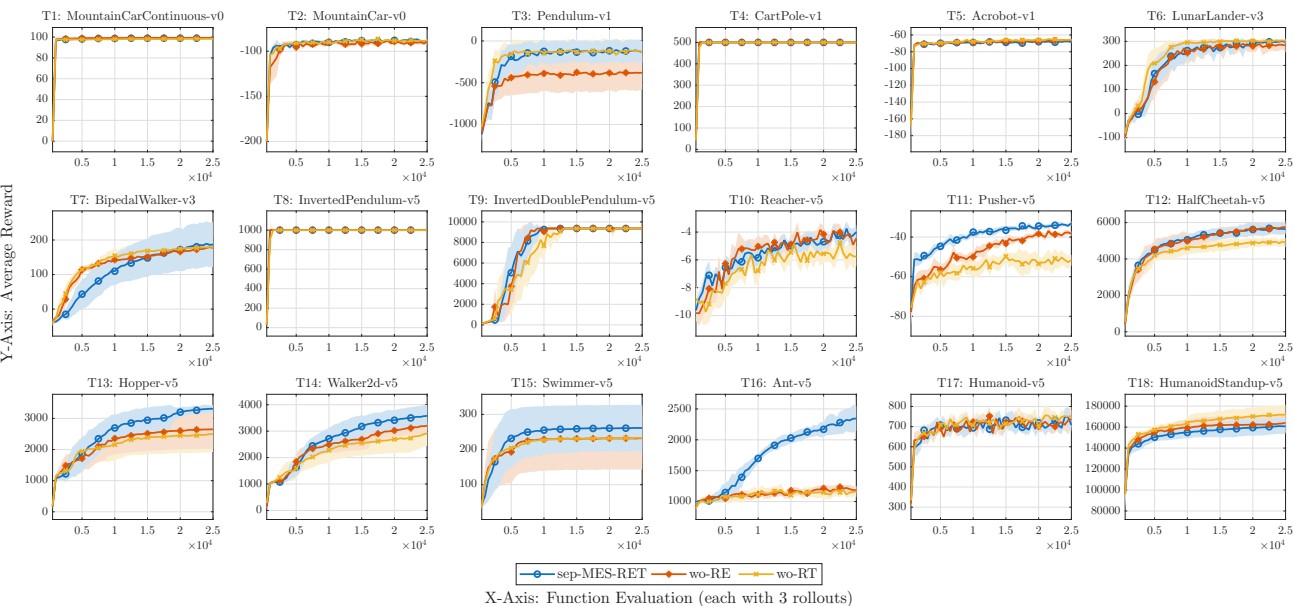

*Figure 10.* Convergence curves of average rewards obtained by the proposed MES-RET with ablation variants on the 18 reinforcement learning tasks. The x-axis denotes the number of evaluations (where 1 evaluation equals 3 rollouts), while the y-axis shows the average reward return over 3 rollouts. Solid lines represent the mean performance of the sampled policies per generation, and the shaded areas indicate the 95% confidence intervals computed from 10 independent runs.

## I. Detailed Results

Table 9 presents the comparative results on 87 synthetic benchmark optimization tasks. Table 10 presents the comparative results on 42 real-world constrained optimization tasks. Table 11 provides the average rewards across 18 policy search tasks.

## J. Additional Ablation Studies on Weak Guidance Injection

To explicitly isolate the effect of the Weak Guidance Injection mechanism, we introduced two additional variants and evaluated them on the benchmark suites:

- **w/o-TS** (Without Truncation Safeguard): Randomly replaces top-$\mu$ native elites with transferred samples, allowing them to influence the distribution update with positive weights regardless of their actual fitness.

- **w/-SI** (Strong Injection): Increases the injection size from $\tau = 1$ to $\tau = 10$, forcing a massive influx of external knowledge into the evaluation batch.

The average Friedman rankings for (MES-RET, w/o-RT, w/o-TS, w/-SI) are respectively 1.89, 2.01, 2.78, 3.32 on Synthetic tasks; 1.74, 1.89, 2.86, 3.51 on Real-World tasks; and 1.67, 2.33, 2.78, 3.22 on Policy Search. The corresponding Wilcoxon win/loss/tie metrics of the variants against MES-RET show overwhelming losses for w/o-TS and w/-SI. Unsurprisingly, absorbing unfiltered minimal knowledge (w/o-TS) or excessive external knowledge (w/-SI) inevitably injects severe negative transfer. This catastrophic performance drop explicitly validates that the minimal and truncation-guarded injection is a necessary mathematical firewall.

## K. Scalability to Extreme-Scale Many-Task Scenarios

To explicitly validate extreme scalability where categorical sampling probabilities might suffer from dilution, we designed a 1000-task ($K = 1000$) Planar Kinematic Arm Control Problem with four distinct heterogeneous modes (Easy, Normal, Unreachable, and Conflicting) and random dimensionalities ranging from 2 to 50.

Contrary to the theoretical expectation that categorical sampling might fail due to severe probability dilution at this scale, empirical results demonstrate that the base MES-RET remains remarkably effective. Even at $K = 1000$, the base framework

*Table 9.* Average objective values and standard deviations of the best solutions found by each algorithm on the 30 independent runs of the 87 synthetic unconstrained optimization tasks. The best results are highlighted in **bold**. The symbols +, −, and = indicate whether the algorithm is significantly better, worse, or equivalent to the proposed MES-RET according to the Wilcoxon rank-sum test with a significance level of 0.05. The last two rows show the number of best solutions found (#Best) and the overall Friedman rankings (Rank) of each algorithm.

| Task | MES-RET | w/o-RE | w/o-RT | CMA-ES | SBCMAES | MTES-KG | TNG-SNES | L-SHADE | AT-MFEA | EMaTO-MKT |
|---|---|---|---|---|---|---|---|---|---|---|
| T1 | **0.00E+00 ± 0.0E+00** | **0.00E+00 ± 0.0E+00** = | **0.00E+00 ± 0.0E+00** = | **0.00E+00 ± 0.0E+00** = | 9.28E+08 ± 7.6E+08 − | 2.17E+02 ± 1.0E+03 − | 1.01E+05 ± 2.9E+05 − | 1.10E−04 ± 2.1E−04 − | 1.35E+03 ± 1.7E+03 − | 3.77E+02 ± 4.6E+02 − |
| T2 | **0.00E+00 ± 0.0E+00** | **0.00E+00 ± 0.0E+00** = | **0.00E+00 ± 0.0E+00** = | **0.00E+00 ± 0.0E+00** = | 1.92E+04 ± 7.6E+03 − | **0.00E+00 ± 0.0E+00** = | 7.01E+02 ± 1.4E+03 − | **0.00E+00 ± 0.0E+00** = | 1.40E+02 ± 3.0E+02 − | 6.56E+02 ± 6.7E+02 − |
| T3 | **0.00E+00 ± 0.0E+00** | **0.00E+00 ± 0.0E+00** = | **0.00E+00 ± 0.0E+00** = | **0.00E+00 ± 0.0E+00** = | 1.96E+01 ± 2.2E+01 − | 8.36E−03 ± 1.3E−02 − | 3.75E+00 ± 1.0E+00 − | 1.85E−02 ± 3.5E−02 − | 6.75E+00 ± 4.8E−01 − | 7.03E+00 ± 4.6E−01 − |
| T4 | 2.12E+00 ± 1.4E+00 | 1.59E+00 ± 1.2E+00 = | 1.89E+00 ± 1.3E+00 = | 1.63E+00 ± 9.6E−01 = | 2.97E+01 ± 1.6E+01 − | **1.23E+00 ± 1.2E+00** + | 4.42E+00 ± 2.7E+00 − | 6.67E+00 ± 2.1E+00 − | 6.63E+00 ± 2.5E+00 − | 7.50E+00 ± 2.8E+00 − |
| T5 | **0.00E+00 ± 0.0E+00** | **0.00E+00 ± 0.0E+00** = | **0.00E+00 ± 0.0E+00** = | **0.00E+00 ± 0.0E+00** = | 5.96E+00 ± 9.0E+00 − | **0.00E+00 ± 0.0E+00** = | 1.46E−01 ± 5.2E−01 − | 9.47E−04 ± 7.9E−04 − | **0.00E+00 ± 0.0E+00** = | **0.00E+00 ± 0.0E+00** = |
| T6 | 1.18E+01 ± 8.1E−01 | 1.19E+01 ± 1.0E+00 = | **1.17E+01 ± 6.9E−01** = | 1.18E+01 ± 8.0E−01 = | 3.74E+01 ± 1.7E+01 − | 1.28E+01 ± 3.5E+00 − | 1.24E+01 ± 1.6E+00 − | 1.88E+01 ± 2.8E+00 − | 1.65E+01 ± 3.3E+00 − | 1.77E+01 ± 4.0E+00 − |
| T7 | 1.72E+00 ± 1.1E+00 | **1.29E+00 ± 1.1E+00** = | 1.89E+00 ± 1.2E+00 = | 1.79E+00 ± 1.1E+00 = | 3.28E+01 ± 1.6E+01 − | 1.49E+00 ± 1.2E+00 = | 6.35E+00 ± 4.6E+00 − | 7.95E+00 ± 2.1E+00 − | 5.88E+00 ± 2.5E+00 − | 8.02E+00 ± 3.5E+00 − |
| T8 | **0.00E+00 ± 0.0E+00** | **0.00E+00 ± 0.0E+00** = | **0.00E+00 ± 0.0E+00** = | **0.00E+00 ± 0.0E+00** = | 2.79E+01 ± 7.9E+01 − | **0.00E+00 ± 0.0E+00** = | 1.06E−01 ± 3.1E−01 − | **0.00E+00 ± 0.0E+00** = | 3.99E−03 ± 1.7E−02 − | 2.67E−05 ± 7.8E−05 − |
| T9 | 6.33E+01 ± 1.1E+02 | 3.27E+01 ± 5.6E−01 = | 7.56E+01 ± 1.2E+02 = | 9.22E+01 ± 1.4E+02 − | 1.41E+03 ± 1.8E+02 − | **7.67E+00 ± 2.2E+01** + | 8.69E+02 ± 5.1E+02 − | 3.23E+02 ± 1.1E+02 − | 2.92E+02 ± 2.0E+02 − | 3.24E+02 ± 1.7E+02 − |
| T10 | **3.65E−01 ± 5.5E−01** | 7.30E−01 ± 8.2E−01 = | 7.30E−01 ± 7.4E−01 = | 5.97E−01 ± 1.0E+00 = | 7.57E+02 ± 5.1E+02 − | 1.93E+00 ± 1.2E+01 − | 7.21E+00 ± 8.8E+00 − | 2.56E+00 ± 9.3E−01 − | 4.55E+00 ± 1.3E+00 − | 4.58E+00 ± 1.8E+00 − |
| T11 | 2.50E+02 ± 9.6E+01 | 2.37E+02 ± 1.7E+02 = | 3.04E+02 ± 1.9E+02 = | 2.59E+02 ± 1.3E+02 = | 5.72E+07 ± 7.1E+07 − | 1.75E+03 ± 1.1E+03 − | 6.38E+05 ± 8.8E+05 − | **1.58E+02 ± 8.8E+01** + | 3.62E+05 ± 1.0E+06 − | 3.68E+05 ± 7.0E+05 − |
| T12 | 4.70E+00 ± 3.0E+00 | 4.76E+00 ± 3.1E+00 = | 4.91E+00 ± 3.0E+00 = | **4.53E+00 ± 2.9E+00** = | 2.72E+05 ± 5.5E+05 − | 3.88E+02 ± 1.6E+02 − | 8.33E+03 ± 2.8E+03 − | 6.93E+00 ± 2.9E+00 − | 3.91E+03 ± 3.6E+03 − | 7.64E+03 ± 7.7E+03 − |
| T13 | 1.02E+01 ± 1.0E+01 | 1.16E+01 ± 9.8E+00 = | 1.25E+01 ± 1.0E+01 = | 1.16E+01 ± 1.2E+01 = | 9.94E+02 ± 1.2E+03 − | 4.97E+01 ± 1.4E+01 − | 4.20E+03 ± 3.1E+03 − | 2.53E+00 ± 6.7E−01 − | 1.82E+02 ± 2.4E+02 − | 6.67E+02 ± 6.8E+02 − |
| T14 | 8.16E−01 ± 9.4E−01 | 1.10E+00 ± 1.2E+00 = | **7.06E−01 ± 6.2E−01** = | 1.01E+00 ± 1.9E+00 = | 9.33E+02 ± 7.2E+03 − | 1.27E+02 ± 7.0E+01 − | 4.86E+03 ± 3.7E+03 − | 8.19E−01 ± 4.3E−01 − | 4.14E+02 ± 5.3E+02 − | 1.24E+03 ± 2.6E+03 − |
| T15 | 9.37E−01 ± 4.0E−01 | **8.41E−01 ± 2.5E−01** = | 4.81E+00 ± 2.3E+00 = | 2.32E+01 ± 4.5E+01 − | 1.92E+02 ± 4.7E+01 − | 1.49E+01 ± 3.6E+01 − | 2.21E+02 ± 1.7E+02 − | 2.53E+00 ± 6.7E−01 − | 6.50E+01 ± 4.9E+01 − | 5.17E+01 ± 6.9E+01 − |
| T16 | 2.28E+01 ± 1.2E+01 | 2.09E+01 ± 1.0E+01 = | 2.33E+01 ± 9.1E+00 = | 2.09E+01 ± 1.3E+01 = | 1.39E+02 ± 4.6E+01 − | 2.82E+01 ± 1.0E+01 − | 6.47E+01 ± 4.0E+01 − | 1.11E+01 ± 5.7E+00 − | **6.13E+00 ± 8.6E+00** + | 1.43E+01 ± 1.7E+01 + |
| T17 | 1.39E+01 ± 1.0E+01 | 1.22E+01 ± 1.0E+01 = | 1.17E+01 ± 1.0E+01 = | 1.63E+01 ± 8.5E+00 = | 4.14E+05 ± 7.0E+05 − | 3.53E+02 ± 3.4E+02 − | 4.37E+03 ± 2.5E+03 − | 6.52E+00 ± 7.7E+00 − | 7.82E+03 ± 6.8E+03 − | 1.12E+04 ± 1.2E+04 − |
| T18 | 2.01E+00 ± 1.1E+00 | 2.01E+00 ± 1.3E+00 = | 2.07E+00 ± 1.1E+00 = | 2.94E+00 ± 2.2E+00 = | 2.25E+02 ± 2.2E+02 − | 2.67E+02 ± 5.2E+03 − | 4.17E+03 ± 2.2E+03 − | **1.11E+00 ± 3.9E−01** − | 8.99E+02 ± 1.4E+03 − | 2.22E+03 ± 3.1E+03 − |
| T19 | 1.88E+01 ± 1.1E+02 | 1.57E+01 ± 1.1E+01 = | 2.23E+01 ± 2.1E+01 = | 1.98E+01 ± 1.0E+01 = | 2.19E+02 ± 5.8E+01 − | 2.23E+01 ± 6.3E+00 − | 1.42E+02 ± 7.7E+01 − | **6.38E−01 ± 4.3E−01** − | 2.69E+00 ± 4.8E+00 − | 4.47E+00 ± 5.6E+00 + |
| T20 | 2.00E+02 ± 1.2E+01 | 2.01E+02 ± 8.5E+00 = | 2.03E+02 ± 1.6E+00 = | 2.02E+02 ± 4.7E+00 = | 1.89E+02 ± 2.9E+01 + | 1.99E+02 ± 9.9E+00 = | 1.96E+02 ± 1.4E+01 = | 1.65E+02 ± 4.7E+01 = | **1.10E+02 ± 2.4E+01** + | 1.89E+02 ± 4.3E+01 + |
| T21 | **1.00E+02 ± 0.0E+00** | **1.00E+02 ± 0.0E+00** = | 1.00E+02 ± 5.3E−02 = | **1.00E+02 ± 0.0E+00** = | 1.15E+02 ± 2.0E+00 − | **1.00E+02 ± 0.0E+00** = | 1.00E+02 ± 8.1E−01 = | 1.00E+02 ± 1.6E−01 = | 1.00E+02 ± 4.0E−05 = | 1.00E+02 ± 4.1E−01 − |
| T22 | 3.03E+02 ± 1.8E+00 | 3.02E+02 ± 1.9E+00 = | 3.04E+02 ± 1.8E+00 = | 3.02E+02 ± 1.8E+00 = | 3.69E+02 ± 1.2E+01 − | **3.02E+02 ± 2.0E+00** = | 3.02E+02 ± 2.8E+00 = | 3.07E+02 ± 1.5E+00 − | 3.08E+02 ± 2.2E+00 − | 3.11E+02 ± 3.8E+00 − |
| T23 | 3.25E+02 ± 1.8E+01 | 3.28E+02 ± 5.6E+00 = | 3.28E+02 ± 8.2E+00 = | 3.28E+02 ± 6.3E+00 = | 3.83E+02 ± 1.9E+01 − | 3.26E+02 ± 1.2E+01 = | 2.70E+02 ± 9.8E+01 + | 3.17E+02 ± 5.3E+01 + | **2.59E+02 ± 1.8E+02** + | 3.32E+02 ± 4.1E+01 − |
| T24 | 4.43E+02 ± 2.2E−02 | 4.42E+02 ± 8.3E+00 = | 4.42E+02 ± 8.3E+00 = | 4.40E+02 ± 1.2E+01 = | 4.54E+02 ± 4.5E+01 − | 4.44E+02 ± 9.9E−01 − | 4.44E+02 ± 3.5E+00 − | **4.12E+02 ± 2.2E+01** + | 4.41E+02 ± 1.7E+01 + | 4.38E+02 ± 1.9E+01 + |
| T25 | 3.00E+02 ± 0.0E+00 | 3.00E+02 ± 0.0E+00 = | 3.00E+02 ± 0.0E+00 = | 3.00E+02 ± 0.0E+00 = | 7.80E+02 ± 4.2E+02 − | 3.56E+02 ± 2.3E+02 − | **2.83E+02 ± 3.8E−01** − | 3.00E+02 ± 0.0E+00 = | 3.11E+02 ± 2.1E+01 − | 3.12E+02 ± 2.5E+01 − |
| T26 | 3.94E+02 ± 2.2E+00 | 3.95E+02 ± 1.5E+00 = | 3.95E+02 ± 1.7E+00 = | 3.95E+02 ± 1.7E+00 = | 4.06E+02 ± 2.3E+00 − | 3.96E+02 ± 7.0E−01 − | 3.97E+02 ± 6.3E+00 − | **3.89E+02 ± 2.0E−01** + | 3.94E+02 ± 2.8E+00 = | 3.94E+02 ± 2.1E+00 = |
| T27 | 5.36E+02 ± 1.2E+02 | 5.52E+02 ± 1.0E+02 = | 4.70E+02 ± 1.4E+02 + | 6.24E+02 ± 3.9E+01 − | 6.04E+02 ± 1.9E+01 − | 5.52E+02 ± 9.6E+01 − | 3.09E+02 ± 5.2E+01 + | 4.33E+02 ± 1.1E+02 + | 5.67E+02 ± 1.0E+02 − |  |
| T28 | 2.40E+02 ± 8.1E+00 | 2.41E+02 ± 9.2E+00 = | **2.39E+02 ± 8.6E+00** = | 2.43E+02 ± 8.9E+00 = | 3.80E+02 ± 4.4E+01 − | 2.49E+02 ± 1.2E+01 − | 3.00E+02 ± 3.2E+01 − | 2.55E+02 ± 8.6E+00 − | 2.62E+02 ± 1.7E+01 − | 2.71E+02 ± 2.6E+01 − |
| T29 | 1.24E+05 ± 3.3E+05 | 5.49E+04 ± 2.1E+05 = | 1.40E+05 ± 3.3E+05 = | 8.21E+04 ± 2.5E+05 + | 8.41E+05 ± 1.6E+05 − | 3.02E+05 ± 4.4E+05 − | 9.97E+04 ± 2.7E+05 + | **2.81E+04 ± 1.5E+05** + | 6.39E+05 ± 5.6E+05 − | 5.03E+05 ± 5.3E+05 − |
| T30 | **0.00E+00 ± 0.0E+00** | **0.00E+00 ± 0.0E+00** = | **0.00E+00 ± 0.0E+00** = | **0.00E+00 ± 0.0E+00** = | 1.03E+09 ± 1.1E+09 − | 1.56E+01 ± 4.5E+01 − | 5.50E+03 ± 1.8E+04 − | 1.95E+02 ± 1.2E+02 − | 2.53E+03 ± 3.5E+03 − | 2.23E+03 ± 1.9E+03 − |
| T31 | **0.00E+00 ± 0.0E+00** | **0.00E+00 ± 0.0E+00** = | **0.00E+00 ± 0.0E+00** = | **0.00E+00 ± 0.0E+00** = | 1.68E+05 ± 2.1E+04 − | 1.61E−01 ± 2.7E−01 − | 8.01E+04 ± 9.5E+03 − | 2.13E+04 ± 2.9E+04 − | 1.09E+05 ± 1.6E+04 − | 9.36E+04 ± 2.2E+04 − |
| T32 | 8.33E+01 ± 1.1E+01 | 8.17E+01 ± 1.3E+01 = | 8.25E+01 ± 1.1E+01 = | **6.82E+01 ± 1.1E+01** + | 1.45E+03 ± 2.0E+03 − | 8.33E+02 ± 1.6E+02 − | 1.11E+02 ± 9.0E+00 − | 8.60E+01 ± 9.1E−01 − | 1.09E+02 ± 1.2E+01 − | 1.14E+02 ± 9.2E+00 − |
| T33 | 9.29E+00 ± 2.8E+00 | 1.01E+01 ± 2.6E+00 = | **8.95E+00 ± 2.9E+00** = | 9.45E+00 ± 2.2E+00 = | 7.09E+00 ± 6.7E+00 − | **8.95E+00 ± 2.6E+00** = | 2.18E+01 ± 1.3E+01 − | 7.74E+01 ± 1.1E+01 − | 3.24E+01 ± 1.2E+01 − | 3.33E+01 ± 1.0E+01 − |
| T34 | 4.63E−04 ± 5.2E−04 | **0.00E+00 ± 0.0E+00** = | **0.00E+00 ± 0.0E+00** = | **0.00E+00 ± 0.0E+00** = | 1.30E−03 ± 7.1E−03 − | 8.61E+00 ± 9.1E+00 − | 1.66E−03 ± 9.1E−03 − | 5.98E−03 ± 8.9E−03 − | 4.13E−02 ± 6.1E−02 − | 2.60E−04 ± 1.3E−04 − |
| T35 | 3.95E+01 ± 3.0E+00 | 3.86E+01 ± 2.2E+00 = | 3.82E+01 ± 1.8E+00 = | **3.78E+01 ± 2.0E+00** + | 8.32E+02 ± 6.8E+01 − | 3.81E+01 ± 2.3E+00 = | 3.98E+01 ± 2.3E+00 − | 1.06E+02 ± 1.2E+01 − | 7.06E+01 ± 1.7E+01 − | 6.24E+01 ± 7.6E+00 − |
| T36 | 1.03E+01 ± 2.4E+00 | 9.68E+00 ± 3.1E+00 = | 9.12E+00 ± 2.5E+00 + | 9.82E+00 ± 2.5E+00 = | 1.01E+02 ± 7.7E+01 − | **8.36E+00 ± 1.8E+00** + | 2.96E+01 ± 1.7E+01 − | 8.04E+01 ± 8.7E+00 − | 2.98E+01 ± 8.8E+00 − | 3.24E+01 ± 1.0E+01 − |
| T37 | **0.00E+00 ± 0.0E+00** | **0.00E+00 ± 0.0E+00** = | **0.00E+00 ± 0.0E+00** = | **0.00E+00 ± 0.0E+00** = | 6.47E+01 ± 2.0E+01 − | **0.00E+00 ± 0.0E+00** = | 3.33E−02 ± 1.2E−01 − | 5.97E−03 ± 2.3E−02 − | 2.56E−01 ± 5.3E−01 − | **0.00E+00 ± 0.0E+00** = |
| T38 | 7.14E+02 ± 3.0E+02 | 6.75E+02 ± 4.2E+02 = | **5.46E+02 ± 2.9E+02** = | 9.79E+02 ± 4.2E+02 − | 6.85E+03 ± 7.8E+02 − | 7.37E+02 ± 3.2E+02 − | 2.78E+03 ± 9.6E+02 − | 4.12E+03 ± 3.5E+02 − | 3.09E+03 ± 9.2E+02 − | 2.48E+03 ± 4.6E+02 − |
| T39 | 1.01E+02 ± 3.7E+01 | 9.73E+01 ± 3.6E+01 = | 9.22E+01 ± 3.7E+01 = | 9.57E+01 ± 4.1E+01 = | 1.37E+04 ± 5.9E+03 − | 1.07E+02 ± 4.0E+01 − | **4.60E+01 ± 3.9E+01** + | 5.48E+01 ± 3.7E+01 + | 1.84E+02 ± 2.0E+02 − | 1.37E+02 ± 1.2E+02 − |
| T40 | 1.66E+03 ± 1.9E+03 | 2.47E+03 ± 2.1E+03 = | **1.58E+03 ± 1.8E+03** = | 2.28E+03 ± 6.4E+02 − | 3.40E+08 ± 3.3E+08 − | 3.79E+04 ± 2.9E+04 − | 2.62E+05 ± 2.1E+05 − | 7.44E+03 ± 3.5E+03 − | 9.85E+05 ± 9.1E+05 − | 8.73E+05 ± 5.1E+05 − |
| T41 | **3.33E+01 ± 1.6E+01** | 1.35E+03 ± 4.1E+02 − | 3.98E+01 ± 2.6E+01 = | 1.26E+03 ± 3.6E+02 − | 5.27E+06 ± 9.8E+06 − | 1.05E+04 ± 4.2E+03 − | 2.69E+04 ± 9.6E+03 − | 2.31E+02 ± 7.3E+01 − | 9.77E+03 ± 9.2E+03 − | 1.20E+04 ± 7.7E+03 − |
| T42 | 1.42E+02 ± 2.9E+01 | 1.39E+02 ± 2.6E+01 = | 1.33E+02 ± 2.8E+01 = | 1.39E+02 ± 3.5E+01 = | 2.60E+06 ± 1.8E+06 − | 1.96E+02 ± 3.2E+01 − | 2.37E+05 ± 5.6E+05 − | **4.52E+01 ± 6.7E+00** + | 2.54E+05 ± 4.3E+05 − | 2.62E+05 ± 3.0E+05 − |
| T43 | 1.87E+02 ± 8.3E+01 | 2.95E+02 ± 9.8E+01 = | 1.90E+02 ± 9.1E+01 = | 2.61E+02 ± 7.4E+01 − | 2.65E+02 ± 3.9E+08 − | 3.75E+03 ± 2.2E+03 − | 1.27E+05 ± 3.2E+05 − | **6.97E+01 ± 2.7E+01** + | 3.21E+03 ± 3.2E+03 − | 4.43E+03 ± 7.1E+03 − |
| T44 | 2.29E+02 ± 1.5E+02 | **1.82E+02 ± 1.7E+02** = | 2.01E+02 ± 1.6E+02 = | 2.29E+02 ± 1.6E+02 = | 1.47E+03 ± 8.8E+02 − | 2.00E+02 ± 1.7E+02 = | 6.92E+02 ± 2.9E+02 − | 7.52E+02 ± 1.7E+02 − | 6.17E+02 ± 3.2E+02 − | 6.82E+02 ± 2.9E+02 − |
| T45 | **9.68E+01 ± 7.7E+01** | 1.07E+02 ± 1.0E+02 = | 1.06E+02 ± 9.4E+01 = | 1.09E+02 ± 8.4E+01 = | 4.24E+02 ± 5.1E+02 − | 1.03E+02 ± 7.5E+01 − | 3.45E+02 ± 2.6E+02 − | 1.69E+02 ± 4.4E+01 − | 2.32E+02 ± 1.3E+02 − | 2.19E+02 ± 1.7E+02 − |
| T46 | **1.29E+02 ± 5.4E+01** | 1.70E+02 ± 6.9E+01 − | 1.50E+02 ± 5.5E+01 − | 1.52E+02 ± 6.2E+01 − | 2.39E+07 ± 2.6E+07 − | 7.34E+03 ± 3.9E+03 − | 4.24E+05 ± 5.3E+05 − | 1.70E−06 ± 1.5E−06 − | 1.78E+06 ± 1.1E+06 − | 4.78E+06 ± 1.1E+06 − |
| T47 | 1.07E+02 ± 3.5E+01 | 1.06E+02 ± 3.1E+01 = | 1.01E+02 ± 3.4E+01 = | 1.00E+02 ± 2.6E+01 = | 2.64E+08 ± 3.0E+08 − | 4.55E+03 ± 3.4E+03 − | 2.71E+04 ± 4.9E+04 − | **2.82E+01 ± 7.1E+00** + | 1.19E+04 ± 1.2E+04 − | 4.98E+03 ± 6.5E+03 − |
| T48 | **1.29E+02 ± 8.4E+01** | 1.48E+02 ± 5.8E+01 = | 1.37E+02 ± 6.9E+01 = | 1.44E+02 ± 5.9E+01 = | 1.09E+04 ± 1.8E+04 − | 1.39E+02 ± 6.2E+01 = | 4.05E+02 ± 2.0E+02 − | 2.27E+02 ± 7.9E+01 − | 3.48E+02 ± 1.8E+02 − | 3.19E+02 ± 1.7E+02 − |
| T49 | 2.12E+02 ± 5.9E+00 | 2.12E+02 ± 3.7E+00 = | **2.11E+02 ± 4.7E+00** = | 2.14E+02 ± 4.9E+00 = | 3.03E+02 ± 7.3E+01 − | 2.12E+02 ± 4.1E+00 = | 2.15E+02 ± 6.9E+00 = | 2.76E+02 ± 1.0E+01 − | 2.36E+02 ± 1.2E+01 − | 2.36E+02 ± 1.3E+01 − |
| T50 | **1.00E+02 ± 0.0E+00** | **1.00E+02 ± 0.0E+00** = | **1.00E+02 ± 0.0E+00** = | **1.00E+02 ± 0.0E+00** = | 4.12E+03 ± 3.2E+03 − | **1.00E+02 ± 0.0E+00** = | 1.00E+02 ± 1.8E−05 = | 1.00E+02 ± 2.2E−04 = | 1.00E+02 ± 7.5E−03 = | 1.00E+02 ± 6.8E−04 = |
| T51 | 3.60E+02 ± 9.0E+00 | 3.58E+02 ± 8.5E+00 = | 3.59E+02 ± 7.5E+00 = | 3.61E+02 ± 6.2E+00 = | 7.50E+02 ± 1.1E+02 − | 3.61E+02 ± 6.1E+00 − | **3.57E+02 ± 1.9E+01** = | 4.27E+02 ± 8.1E+00 − | 3.85E+02 ± 1.1E+01 − | 3.87E+02 ± 1.1E+01 − |
| T52 | 4.25E+02 ± 4.8E+00 | **4.24E+02 ± 4.8E+00** = | 4.26E+02 ± 3.6E+00 = | 4.25E+02 ± 4.2E+00 = | 8.01E+02 ± 2.0E+01 − | 4.25E+02 ± 5.2E+00 = | 4.25E+02 ± 4.5E+00 = | 4.90E+02 ± 9.6E+00 − | 4.79E+02 ± 3.3E+01 − | 4.60E+02 ± 1.2E+01 − |
| T53 | 3.87E+02 ± 5.3E−01 | 3.87E+02 ± 3.2E−01 = | 3.87E+02 ± 4.3E−01 = | **3.87E+02 ± 3.2E−02** = | 3.98E+02 ± 1.3E+01 − | 3.87E+02 ± 6.5E−01 − | 3.87E+02 ± 6.8E−01 − | 3.87E+02 ± 4.0E−02 − | 3.90E+02 ± 2.5E+00 − | 3.89E+02 ± 1.4E+00 − |
| T54 | 9.24E+02 ± 7.7E+01 | 9.15E+02 ± 7.7E+01 = | 9.05E+02 ± 1.6E+02 = | 8.15E+02 ± 2.7E+02 = | 5.93E+03 ± 8.8E+02 − | 8.35E+02 ± 2.1E+02 − | **4.00E+02 ± 3.1E+02** + | 1.63E+03 ± 9.6E+01 − | 1.41E+03 ± 1.7E+02 − | 1.41E+03 ± 1.3E+02 − |
| T55 | 5.12E+02 ± 5.6E+00 | 5.13E+02 ± 9.1E+00 = | 5.13E+02 ± 9.1E+00 = | 5.19E+02 ± 7.6E+00 = | 6.77E+02 ± 1.3E+01 − | 5.19E+02 ± 1.1E+01 − | 5.16E+02 ± 8.2E+00 − | 5.16E+02 ± 4.5E+00 + | 5.14E+02 ± 1.0E+01 − | 5.12E+02 ± 6.5E+00 = |
| T56 | **3.49E+02 ± 5.8E+01** | 3.92E+02 ± 5.6E+01 = | 3.64E+02 ± 4.3E+01 = | 3.64E+02 ± 5.4E+01 = | 3.19E+03 ± 1.0E+03 − | 3.71E+02 ± 5.9E+01 = | 3.96E+02 ± 2.4E+01 − | 3.90E+02 ± 3.5E+01 − | 4.56E+02 ± 1.7E+01 − | 4.43E+02 ± 2.0E+01 − |
| T57 | 4.77E+02 ± 4.3E+01 | 4.92E+02 ± 6.3E+01 = | 4.79E+02 ± 4.0E+01 = | **4.74E+02 ± 3.2E+01** = | 6.37E+02 ± 2.5E+02 − | 5.14E+02 ± 5.7E+01 − | 5.28E+02 ± 1.2E+01 − | 6.02E+02 ± 3.5E+01 − | 6.29E+02 ± 1.8E+01 − | 5.47E+02 ± 1.1E+01 − |
| T58 | **2.16E+03 ± 1.3E+02** | 2.82E+03 ± 3.7E+02 = | 2.49E+03 ± 1.0E+03 = | 2.85E+03 ± 2.5E+02 = | 6.37E+06 ± 2.7E+07 − | 2.59E+04 ± 9.4E+03 − | 2.05E+05 ± 8.4E+05 − | 2.49E+03 ± 2.7E+02 − | 6.40E+03 ± 2.9E+03 − | 5.17E+03 ± 2.0E+03 − |
| T59 | 1.00E+02 ± 3.9E+02 | 1.15E+02 ± 1.4E+02 = | 2.49E+02 ± 4.8E+02 = | 1.78E+02 ± 2.0E+02 − | 1.48E+10 ± 1.1E+10 − | **2.76E−01 ± 6.4E−01** − | 1.54E+03 ± 1.4E+03 − | 2.66E+03 ± 1.4E+03 − | 2.40E+05 ± 1.2E+05 − | 3.04E+03 ± 7.0E+03 − |
| T60 | **0.00E+00 ± 0.0E+00** | 5.55E+03 ± 3.8E+03 − | **0.00E+00 ± 0.0E+00** = | 4.62E+03 ± 2.7E+03 − | 2.50E+05 ± 3.9E+04 − | 1.46E+03 ± 1.0E+03 − | 1.74E+05 ± 1.8E+04 − | 6.43E+04 ± 7.5E+04 − | 2.55E+05 ± 2.4E+04 − | 2.64E+05 ± 4.4E+04 − |
| T61 | 7.57E+01 ± 1.3E+01 | **6.82E+01 ± 4.9E+01** = | 7.11E+01 ± 9.1E+00 = | 1.24E+02 ± 4.6E+01 − | 9.29E+03 ± 8.1E+03 − | 1.04E+02 ± 5.6E+01 − | 1.66E+02 ± 3.8E+01 − | 1.59E+02 ± 3.1E+01 − | 1.89E+02 ± 3.4E+01 − | 1.73E+02 ± 3.3E+01 − |
| T62 | 2.18E+01 ± 4.1E+00 | 2.02E+01 ± 3.4E+00 = | 1.95E+01 ± 4.0E+00 = | **1.94E+01 ± 4.2E+00** = | 6.77E+01 ± 5.6E+01 − | 1.97E+01 ± 3.9E+00 = | 3.78E+01 ± 1.7E+01 − | 1.60E+02 ± 1.3E+01 − | 1.16E+02 ± 6.3E+01 − | 7.13E+01 ± 1.3E+01 − |
| T63 | 2.98E−03 ± 7.4E−03 | **1.05E−03 ± 3.2E−03** = | 1.62E−03 ± 5.9E−03 = | 3.83E−03 ± 8.3E−03 − | 1.98E+01 ± 1.8E+01 − | 3.82E−03 ± 5.7E−03 − | 1.82E−03 ± 4.1E−03 − | 9.32E−03 ± 4.8E−03 − | 6.46E−02 ± 1.1E−02 − | 6.89E−03 ± 2.7E−03 − |
| T64 | 6.86E+01 ± 2.8E+00 | 6.89E+01 ± 3.7E+00 = | 6.73E+01 ± 3.7E+00 = | **6.63E+01 ± 2.8E+00** + | 8.73E+01 ± 7.3E+01 − | 6.67E+01 ± 3.7E+00 + | 7.13E+01 ± 5.0E+00 − | 2.14E+02 ± 1.3E+01 − | 3.16E+02 ± 8.3E+01 − | 1.32E+02 ± 2.2E+01 − |
| T65 | 1.97E+00 ± 7.4E+00 | 2.02E+01 ± 4.5E+00 = | **1.82E+01 ± 4.7E+00** = | 2.15E+01 ± 1.7E+00 − | 1.06E+02 ± 7.3E+01 − | 2.15E+01 ± 5.6E+00 − | 3.66E+01 ± 1.8E+01 − | 1.65E+02 ± 1.1E+01 − | 1.19E+02 ± 5.5E+01 − | 7.15E+01 ± 1.7E+01 − |
| T66 | **0.00E+00 ± 0.0E+00** | **0.00E+00 ± 0.0E+00** = | **0.00E+00 ± 0.0E+00** = | **0.00E+00 ± 0.0E+00** = | 1.04E+03 ± 1.5E+03 − | 2.98E−03 ± 1.6E−02 − | 3.03E−02 ± 1.2E−01 − | 8.08E−02 ± 1.3E−01 − | 3.48E+00 ± 2.6E+00 − | 1.92E−01 ± 3.7E−01 − |
| T67 | 1.32E+03 ± 4.8E+02 | 1.40E+03 ± 4.6E+02 = | 1.15E+03 ± 5.3E+02 = | **1.14E+03 ± 5.1E+02** = | 1.27E+04 ± 2.4E+03 − | 1.54E+03 ± 4.5E+02 − | 4.94E+03 ± 1.8E+03 − | 8.14E+03 ± 3.7E+02 − | 8.32E+03 ± 2.8E+03 − | 4.59E+03 ± 9.1E+02 − |
| T68 | 1.78E+02 ± 4.0E+01 | 1.74E+02 ± 3.7E+01 = | 1.89E+02 ± 4.1E+01 = | 1.86E+02 ± 3.9E+01 = | 3.53E+04 ± 1.1E+04 − | **1.69E+02 ± 4.3E+01** = | 1.42E+03 ± 1.3E+03 − | 2.44E+02 ± 2.7E+02 − | 7.55E+02 ± 5.3E+02 − | 1.03E+03 ± 8.2E+02 − |
| T69 | **1.24E+04 ± 1.0E+04** | 1.03E+06 ± 4.6E+05 − | 1.28E+04 ± 7.4E+03 = | 7.81E+05 ± 3.3E+05 − | 1.31E+10 ± 1.1E+10 − | 7.54E+05 ± 3.4E+05 − | 1.41E+06 ± 4.2E+05 − | 1.29E+05 ± 6.3E+04 − | 7.35E+06 ± 4.1E+06 − | 3.01E+06 ± 1.9E+06 − |
| T70 | 1.19E+04 ± 1.4E+04 | 3.01E+04 ± 1.4E+04 − | 6.10E+03 ± 6.6E+03 − | 3.50E+04 ± 1.2E+04 − | 6.03E+09 ± 5.7E+09 − | 1.35E+04 ± 4.8E+03 − | 2.45E+04 ± 2.5E+04 − | 2.21E+03 ± 5.7E+02 + | 3.37E+03 ± 2.7E+03 + | **1.74E+03 ± 2.3E+03** + |
| T71 | 2.33E+02 ± 5.4E+01 | 1.16E+03 ± 5.4E+02 − | 2.53E+02 ± 4.8E+01 = | 8.94E+02 ± 4.0E+02 − | 1.04E+07 ± 1.1E+07 − | 7.98E+02 ± 3.6E+02 − | 1.68E+05 ± 1.2E+05 − | **1.20E+02 ± 2.4E+01** + | 8.38E+05 ± 6.2E+05 − | 6.75E+05 ± 4.5E+05 − |
| T72 | 5.78E+03 ± 7.7E+03 | 2.18E+04 ± 1.1E+04 − | 2.86E+03 ± 3.6E+03 − | 1.81E+04 ± 7.0E+03 − | 5.90E+08 ± 6.2E+08 − | 3.21E+03 ± 1.2E+03 − | 1.29E+05 ± 3.1E+05 − | **3.99E+02 ± 9.0E+01** + | 2.11E+03 ± 3.3E+03 + | 4.58E+03 ± 3.3E+03 + |
| T73 | **4.12E+02 ± 1.4E+02** | 4.51E+02 ± 1.8E+02 = | 4.88E+02 ± 1.7E+02 = | 5.60E+02 ± 1.9E+02 − | 1.51E+03 ± 9.1E+02 − | 5.61E+02 ± 2.5E+02 − | 8.37E+02 ± 3.3E+02 − | 1.77E+03 ± 2.6E+02 − | 1.38E+03 ± 4.7E+02 − | 1.61E+03 ± 4.5E+02 − |
| T74 | 4.83E+02 ± 1.5E+02 | 5.40E+02 ± 1.7E+02 = | **4.37E+02 ± 1.6E+02** = | 4.88E+02 ± 1.5E+02 = | 6.09E+02 ± 1.9E+02 − | 6.09E+02 ± 1.9E+02 − | 1.08E+03 ± 4.3E+02 − | 1.24E+03 ± 1.7E+02 − | 1.18E+03 ± 2.9E+02 − | 1.43E+03 ± 4.1E+02 − |
| T75 | 2.74E+02 ± 1.0E+02 | 3.69E+04 ± 1.5E+04 − | **2.72E+02 ± 7.3E+01** = | 3.62E+04 ± 1.4E+04 − | 6.71E+07 ± 6.8E+07 − | 2.48E+04 ± 1.1E+04 − | 1.53E+06 ± 8.6E+05 − | 4.20E+02 ± 1.4E+02 − | 4.18E+06 ± 3.6E+06 − | 4.67E+06 ± 5.0E+06 − |
| T76 | 1.13E+02 ± 3.6E+01 | 1.98E+04 ± 7.3E+03 − | 1.19E+02 ± 3.1E+01 = | 1.96E+04 ± 9.7E+03 − | 1.65E+08 ± 2.2E+08 − | 1.69E+04 ± 6.2E+03 − | 9.68E+04 ± 1.0E+05 − | **8.71E+01 ± 2.3E+01** + | 1.09E+04 ± 6.3E+03 − | 6.24E+04 ± 6.4E+03 − |
| T77 | **1.34E+02 ± 4.0E+01** | 1.50E+02 ± 6.8E+01 = | 1.42E+02 ± 5.4E+01 = | 1.38E+02 ± 3.1E+01 = | 2.32E+03 ± 3.0E+02 − | 1.45E+02 ± 4.9E+01 − | 5.00E+02 ± 2.2E+02 − | 1.03E+03 ± 1.9E+02 − | 1.05E+03 ± 3.4E+02 − | 8.17E+02 ± 2.7E+02 − |
| T78 | **2.22E+02 ± 5.2E+00** | 2.23E+02 ± 4.1E+00 = | 2.23E+02 ± 3.2E+00 = | 2.23E+02 ± 4.2E+00 = | 2.24E+02 ± 6.4E+00 = | 2.24E+02 ± 6.4E+00 = | 2.29E+02 ± 7.5E+00 − | 3.07E+02 ± 6.5E+00 − | 2.79E+02 ± 1.9E+01 − | 2.26E+02 ± 1.9E+01 − |
| T79 | 1.44E+02 ± 2.4E+02 | **1.00E+02 ± 0.0E+00** + | 5.59E+02 ± 9.0E+02 = | 2.15E+02 ± 4.7E+02 = | 1.09E+06 ± 3.3E+03 − | 3.46E+02 ± 7.5E+02 − | 1.81E+03 ± 3.0E+03 − | 5.91E+03 ± 3.0E+03 − | 7.68E+02 ± 2.0E+03 − | 4.87E+03 ± 1.8E+03 − |
| T80 | 4.40E+02 ± 9.7E+00 | 4.42E+02 ± 9.0E+00 = | **4.40E+02 ± 7.2E+00** = | 4.46E+02 ± 1.3E+01 − | 1.25E+03 ± 3.5E+01 − | 4.48E+02 ± 1.1E+01 − | 4.47E+02 ± 1.3E+01 − | 6.57E+02 ± 1.3E+01 − | 5.15E+02 ± 2.4E+01 − | 4.99E+02 ± 2.2E+01 − |
| T81 | 5.06E+02 ± 1.0E+01 | 5.05E+02 ± 9.4E+00 = | 5.10E+02 ± 6.2E+00 = | **5.04E+02 ± 8.7E+00** = | 1.45E+03 ± 3.3E+01 − | 5.14E+02 ± 1.1E+01 − | 5.08E+02 ± 1.1E+01 − | 6.57E+02 ± 1.1E+01 − | 8.13E+02 ± 1.0E+02 − | 5.73E+02 ± 1.9E+01 − |
| T82 | **4.83E+02 ± 4.8E+00** | 4.92E+02 ± 1.3E+01 = | 4.86E+02 ± 1.7E+01 = | 4.88E+02 ± 2.6E+01 = | 8.61E+02 ± 6.9E+01 − | 5.04E+02 ± 3.2E+01 − | 4.85E+02 ± 5.5E+00 = | 5.68E+02 ± 1.8E+01 − | 5.76E+02 ± 1.7E+01 − |  |
| T83 | 1.16E+03 ± 1.0E+02 | 1.20E+03 ± 9.6E+01 = | 1.15E+03 ± 1.1E+02 = | 1.16E+03 ± 9.1E+01 = | 1.18E+04 ± 3.1E+03 − | 1.20E+03 ± 8.9E+01 − | **6.94E+02 ± 4.6E+02** + | 2.18E+03 ± 5.4E+02 − | 1.83E+03 ± 1.8E+02 − |  |
| T84 | 5.51E+02 ± 1.8E+01 | 5.58E+02 ± 2.1E+01 = | 5.49E+02 ± 2.5E+01 = | 5.71E+02 ± 2.3E+01 − | 1.08E+03 ± 4.5E+01 − | 5.72E+02 ± 3.9E+01 − | 6.77E+02 ± 1.0E+01 − | **5.40E+02 ± 1.2E+01** + | 6.26E+02 ± 3.7E+01 − | 6.12E+02 ± 3.3E+01 − |
| T85 | 4.86E+02 ± 2.4E+01 | 4.87E+02 ± 2.3E+01 = | 4.91E+02 ± 2.3E+01 = | 4.92E+02 ± 2.3E+01 = | 6.20E+03 ± 1.1E+03 − | 5.01E+02 ± 1.7E+01 − | **4.69E+02 ± 2.6E+01** + | 4.81E+02 ± 2.3E+01 − | 5.75E+02 ± 3.6E+01 − | 5.67E+02 ± 2.6E+01 − |
| T86 | 6.56E+02 ± 1.8E+02 | 6.78E+02 ± 1.7E+02 = | 6.89E+02 ± 1.8E+02 = | 6.89E+02 ± 1.8E+02 − | 4.29E+03 ± 1.1E+03 − | 7.37E+02 ± 1.8E+02 − | **5.79E+02 ± 2.2E+02** + | 7.57E+02 ± 1.1E+02 − | 7.57E+02 ± 2.7E+02 − | 6.20E+02 ± 2.9E+02 − |
| T87 | **6.57E+05 ± 7.4E+04** | 3.03E+06 ± 5.8E+05 − | 6.78E+05 ± 7.9E+04 = | 3.48E+06 ± 7.4E+05 − | 1.22E+08 ± 1.8E+08 − | 3.11E+06 ± 6.7E+05 − | 1.15E+06 ± 5.4E+05 − | 7.36E+05 ± 1.3E+05 − | 9.08E+05 ± 1.1E+05 − | 8.87E+05 ± 1.5E+05 − |
| +/−/= | Ours | 1/17/69 | 1/7/79 | 6/21/60 | 1/83/3 | 4/43/40 | 7/63/17 | 20/57/10 | 8/72/7 | 5/73/9 |
| #Best ↑ | **25** | 19 | 23 | 20 | 0 | 13 | 7 | 19 | 4 | 4 |
| Rank ↓ | **3.13** | 3.82 | 3.28 | 3.90 | 9.68 | 4.97 | 6.70 | 5.33 | 7.02 | 7.17 |

*Table 10.* Average objective values and standard deviations of the best solutions found by each algorithm on the 30 independent runs of the 42 real-world constrained optimization tasks. "NaN" indicates that the algorithm failed to find a feasible solution. The best results are highlighted in **bold**. The symbols +, −, and = indicate whether the algorithm is significantly better, worse, or equivalent to the proposed MES-RET according to the Wilcoxon rank-sum test with a significance level of 0.05, where "NaN" is replaced by a large number 1.00E+8 for the comparison. The last two rows show the number of best solutions found (#Best) and the Friedman rankings (Rank) of each algorithm.

| Task | MES-RET | w/o-RE | w/o-RT | CMA-ES | SBCMAES | MTES-KG | TNG-SNES | CCEF-ECHT | AT-MFEA | CEDA-MP |
|---|---|---|---|---|---|---|---|---|---|---|
| T1 | **2.0887E+02 ± 2.9E-01** | 2.7149E+02 ± 9.8E+01 = | 4.2772E+02 ± 6.2E+01 = | NaN = | NaN = | NaN − | 3.9275E+02 ± 2.7E+01 = | NaN − | 4.9305E+02 ± 1.2E+02 = | 3.1161E+02 ± 8.8E+01 = |
| T2 | **7.0816E+03 ± 1.7E+02** | 7.2231E+03 ± 3.3E+02 = | 1.2184E+04 ± 2.9E+03 = | 1.2184E+04 ± 2.9E+03 = | NaN = | NaN − | 1.0505E+04 ± 3.2E+03 = | NaN = | NaN = | NaN = |
| T3 | -1.2812E+02 ± 5.6E-01 | -1.1458E+02 ± 6.3E+00 = | -8.6205E+00 ± 1.4E+02 − | -8.6205E+00 ± 1.4E+02 − | NaN = | -7.7392E+02 ± 1.6E+03 = | -2.5077E+03 ± 2.5E+02 = | **-4.5291E+03 ± 9.3E-13 +** | -2.2337E+02 ± 1.7E+03 + | -2.0910E+02 ± 2.4E+03 + |
| T4 | -3.7843E-01 ± 6.0E-03 | -3.7490E-01 ± 1.7E-16 | **-3.8370E-01 ± 6.4E-03** | -3.8370E-01 ± 6.4E-03 | -0.0000E+00 ± 0.0E+00 = | -3.0210E-01 ± 2.7E-02 − | -2.0711E-01 ± 1.2E-01 − | -2.7632E-01 ± 1.0E-01 − | -3.0378E-01 ± 3.2E-02 − | -3.1006E-01 ± 7.3E-02 − |
| T5 | **-0.0000E+00 ± 0.0E+00** | -8.6667E-05 ± 2.8E-04 | 1.3169E+01 ± 3.4E-01 | -4.3382E-01 ± 1.1E-02 + | -0.0000E+00 ± 0.0E+00 = | 4.2924E+01 ± 4.7E-01 | 5.4936E+01 ± 3.1E+01 | -2.4103E-02 ± 1.1E-01 | -2.5114E+00 ± 3.7E+01 | 1.3991E+01 ± 4.7E+01 |
| T6 | 2.0014E+00 ± 5.5E-03 | 2.0004E+00 ± 9.3E-04 | 2.0004E+00 ± 5.5E-03 | 2.0043E+00 ± 4.5E-03 | 2.0012E+00 ± 1.7E-03 + | 2.0056E+00 ± 4.9E-03 − | 2.0062E+00 ± 6.9E-03 − | **2.0000E+00 ± 0.0E+00 =** | **2.0000E+00 ± 0.0E+00 =** | **2.0000E+00 ± 0.0E+00 +** |
| T7 | **2.5577E+00 ± 1.8E-15** | **2.5577E+00 ± 1.8E-15** | **2.5577E+00 ± 1.8E-15** | **2.5577E+00 ± 1.8E-15** | 3.4231E+00 ± 4.6E-01 | 2.5666E+00 ± 1.8E-02 | 2.7168E+00 ± 6.4E-02 | **2.5577E+00 ± 1.8E-15** | 2.6175E+00 ± 5.6E-02 | 2.6096E+00 ± 5.9E-02 |
| T8 | **1.0765E+00 ± 4.5E-16** | 1.0785E+00 ± 1.1E-02 | 1.1343E+00 ± 8.3E-02 | 1.1283E+00 ± 8.1E-02 | 1.2194E+00 ± 4.6E-02 | 1.1864E+00 ± 8.5E-02 | 1.1905E+00 ± 8.2E-02 | **1.0765E+00 ± 4.5E-16** | 1.0805E+00 ± 3.9E-03 | 1.0998E+00 ± 6.0E-02 |
| T9 | **1.0030E+02 ± 2.8E+00** | 1.0036E+02 ± 2.9E+00 | 1.0199E+02 ± 6.5E+00 | 1.0208E+02 ± 3.6E+00 | NaN = | 1.2425E+02 ± 9.1E+00 | 1.0081E+02 ± 3.8E+00 | 1.1245E+02 ± 6.7E+00 | 1.1455E+02 ± 9.8E+00 |
| T10 | 3.7134E-01 ± 7.2E-01 | 4.8332E+00 ± 8.1E-01 | 3.6587E+00 ± 5.0E-01 | 3.0184E+00 ± 2.6E-01 | 7.6013E+00 ± 9.9E-01 | 4.1833E+00 ± 5.2E-01 | 3.4962E+00 ± 5.9E-01 | **2.9303E+00 ± 9.1E-03** | 2.9938E+00 ± 7.0E-02 | 2.9853E+00 ± 6.4E-02 |
| T11 | **2.6887E+04 ± 1.9E-11** | **2.6887E+04 ± 1.9E-11** | **2.6887E+04 ± 1.9E-11** | **2.6887E+04 ± 1.9E-11** | 2.6890E+04 ± 2.3E+00 | 2.6887E+04 ± 2.6E-03 | **2.6887E+04 ± 1.9E-11** | **2.6887E+04 ± 1.9E-11** | 2.6901E+04 ± 7.3E+01 | **2.6887E+04 ± 1.9E-11** |
| T12 | 5.7601E+04 ± 2.1E+03 | 5.7086E+04 ± 2.2E+03 | 5.8503E+04 ± 2.5E-05 | 5.8503E+04 ± 4.4E-11 | 8.2921E+04 ± 1.2E+04 | 8.2921E+04 ± 4.1E+01 | 5.8517E+04 ± 2.3E+03 | 5.8517E+04 ± 2.3E+03 | 6.0207E+04 ± 7.7E+02 | 5.9123E+04 ± 2.3E+03 |
| T13 | **2.9944E+03 ± 1.4E-12** | **2.9944E+03 ± 1.4E-12** | **2.9944E+03 ± 1.4E-12** | **2.9944E+03 ± 1.4E-12** | 3.0653E+03 ± 1.1E-01 | 2.9960E+03 ± 1.6E+00 | **2.9944E+03 ± 1.4E-12** | 2.9944E+03 ± 1.8E-05 | **2.9944E+03 ± 1.4E-12** | 2.9944E+03 ± 4.3E-13 |
| T14 | 3.3040E-02 ± 3.2E-03 | 3.2617E-02 ± 2.3E-03 | 3.4450E-02 ± 3.8E-03 | 3.4450E-02 ± 3.8E-03 | 9.1515E+06 ± 4.1E+03 | 1.3943E+00 ± 2.2E+00 | 5.7290E-02 ± 6.1E-13 | 3.4055E-02 ± 8.7E-04 | 1.0484E-01 ± 3.3E-02 | 1.3828E-01 ± 5.8E-02 |
| T15 | **1.2700E-02 ± 7.1E-18** | **1.2700E-02 ± 7.1E-18** | **1.2700E-02 ± 7.1E-18** | **1.2700E-02 ± 7.1E-18** | 1.3150E-02 ± 1.2E-04 | 1.3150E-02 ± 1.2E-04 | **1.2700E-02 ± 7.1E-18** | **1.2700E-02 ± 7.1E-18** | 1.3810E-02 ± 9.8E-04 | 1.3353E-02 ± 8.2E-04 |
| T16 | 6.0883E+03 ± 1.2E+02 | **6.0597E+03 ± 4.6E-12** | 6.0756E+03 ± 6.2E+01 | 6.0618E+03 ± 2.6E+03 | 6.2557E+03 ± 1.5E+02 | 6.5499E+03 ± 1.8E+02 | **6.0597E+03 ± 4.6E-12 +** | 6.4556E+03 ± 2.6E+02 | 6.6184E+03 ± 4.1E+02 |
| T17 | **1.6702E+00 ± 9.0E-16** | **1.6702E+00 ± 9.0E-16** | **1.6702E+00 ± 9.0E-16** | **1.6702E+00 ± 9.0E-16** | 2.0687E+00 ± 9.0E-16 | 1.7419E+00 ± 6.8E-02 | 2.0044E+00 ± 1.7E-01 | **1.6702E+00 ± 9.0E-16** | 1.7918E+00 ± 1.2E-01 | 1.7964E+00 ± 1.6E-01 |
| T18 | 2.6690E+02 ± 1.3E-04 | 2.6690E+02 ± 2.2E-04 | 2.6390E+02 ± 1.7E-13 | **2.6390E+02 ± 1.7E-13** | 2.6652E+02 ± 6.8E-01 | 2.6789E+02 ± 3.6E+00 | 2.6868E+02 ± 3.1E+00 | **2.6390E+02 ± 1.7E-13** | **2.6390E+02 ± 1.1E-02** | 2.6393E+02 ± 3.8E-02 |
| T19 | **2.3520E-01 ± 8.5E-17** | **2.3520E-01 ± 8.5E-17** | **2.3520E-01 ± 8.5E-17** | **2.3520E-01 ± 8.5E-17** | 2.3531E-01 ± 4.5E-05 | **2.3520E-01 ± 8.5E-17** | 2.3982E-01 ± 3.2E-03 | **2.3520E-01 ± 8.5E-17** | 2.3560E-01 ± 1.0E-03 | 2.3521E-01 ± 7.3E-05 |
| T20 | 5.2800E-01 ± 2.0E-03 | 5.2900E-01 ± 3.9E-03 | 5.2982E-01 ± 4.8E-03 | 5.3339E-01 ± 5.3E-01 | 1.4528E+00 ± 5.3E+00 | 5.3145E-01 ± 6.9E-03 | 8.8034E-01 ± 4.6E-01 | 5.8070E-01 ± 9.1E-02 | 6.2642E-01 ± 1.7E-01 | 5.9020E-01 ± 2.3E-01 |
| T21 | 1.6349E+01 ± 4.7E-01 | 1.6175E+01 ± 3.2E-01 | 1.6558E+01 ± 5.3E-01 | **1.6070E+01 ± 7.2E-15** | 1.7268E+01 ± 6.5E-01 | 1.6843E+01 ± 6.5E-01 | 5.1930E+00 ± 3.2E-01 | **1.6070E+01 ± 4.1E-05** | 1.6752E+01 ± 2.8E-01 | 1.6802E+01 ± 3.3E-01 |
| T22 | **2.5438E+00 ± 4.5E-16** | **2.5438E+00 ± 4.5E-16** | **2.5438E+00 ± 4.5E-16** | 2.5438E+00 ± 4.5E-16 | 6.194E+00 ± 1.5E+00 | 3.083E+01 ± 8.7E-02 | 5.1930E+00 ± 3.2E-01 | 2.5448E+00 ± 9.3E-03 | 4.4577E+00 ± 7.8E-01 | 3.8561E+00 ± 7.5E-01 |
| T23 | 2.2526E+03 ± 8.2E+02 | 2.6041E+03 ± 7.3E+02 | 3.2148E+03 ± 1.6E+03 | 3.4630E+03 ± 1.5E+03 | 6.8032E+03 ± 2.2E+03 | 2.7423E+03 ± 6.6E+02 | 5.7917E+03 ± 2.1E+03 | 3.0071E+03 ± 4.4E+02 | 2.8549E+01 ± 4.9E+02 |
| T24 | 4.2985E+01 ± 1.6E+01 | **4.0397E+01 ± 5.3E+00** | 7.3458E+01 ± 2.7E+01 | NaN = | 9.5738E+01 ± 3.3E+01 | NaN = | 5.1156E+01 ± 6.8E+00 | 5.6810E+01 ± 1.3E+01 | 4.8874E+01 ± 9.2E+00 |
| T25 | 5.2465E+02 ± 1.1E+00 | **5.2445E+02 ± 2.3E-13** | **5.2445E+02 ± 2.3E-13 +** | 5.2506E+02 ± 1.9E+00 + | 5.2448E+02 ± 1.8E-02 + | 5.2446E+02 ± 1.8E-02 | 5.2637E+00 ± 2.8E+00 | 5.2450E+02 ± 1.7E-02 | 5.2709E+02 ± 2.8E+00 | 5.3070E+02 ± 4.0E+00 |
| T26 | **1.6958E+04 ± 0.0E+00** | **1.6958E+04 ± 0.0E+00** | **1.6958E+04 ± 0.0E+00** | **1.6958E+04 ± 0.0E+00** | 1.7154E+04 ± 1.6E+02 | 1.6964E+04 ± 5.3E+00 | **1.6958E+04 ± 0.0E+00** | 1.6958E+04 ± 1.4E+00 | 1.6958E+04 ± 2.8E+00 | 1.6958E+04 ± 2.8E-02 |
| T27 | 2.9649E+06 ± 1.3E-04 | 2.9649E+06 ± 4.8E-05 | 2.9649E+06 ± 8.8E-05 | **2.9649E+06 ± 4.7E-10** | 3.5192E+06 ± 1.0E+05 | 2.9806E+06 ± 1.6E+04 | 2.9674E+06 ± 2.8E+03 | **2.9649E+06 ± 4.7E-10** | 2.9981E+06 ± 4.8E+04 | 2.9765E+06 ± 1.6E+04 |
| T28 | 3.0971E+02 ± 3.5E-01 | 2.9244E+02 ± 4.2E-01 | 2.6773E+02 ± 1.9E-02 | 2.6587E+02 ± 1.2E+00 | 4.6568E+00 ± 1.2E+00 | 2.6764E+02 ± 2.0E-02 | 3.3444E+00 ± 3.7E-01 | **2.6586E+00 ± 1.4E-15** | 2.7886E+00 ± 9.9E-02 | 2.7279E+00 ± 1.0E-01 |
| T29 | **0.0000E+00 ± 0.0E+00** | **0.0000E+00 ± 0.0E+00** | **0.0000E+00 ± 0.0E+00** | **0.0000E+00 ± 0.0E+00** | **0.0000E+00 ± 0.0E+00** | **0.0000E+00 ± 0.0E+00** | **0.0000E+00 ± 0.0E+00** | **0.0000E+00 ± 0.0E+00** | **0.0000E+00 ± 0.0E+00** | **0.0000E+00 ± 0.0E+00** |
| T30 | **-3.0666E+04 ± 0.0E+00** | -3.0666E+04 ± 1.8E-05 | -3.0666E+04 ± 7.4E-12 | -3.0666E+04 ± 7.4E-12 | -3.0522E+04 ± 6.4E-02 | -3.0655E+04 ± 6.4E-02 | -3.0666E+04 ± 3.5E-01 | -3.0666E+04 ± 7.4E-12 | -3.0546E+04 ± 2.2E-01 | -3.0647E+04 ± 2.2E-01 |
| T31 | **2.6393E+00 ± 4.5E-16** | **2.6393E+00 ± 4.5E-16** | **2.6393E+00 ± 4.5E-16** | 2.6669E+00 ± 7.9E-03 | **2.6393E+00 ± 4.5E-16** | 2.6407E+00 ± 3.0E-03 | 2.6395E+00 ± 3.5E-05 | **2.6393E+00 ± 4.5E-16** | **2.6393E+00 ± 4.5E-16** | **2.6393E+00 ± 4.5E-16** |
| T32 | -6.0817E+03 ± 7.7E+01 | -6.0974E+03 ± 8.6E+01 = | **-6.1055E+03 ± 8.4E+01 =** | -5.9951E+03 ± 1.2E+02 = | -5.8534E+03 ± 3.7E+02 = | -6.0369E+03 ± 1.2E+02 = | -6.091E+03 ± 6.8E+01 = | -6.0029E+03 ± 1.4E+02 = | -6.0897E+03 ± 8.3E+01 = |
| T33 | 6.1597E-02 ± 1.0E-02 | 7.2080E-02 ± 3.7E-02 = | **5.6247E-02 ± 6.8E-02 =** | 5.6247E-02 ± 4.9E-01 = | NaN = | 3.0085E-01 ± 1.2E-01 = | 4.8660E-01 ± 2.1E-01 = | 8.6977E-01 ± 2.2E-01 = | 5.4783E-01 ± 2.7E-01 = | 8.7770E-01 ± 2.9E-01 = |
| T34 | **6.3928E-02 ± 2.7E-02** | 8.2960E-02 ± 9.2E-02 = | 2.0448E-02 ± 2.3E-01 = | 5.8575E-01 ± 3.8E-01 = | NaN = | 4.4930E-01 ± 1.0E-01 = | 5.1686E-01 ± 2.1E-01 = | 3.3827E-01 ± 8.5E-02 = | 5.4783E-01 ± 1.8E-01 = | 4.3158E-01 ± 2.6E-01 = |
| T35 | **2.7727E-02 ± 2.7E-02** | 4.0237E-02 ± 3.8E-02 = | 3.2547E-02 ± 3.6E-02 = | 4.7132E-01 ± 3.1E-01 = | NaN = | 4.8264E-01 ± 1.3E-01 = | 6.5939E-01 ± 2.6E-01 = | 3.7598E-01 ± 9.9E-02 = | 3.7835E-01 ± 1.0E-01 = | 3.0642E-01 ± 1.4E-01 = |
| T36 | 2.076E+00 ± 2.7E-02 | 5.2631E-01 ± 2.4E-01 = | 3.3531E-01 ± 3.2E-01 = | 3.7294E-01 ± 3.1E-01 = | NaN = | 7.9829E-01 ± 1.1E-01 = | 8.6203E-01 ± 2.4E-01 = | 3.8106E-01 ± 9.5E-02 = | 3.8106E-01 ± 1.4E-01 = | 2.5196E-01 ± 1.4E-01 = |
| T37 | 4.3897E-02 ± 1.4E-02 | 1.0493E-01 ± 2.2E-02 = | 3.5657E-02 ± 1.3E-02 = | 1.8840E-01 ± 1.1E-01 = | NaN = | 3.0230E-01 ± 6.7E-02 = | 3.9847E-01 ± 7.9E-02 = | 3.5570E-01 ± 1.8E-01 = | 3.5570E-01 ± 1.8E-01 = | 6.2980E-01 ± 0.0E+00 = |
| T38 | **7.3170E-02 ± 9.0E-02** | 1.8779E-01 ± 3.7E-02 = | 1.0908E-01 ± 1.0E-01 = | 1.7873E-01 ± 9.0E-02 = | NaN = | 3.0230E-01 ± 6.7E-02 = | NaN = | 9.0956E-02 ± 4.6E-02 = | 1.4782E-01 ± 8.1E-02 = | 1.3947E-01 ± 7.3E-02 = |
| T39 | NaN | NaN = | NaN = | NaN = | NaN = | NaN = | NaN = | NaN = | NaN = | NaN = |
| T40 | **4.9686E+03 ± 5.1E+02** | 5.5376E+03 ± 4.3E+02 = | 5.1032E+03 ± 6.9E+02 = | 5.6448E+03 ± 4.5E+03 = | 5.5824E+03 ± 6.1E+02 = | 6.0891E+03 ± 5.1E+02 = | 5.5755E+03 ± 2.9E+02 = | 5.2035E+03 ± 3.0E+02 = |
| T41 | 6.250E+03 ± 0.0E+00 | NaN | 6.0422E+03 ± 3.6E+02 = | **5.6363E+03 ± 1.3E+01 +** | NaN | NaN | NaN | NaN | NaN |
| T42 | NaN | NaN | NaN | NaN | NaN | NaN | NaN | NaN | NaN | NaN |
| +/−/= | Ours | 4/11/27 | 5/10/27 | 7/19/16 | 1/35/6 | 2/31/9 | 0/30/12 | 9/17/16 | 5/29/8 | 4/28/10 |
| #Best | **18** | 15 | 17 | 15 | 2 | 3 | 5 | 17 | 4 | 4 |
| Rank ↓ | **3.54** | 3.65 | 3.79 | 4.95 | 8.90 | 6.75 | 7.05 | 4.23 | 6.24 | 5.90 |

*Table 11.* Average reward (3 rollouts on the 10 independent runs) and standard deviation of the best current solutions found by each algorithm of the 18 reinforcement learning tasks. The best results are highlighted in **bold**. The symbols +, −, and = indicate whether the algorithm is significantly better, worse, or equivalent to the proposed MES-RET according to the Wilcoxon rank-sum test with a significance level of 0.05.

| Task | Environment | sep-MES-RET | wo-RE | wo-RT | sep-CMA-ES | sep-SBCMAES | sep-MTES-KG | TNG-SNES | OpenAI-ES | PPO | A2C |
|---|---|---|---|---|---|---|---|---|---|---|---|
| T1 | MountainCarContinuous-v0 | $9.90E+01 \pm 1.2E-01$ | $\mathbf{9.92E+01 \pm 8.0E-02}$ + | $9.85E+01 \pm 7.9E-01$ = | $9.90E+01 \pm 1.4E-01$ = | $9.64E+01 \pm 1.2E+00$ − | $9.81E+01 \pm 6.5E-01$ − | $9.81E+01 \pm 1.2E+00$ = | $9.72E+01 \pm 9.0E-01$ = | $-1.15E-02 \pm 3.5E-02$ = | $-1.48E-03 \pm 2.2E-03$ = |
| T2 | MountainCar-v0 | $-8.86E+01 \pm 2.8E+00$ | $-8.99E+01 \pm 3.7E+00$ = | $\mathbf{-8.79E+01 \pm 2.0E+00}$ + | $-8.81E+01 \pm 2.0E+00$ = | $-8.88E+01 \pm 1.9E+00$ = | $-9.06E+01 \pm 4.1E+00$ = | $-8.83E+01 \pm 2.6E+00$ = | $-8.81E+01 \pm 2.3E+00$ = | $-1.01E+02 \pm 8.0E+00$ = | $-2.00E+02 \pm 0.0E+00$ = |
| T3 | Pendulum-v1 | $-1.33E+02 \pm 2.1E+02$ | $-3.80E+02 \pm 3.2E+02$ − | $-1.34E+02 \pm 2.1E+02$ = | $-5.35E+01 \pm 3.2E+01$ = | $-1.07E+02 \pm 2.1E+02$ = | $-1.39E+02 \pm 2.1E+02$ = | $\mathbf{-4.33E+01 \pm 3.2E+01}$ + | $-1.76E+02 \pm 2.6E+02$ = | $-1.82E+02 \pm 6.8E+01$ = | $-4.69E+02 \pm 4.8E+02$ = |
| T4 | CartPole-v1 | $\mathbf{5.00E+02 \pm 0.0E+00}$ | $\mathbf{5.00E+02 \pm 0.0E+00}$ = | $\mathbf{5.00E+02 \pm 0.0E+00}$ = | $\mathbf{5.00E+02 \pm 0.0E+00}$ = | $\mathbf{5.00E+02 \pm 0.0E+00}$ = | $\mathbf{5.00E+02 \pm 0.0E+00}$ = | $\mathbf{5.00E+02 \pm 0.0E+00}$ = | $\mathbf{5.00E+02 \pm 0.0E+00}$ = | $\mathbf{5.00E+02 \pm 0.0E+00}$ = | $\mathbf{5.00E+02 \pm 0.0E+00}$ = |
| T5 | Acrobot-v1 | $-6.84E+01 \pm 2.2E+00$ | $-6.63E+01 \pm 1.9E+00$ = | $-6.57E+01 \pm 2.0E+00$ + | $-6.52E+01 \pm 2.4E+00$ + | $-6.58E+01 \pm 3.2E+00$ + | $-6.97E+01 \pm 2.1E+00$ = | $-6.64E+01 \pm 1.6E+00$ + | $-6.78E+01 \pm 1.7E+00$ = | $\mathbf{-6.08E+01 \pm 5.3E-01}$ + | $-6.68E+01 \pm 5.1E+00$ = |
| T6 | LunarLander-v3 | $2.97E+02 \pm 5.6E+00$ | $2.83E+02 \pm 4.7E+01$ = | $\mathbf{3.02E+02 \pm 5.5E+00}$ + | $2.89E+02 \pm 2.0E+01$ = | $2.95E+02 \pm 1.6E+01$ = | $2.96E+02 \pm 5.6E+00$ = | $2.98E+02 \pm 6.6E+00$ = | $2.98E+02 \pm 1.5E+01$ = | $2.82E+02 \pm 1.0E+01$ = | $1.38E+02 \pm 1.2E+02$ = |
| T7 | BipedalWalker-v3 | $1.87E+02 \pm 1.0E+02$ | $1.77E+02 \pm 3.4E+01$ = | $1.79E+02 \pm 4.4E+01$ = | $2.21E+02 \pm 5.0E+01$ = | $1.70E+02 \pm 3.0E+01$ = | $1.60E+02 \pm 4.1E+01$ = | $1.81E+02 \pm 4.5E+01$ = | $2.01E+02 \pm 3.6E+01$ = | $\mathbf{2.30E+02 \pm 1.2E+02}$ + | $1.36E+02 \pm 1.4E+02$ = |
| T8 | InvertedPendulum-v5 | $\mathbf{1.00E+03 \pm 0.0E+00}$ | $\mathbf{1.00E+03 \pm 0.0E+00}$ = | $\mathbf{1.00E+03 \pm 0.0E+00}$ = | $\mathbf{1.00E+03 \pm 0.0E+00}$ = | $\mathbf{1.00E+03 \pm 0.0E+00}$ = | $\mathbf{1.00E+03 \pm 0.0E+00}$ = | $\mathbf{1.00E+03 \pm 0.0E+00}$ = | $\mathbf{1.00E+03 \pm 0.0E+00}$ = | $\mathbf{1.00E+03 \pm 0.0E+00}$ = | $6.57E+02 \pm 4.4E+02$ = |
| T9 | InvertedDoublePendulum-v5 | $\mathbf{9.36E+03 \pm 5.2E-02}$ | $9.36E+03 \pm 8.4E-02$ = | $9.36E+03 \pm 1.5E+00$ = | $9.36E+03 \pm 1.4E+00$ = | $1.31E+03 \pm 1.5E+03$ − | $7.64E+03 \pm 3.3E+03$ = | $9.36E+03 \pm 8.5E-02$ = | $9.36E+03 \pm 7.6E-02$ = | $5.02E+03 \pm 4.1E+03$ = | $9.36E+03 \pm 1.3E+00$ = |
| T10 | Reacher-v5 | $-4.00E+00 \pm 8.3E-01$ | $-4.45E+00 \pm 1.2E+00$ = | $-5.77E+00 \pm 1.1E+00$ − | $-4.84E+00 \pm 7.2E-01$ = | $-1.05E+01 \pm 2.1E+00$ − | $-5.96E+00 \pm 1.9E+00$ = | $-5.04E+00 \pm 5.6E-01$ = | $\mathbf{-3.51E+00 \pm 6.1E-01}$ + | $-3.75E+00 \pm 4.3E-01$ = | $-3.53E+00 \pm 7.1E-01$ = |
| T11 | Pusher-v5 | $-3.30E+01 \pm 1.6E+00$ | $-3.83E+01 \pm 1.9E+00$ − | $-5.17E+01 \pm 4.2E+00$ − | $-5.14E+01 \pm 6.2E+00$ = | $-7.52E+01 \pm 1.1E+01$ − | $-4.81E+01 \pm 5.2E+00$ = | $-4.42E+01 \pm 3.3E+00$ = | $-3.21E+01 \pm 1.9E+00$ = | $\mathbf{-2.63E+01 \pm 1.6E+00}$ + | $-3.74E+01 \pm 4.3E+00$ = |
| T12 | HalfCheetah-v5 | $\mathbf{5.76E+03 \pm 5.0E+02}$ | $5.63E+03 \pm 4.6E+02$ = | $4.90E+03 \pm 4.2E+02$ − | $2.68E+03 \pm 8.7E+02$ = | $3.63E+03 \pm 1.3E+03$ = | $4.26E+03 \pm 8.8E+02$ = | $4.87E+03 \pm 5.2E+02$ = | $5.54E+03 \pm 3.0E+02$ = | $2.98E+03 \pm 1.6E+03$ = | $1.29E+03 \pm 9.2E+02$ = |
| T13 | Hopper-v5 | $\mathbf{3.31E+03 \pm 2.0E+02}$ | $2.65E+03 \pm 1.0E+03$ = | $2.49E+03 \pm 9.2E+02$ = | $3.56E+03 \pm 8.9E+02$ = | $2.55E+03 \pm 8.6E+02$ = | $2.70E+03 \pm 8.0E+02$ = | $2.46E+03 \pm 8.9E+02$ = | $3.02E+03 \pm 8.9E+02$ = | $2.81E+03 \pm 7.9E+02$ = | $1.58E+03 \pm 1.1E+03$ = |
| T14 | Walker2d-v5 | $3.58E+03 \pm 6.8E+02$ | $3.20E+03 \pm 4.8E+02$ = | $2.92E+03 \pm 7.5E+02$ = | $3.42E+02 \pm 6.2E+01$ = | $3.13E+03 \pm 4.9E+02$ = | $3.15E+03 \pm 7.1E+02$ = | $3.31E+03 \pm 7.5E+02$ = | $2.49E+03 \pm 1.2E+03$ = | $\mathbf{4.47E+03 \pm 4.7E+02}$ + | $1.52E+03 \pm 1.2E+03$ = |
| T15 | Swimmer-v5 | $2.61E+02 \pm 1.1E+02$ | $2.32E+02 \pm 1.4E+02$ = | $2.33E+02 \pm 1.1E+02$ = | $1.04E+03 \pm 1.1E+00$ = | $2.97E+02 \pm 9.6E+01$ = | $3.20E+02 \pm 8.6E+01$ = | $\mathbf{3.43E+02 \pm 6.2E+01}$ + | $7.92E+01 \pm 4.4E+01$ = | $1.31E+02 \pm 2.9E+00$ = | $3.66E+01 \pm 3.4E+00$ = |
| T16 | Ant-v5 | $\mathbf{2.35E+03 \pm 3.2E+02}$ | $1.19E+03 \pm 8.4E+01$ − | $1.16E+03 \pm 1.4E+02$ − | $1.01E+03 \pm 9.9E+01$ = | $1.01E+03 \pm 1.2E+02$ = | $1.29E+03 \pm 9.0E+01$ = | $1.58E+03 \pm 1.7E+02$ = | $2.12E+03 \pm 3.2E+02$ = | $2.24E+03 \pm 4.1E+02$ = | $2.89E+02 \pm 3.7E+02$ = |
| T17 | Humanoid-v5 | $7.08E+02 \pm 5.0E+01$ | $7.37E+02 \pm 7.0E+01$ = | $7.52E+02 \pm 7.4E+01$ = | $7.45E+02 \pm 5.6E+01$ = | $\mathbf{7.94E+02 \pm 7.5E+01}$ + | $7.09E+02 \pm 4.4E+01$ = | $7.46E+02 \pm 7.4E+01$ = | $7.65E+02 \pm 8.9E+01$ = | $4.17E+02 \pm 5.6E+01$ = | $4.38E+02 \pm 8.5E+01$ = |
| T18 | HumanoidStandup-v5 | $1.61E+05 \pm 1.1E+04$ | $1.64E+05 \pm 1.4E+04$ = | $1.72E+05 \pm 1.5E+04$ + | $\mathbf{1.73E+05 \pm 2.5E+04}$ + | $1.66E+05 \pm 2.3E+04$ = | $1.72E+05 \pm 2.2E+04$ = | $1.67E+05 \pm 2.1E+04$ = | $1.55E+05 \pm 7.5E+03$ = | $8.84E+04 \pm 7.8E+03$ − | $7.00E+04 \pm 1.6E+04$ = |
| | +/−/= | Ours | 1/3/14 | 3/6/9 | 1/5/12 | 2/7/9 | 0/7/11 | 1/5/12 | 0/4/14 | 3/10/5 | 0/14/4 |
| | #Best ↑ | 6 | 3 | 4 | 3 | 3 | 2 | 4 | 3 | 6 | 1 |
| | Rank ↓ | 4.03 | 5.42 | 5.08 | 4.39 | 6.47 | 6.31 | 4.47 | 4.67 | 5.75 | 8.42 |

(average rank 1.65) successfully resists performance degradation and significantly outperforms the uniform allocation baseline w/o-RE (rank 2.85, win/loss/tie: 5/230/765 against MES-RET).

Furthermore, to counteract long-tail probability dilution, we introduced a hierarchical grouping mechanism (w/-Group) based on momentum-based stratified clustering. Specifically, tasks are sorted descendingly by their current reward signals and partitioned into fixed-size strata (*e.g.*, 50 tasks per group). To prevent negative transfer across tasks with disparate evolutionary dynamics, knowledge aggregation (mean and covariance) is restricted strictly within each intra-stratum subspace. The evaluation budget is then allocated hierarchically: a macro-level budget is first distributed to each group proportional to its aggregate reward, followed by a micro-level categorical sampling to allocate evaluations to specific tasks within the group. This extension successfully isolated conflicting sub-manifolds and yielded a further efficiency boost, achieving a rank of 1.50 and a win/loss/tie of 51/15/934 against the base framework. This confirms the robustness and adaptability of our resource allocation strategy at massive scales.

