# OpenReview forum: "Breaking Multi-Task Curse: Reward-Weighted Evolution for Black-Box Many-Task Optimization"
_ICML.cc/2026/Conference — ICML 2026 regular_

### Official Review · Reviewer_Bgwp · 2026-03-01

**Soundness:** 3
**Presentation:** 3
**Significance:** 3
**Originality:** 3
**Overall Recommendation:** 4
**Confidence:** 4

**Summary:**

This paper introduces a black-box optimization framework named MES-RET to solve the "multitasking curse" problem in evolutionary multitasking processing. This method dynamically allocates computing resources through a reward-weighted evaluation mechanism, giving priority to tasks with higher convergence or exploration potential, thereby significantly improving the solution efficiency. To prevent negative transfer between unrelated tasks, researchers have designed a robust knowledge transfer mechanism that uses global statistics to provide weak guidance and ensures that it can automatically revert to an independent evolutionary state when the effect is poor. In addition, to address the challenge of dimension mismatch in reinforcement learning, this paper proposes a semantic policy parameter alignment strategy, achieving skill sharing across heterogeneous tasks. A large number of experiments have proved that this method is superior to the existing single-task and multi-task baseline algorithms in fields such as synthetic functions, engineering design, and complex robot control.

**Compliance With Llm Reviewing Policy:**

Affirmed.

**Key Questions For Authors:**

1) In the semantic policy parameter alignment mechanism, if the hidden layer dimensions between tasks are completely different or there are non-fully connected layers (such as convolutional layers), how can this framework be effectively extended?

2) When the number of tasks expands from dozens to hundreds or even thousands, will categorical sampling in reward-weighted evaluation fail due to extremely diluted probabilities?

**Limitations:**

yes

**Strengths And Weaknesses:**

**Strengths**
1) Clearly identify and define the two major causes of the "multitasking curse" : evaluation budget dispersion and negative transfer.

2) The paper not only presents the algorithm but also provides theoretical proof.

3) The proposed reward weighted evaluation and reward weighted transfer mechanisms can dynamically adjust resources and aggregate cross-task knowledge. Meanwhile, this algorithm maintains the same asymptotic time complexity as the standard CMA-ES and has extremely low computational overhead.

**Weaknesses**
1) The semantic policy alignment policy currently relies on a fixed size of hidden layers, which limits its direct application between tasks with completely different neural network architectures, such as different numbers of layers or different types.

2) The sample efficiency of the ES strategy is low.

3) Currently, the alignment is mainly focused on MLP structures. More complex structures (such as CNN or Transformer) are not discussed in depth in this paper.

---

> ### Author Rebuttal · Authors · 2026-03-30
>
> We sincerely thank you for the insightful comments and thoughtful questions.
>
> # For W1, W3 & Q1 (Extensibility of Semantic Alignment)
>
> MES-RET establishes a rigorously bounded mechanism for cross-task transfer: **Semantic Alignment** maximizes the transfer **upper bound** when parameter semantics align, while **Weak Guidance Injection** acts as a strict safeguard, guaranteeing the **lower bound** by neutralizing negative transfer. Although empirically validated on 2-layer MLPs, our functional alignment strategy, supported by the computationally efficient $\\mathcal{O}(n)$ covariance updates can safely generalize to diverse architectures:
>
> **1. Other Homogeneous Architectures (CNN/Transformer)**
> Align tensors by flattening while preserving layer-wise semantics:
> - CNNs: Direct alignment is performed exclusively across identical topological structures (matching $C\_{out}, C\_{in}, K\_h, K\_w$). For networks with varying channel capacities, alignment can be restricted to the terminal fully-connected policy heads. This allows independently optimize feature extractors for diverse visual inputs while safely sharing motor-control semantics in the final layers.
> - Transformers: Core sequence-modeling semantics are transferred by aligning functionally consistent matrices (attention projections $W\_Q, W\_K, W\_V, W\_O$, LayerNorm $\\gamma, \\beta$, and shared FFNs). Task-specific components (e.g., positional encodings, prediction heads) remain strictly independent.
>
> **2. Homogeneous Architectures with Dimensionality Mismatches**
> Semantic correspondence across varying depths or widths:
> - Direct Matching: Alignment is constrained to structural intersections, such as common prefix layers ($L\_{\\min}$) or overlapping hidden dimensions ($\\min(d\_A, d\_B)$), leaving residual parameters isolated.
> - Latent Mapping: Leveraging domain adaptation, orthogonal projections or autoencoders map disparate parameter covariances into a unified latent space.
> - Note that these approaches may not always yield positive transfer. However, the Weak Guidance Injection mechanism ensures that negative transfer is neutralized.
>
> **3. Heterogeneous Architectures**
> - Asymmetric Alignment: If vision-state task A uses CNN+MLP and vector-state task B uses MLP, we selectively align the terminal policy heads sharing identical motor-control semantics, leaving the heterogeneous feature extractors independent.
> - Extreme Heterogeneity: While complete architectural disparity implies minimal reusable knowledge, attempting transfer (e.g. latent mapping) remains mathematically safe. Our mechanism automatically assigns zero weight to negative knowledge.
>
> While MLPs serve as an empirical baseline, our core novelty is systematically breaking the Multi-Task Curse. The framework optimizes transfer whenever a semantic mapping is definable and guarantees a safe fallback to independent evolution.
>
> # For W2 (Sample Efficiency of ES)
>
> We agree that ES inherently exhibits lower sample efficiency. However, this trade-off yields exceptional algorithmic robustness and stability. Unlike gradient-based RL, which frequently suffers from deceptive gradients or extreme hyperparameter sensitivity, ES reliably converges across diverse environments with sufficient budget, using a single standard hyperparameter set. Furthermore, ES features low time complexity and is highly parallelizable, effortlessly scaling to distributed multi-CPU/GPU evaluations to offset wall-clock time.
>
> Crucially, MES-RET is designed precisely to alleviate this inherent sample inefficiency. By enabling cross-task knowledge transfer and dynamic resource allocation, our method structurally accelerates convergence in many-task regimes. This effectively bridges the gap between gradient-free stability, massive parallelizability, and practical sample efficiency.
>
> # For Q2 (Scalability of Categorical Sampling)
>
> We agree with the insightful observation. Under our current standard Softmax formulation, the categorical sampling will indeed suffer from probability dilution. As the number of tasks $K$ scales to hundreds or thousands, the normalized probabilities $p\_k$ mathematically flatten out (approaching $1/K$). This severe dilution diminishes the selection pressure, essentially degrading the reward-weighted allocation back to a uniform baseline.
>
> In practical extreme-scale scenarios, this limitation can be resolved by introducing sparsity and hierarchical mechanisms:
> 1. Filtering (Top-K Truncation): Instead of assigning probabilities across all tasks, we can apply a hard threshold to sample exclusively from the top-tier tasks (e.g., top 10% with the highest progress), setting the allocation probabilities of the long-tail tasks strictly to zero.
> 2. Grouping (Task Clustering): Thousands of tasks can be dynamically grouped into a smaller number of subsets based on variation similarity. The reward-weighted sampling can then be executed hierarchically (first selecting a high-yield cluster, then sampling within it).

---

> > ### Author Rebuttal · Reviewer_Bgwp · 2026-04-03
> >
> > The authors answered some of my confusions. However, for the questions raised, if they are used as supplementary experiments, they can enrich the content of the paper. Therefore, I choose not to change the score.

---

> > > ### Author Response · Authors · 2026-04-04
> > >
> > > We sincerely appreciate your constructive feedback. To further validate the robustness and scalability of our method, we have conducted additional experiments:
> > >
> > > # Network Architecture Extension Validation
> > >
> > > To address your insightful query regarding tasks with completely different neural network architectures (e.g., CNNs), we designed a minimal viable experiment to explicitly validate our semantic policy parameter alignment across asymmetric, heterogeneous architectures. We selected four continuous control tasks: `HalfCheetah` and `Hopper` employ standard MLPs, whereas `Walker2d` and `Ant` utilize a 1D-CNN feature extractor coupled with an MLP policy head.
> > >
> > > To effectively extend our semantic alignment mechanism across these disparate structures, we implemented a strict topological decoupling of the parameter vectors. Specifically, the network parameters are partitioned into shared and task-specific segments. For the shared segment, we strictly align the parameters of the identical decision-making layers: the biases of the first hidden layer ($b\_1 \\in \\mathbb{R}^h$) and the weights and biases of all subsequent intermediate layers ($W^{(l)} \\in \\mathbb{R}^{h \\times h}, b^{(l)} \\in \\mathbb{R}^h$). These shared parameters are uniformly packed at the very front of the parameter array. Conversely, the task-specific parameters are isolated at the tail. For the MLPs, this tail includes the linear input weights ($W\_{in} \\in \\mathbb{R}^{d\_{obs} \\times h}$), while for the CNNs, it includes the 1D convolutional filters ($W\_{conv}$) and the flattening projection weights ($W\_{fc} \\in \\mathbb{R}^{D\_{flatten} \\times h}$), alongside the varying output layers ($W\_{out}$) for different action spaces.
> > >
> > > The empirical results (Reward with Standard Deviation) of this asymmetric alignment are summarized in the table below:
> > >
> > > | Task (Architecture) | sep-MES-RET          | w/o-RT                |
> > > | :------------------ | :------------------- | :------------------- |
> > > | HalfCheetah (MLP)   | **5527.16 (525.94)** | 4978.99 (638.23)     |
> > > | Hopper (MLP)        | **2913.40 (710.51)** | 2281.76 (878.17)     |
> > > | Walker2d (CNN+MLP)  | **2845.42 (232.21)** | 2716.71 (703.37)     |
> > > | Ant (CNN+MLP)       | 1322.88 (249.46)     | **1397.13 (187.65)** |
> > >
> > > The data substantiates the efficacy of our topological decoupling strategy for heterogeneous architectures. `sep-MES-RET` demonstrates significant performance gains over the independent evolution baseline (`w/o-RT`) on `HalfCheetah`, `Hopper`, and `Walker2d`. Crucially, the successful enhancement of `Walker2d` (which employs a CNN feature extractor) validates our claim: latent motor-control semantics can be mapped and positively transferred across highly disparate architectures, provided the terminal decision-making sub-manifolds are rigorously aligned.
> > >
> > > Conversely, for the `Ant` environment, we observe a slight performance deficit compared to the baseline (1322.88 vs. 1397.13). However, this result validates the lower-bound guarantee of our framework. The Weak Guidance Injection mechanism successfully detects this semantic misalignment and strictly bounds the negative transfer. Rather than experiencing catastrophic divergence due to the heterogeneous parameter mismatch, the mechanism enables `sep-MES-RET` to degrade gracefully to an asymptotic performance bound functionally comparable to independent optimization.
> > >
> > > # Extreme-Scale Tasks Validation
> > >
> > > To explicitly validate the extreme scalability ($T=1000$) you mentioned, we conducted additional experiments using the Planar Kinematic Arm Control Problem, which can scale to thousands of tasks by varying the arm length and target position. To rigorously construct a low-similarity search space, we generated $1000$ tasks with four distinct heterogeneous task modes (Easy, Normal, Unreachable, and Conflicting) featuring random dimensionality (2–50).
> > >
> > > Interestingly, contrary to the theoretical expectation that categorical sampling might fail due to severe probability dilution at this scale, empirical results demonstrate that the base `MES-RET` remains remarkably effective. Even at $T=1000$, the base framework (rank 1.65) successfully resists performance degradation and significantly outperforms the `w/o-RE` baseline (rank 2.85, win/loss/tie: 5/230/765). Furthermore, the hierarchical grouping mechanism proposed in our rebuttal (`w/-Group`) successfully isolates conflicting sub-manifolds, yielding a further efficiency boost (rank 1.50, win/loss/tie: 51/15/934 against the Base).
> > >
> > > Finally, thank you again for the insightful comments and questions. We hope this additional analysis and empirical validation further clarifies the robustness and scalability of our work.

---

### Official Review · Reviewer_T2zD · 2026-03-12

**Soundness:** 3
**Presentation:** 2
**Significance:** 3
**Originality:** 3
**Overall Recommendation:** 4
**Confidence:** 2

**Summary:**

This paper studies the multi-task curse in black-box many-task optimization, where jointly optimizing a large number of tasks can lead to worse performance than solving them independently due to evaluation budget dispersion and negative knowledge transfer. To address this issue, the authors propose MES-RET (Many-task Evolution Strategy with Reward-weighted Evaluation and Transfer). The method introduces to dynamically allocate evaluation budgets across tasks and weights cross-task knowledge transfer. Experiments on benchmark many-task optimization problems demonstrate that the proposed method mitigates the degradation observed in existing multi-task evolutionary algorithms and achieves improved performance.

**Compliance With Llm Reviewing Policy:**

Affirmed.

**Final Justification:**

The authors answered my questions in detail. However, I have low confidence in reviewing this paper.

**Key Questions For Authors:**

Q1. In the chosen experiments, how much multi-task curse do they have? Is there any way to show the negative transfer? I am asking because researchers show gradient cosine similarity in multi-objective optimization experiments where gradient conflict is a source of negative transfer. However, in black-box optimization, there is no information about gradients. But if showing some negative transfer in other methods, while showing less negative transfer in this method, may help.

Note: This paper's scope is a bit beyond my background. I would modestly read other reviewers' comments and change my opinion accordingly.

**Limitations:**

Please refer to the weaknesses and questions.

**Strengths And Weaknesses:**

S1. The paper provides a clear diagnosis of why many-task optimization methods degrade as the number of tasks increases, highlighting evaluation budget dispersion and negative transfer.

S2. Reward-weighted evaluation and transfer provide an intuitive mechanism for prioritizing promising tasks and filtering harmful information sharing.

W1. The paper is sometimes difficult to follow due to limited background explanations and unclear presentation of some components. For example, several notations appear before being formally defined (e.g., the symbol $\lambda$ in line 180). In addition, some sub-algorithms used in the method, such as the CMA-based evolution procedure, are introduced with minimal explanation. Since these components are essential to understanding the proposed method, the paper would benefit from more detailed descriptions or background introductions.

---

> ### Author Rebuttal · Authors · 2026-03-30
>
> We sincerely thank you for the thoughtful comments and insightful questions.
>
> # For W1 (Clarity, Background, and Notations)
>
> We sincerely apologize for the undefined notations and the brevity of the background section.
>
> To clarify the core concepts: in the Covariance Matrix Adaptation Evolution Strategy (CMA-ES) backbone, $\\lambda$ denotes the total population size (number of candidate solutions generated per generation), and $\\mu$ represents the number of top-performing elites selected to update the search distribution via Eq. (1). CMA-ES optimizes black-box problems by iteratively updating a multivariate or separable Gaussian distribution (parameterized by a mean vector and a covariance matrix) based on the fitness rankings of these $\\lambda$ samples.
>
> The complete algorithmic steps of the CMA-based evolution procedure, including the specific mathematical update rules, are detailed in Algorithm 2 in Appendix. We will ensure all symbols (e.g., $\\lambda$, $\\mu$) are explicitly defined upon their first appearance and will add more background on CMA-ES in Appendix to improve readability for a broader audience.
>
> # For Q1 (Multi-Task Curse and Negative Transfer Demonstration)
>
> We agree that gradient conflict fundamentally drives negative transfer. While exact analytical gradients are unavailable in BBO, we can rigorously formalize this conflict via the natural gradients of the search distribution.
>
> *(See **Reviewer FpL8** for Multi-Task Curse formulation)*
>
> As detailed in the formulation, negative transfer is mathematically governed by the alignment between the transferred knowledge $\\Theta\_k = (\\delta m\_k, \\delta C\_k)$ and the task's true local landscape:
> $$\\text{Negative Transfer} \\approx \\sum\_{k=1}^K \\left( \\underbrace{\\langle \\nabla f\_k, \\delta m\_k \\rangle}\_{\\text{Mean Shift}} + \\underbrace{\\text{Tr}\\left(\\frac{1}{2}H\_k \\delta C\_k\\right)}\_{\\text{Covariance Distortion}} \\right)$$
>
> In low-similarity many-task environments, optimal task means and covariances drastically diverge. Consequently, external knowledge introduces updates ($\\delta m\_k$, $\\delta C\_k$) that severely conflict with the local gradient ($\\nabla f\_k$) and Hessian ($H\_k$), driving both terms deeply negative.
>
> ### Demonstrating Severe Negative Transfer
> In practical BBO, the true objective function $f\_k$ and its exact derivatives ($\\nabla f\_k$, $H\_k$) are strictly inaccessible a priori. Consequently, the exact occurrence and magnitude of these theoretical gradient conflicts cannot be explicitly predicted. Instead, the severity of negative transfer and the Multi-Task Curse must be empirically quantified through macroscopic performance degradation relative to independent single-task baselines.
>
> Baseline multi-task methods (e.g., TNG-NES, SBCMAES) lack a mathematical firewall against this unpredictability. They unrestrictedly absorb potentially conflicting updates directly into their search distributions. This structural vulnerability empirically manifests as massive negative transfer: as shown in Fig. 1 and Tables 1-2, these baselines consistently fall into the red zone, performing substantially worse than completely independent CMA-ES. This widespread degradation explicitly proves that the Multi-Task Curse in our chosen experiments is exceptionally severe.
>
> ### How MES-RET Neutralizes It
> Our method systematically blocks this degradation by guaranteeing a safe transfer lower bound via **Weak Guidance Injection**. Recognizing that divergent tasks yield conflicting gradients, we refuse to let external knowledge directly update the distribution. Instead, we inject it strictly as a discrete candidate ($\\tau=1$) subject to actual fitness evaluation on the target landscape.
>
> If the transferred knowledge conflicts with the local search, it falls below the truncation boundary $\\mu$. Furthermore, due to the non-linear recombination of CMA-ES:
> $$w\_i \\propto \\max(0, \\ln(\\mu + 0.5) - \\ln(i))$$
> even if mediocre conflicting knowledge barely survives truncation, its update weight $w\_i$ decays exponentially toward $0$. Thus, conflicting $\\delta m\_k$ and $\\delta C\_k$ are automatically filtered before they can corrupt the inner-task optimization. As empirically validated, our method avoids the performance collapse of peer MT methods, rigorously neutralizing negative transfer while harvesting positive transfer.

---

> > ### Author Rebuttal · Reviewer_T2zD · 2026-04-03
> >
> > Thanks for the detailed answer to resolve my question. I would like to raise my rating, but my confidence remains.

---

> > > ### Author Response · Authors · 2026-04-04
> > >
> > > We sincerely appreciate your time in reviewing our rebuttal. We are very glad that our response has resolved your questions.
> > >
> > > Thank you again for your constructive feedback and for raising your rating. Your insights have helped us improve the clarity of our work.

---

### Official Review · Reviewer_jwef · 2026-03-12

**Soundness:** 3
**Presentation:** 3
**Significance:** 3
**Originality:** 3
**Overall Recommendation:** 4
**Confidence:** 2

**Summary:**

This paper addresses the "Multi-Task Curse" in Evolutionary Multi-Tasking (EMT), a problem characterized by evaluation budget dispersion and negative transfer. To address these issues, the authors propose MES-RET.  This framework introduces an adaptive budget allocation strategy based on a reward-weighted evaluation mechanism and employs a reward-weighted transfer mechanism with weak guidance injection to enhance knowledge transfer.

**Compliance With Llm Reviewing Policy:**

Affirmed.

**Final Justification:**

I appreciate the authors’ detailed responses and the additional experiments provided during the rebuttal period. The rebuttal has strengthened my view of the paper and highlighted several positive aspects. However, due to my limited confidence in this area, I maintain my evaluation of the paper as a *weak accept*.

**Key Questions For Authors:**

1. Can you provide a theoretical justification or stronger empirical evidence demonstrating that the proposed reward design reliably satisfies the condition in Equation 3?

2. Regarding Figure 7, the initial starting states for sep-MES-RET and w/o-RT appear to be different. Is this a fair visual comparison? For a more rigorous assessment, shouldn't both models begin from the exact same initial state?

3. Could you provide an additional ablation study that isolates the effect of the Weak Guidance Injection? Seeing the performance drop when only this specific component is removed would help clarify its true impact on the system.

**Limitations:**

yes

**Strengths And Weaknesses:**

**Strengths**

- *Effective Knowledge Transfer*:
The proposed Mean & Covariance aggregation method offers a simple yet highly efficient approach to knowledge transfer.

- *Strong Empirical Performance*:
The method demonstrates excellent performance, consistently outperforming both single-task and many-task Evolution Strategies (ES) baselines. Furthermore, it achieves results comparable to gradient-based RL algorithms like PPO on reinforcement learning benchmarks.

&nbsp;

**Weaknesses**

- *Lack of Theoretical Guarantee for the Reward Design*:
Proposition 3.1 successfully establishes that a reward-weighted strategy is superior to a uniform baseline. However, the paper lacks a theoretical guarantee or proof demonstrating that the specifically proposed reward design actually satisfies the required condition outlined in Equation 3..

- *Simplistic Mitigation for Negative Transfer*:
While the paper heavily emphasizes the severity of negative transfer, the proposed solution, weak guidance injection, feels somewhat too simplistic to fully address the stated magnitude of the problem.

---

> ### Author Rebuttal · Authors · 2026-03-30
>
> We sincerely thank you for the insightful concerns and thoughtful questions.
>
> # For W1 Q1 (Theoretical Justification of Reward Design)
>
> A universal theoretical guarantee is mathematically precluded in highly deceptive landscapes. Instead, our formulation satisfies Eq. 3 via a dual-mechanism approach, blending continuous-space approximation ($r^{\\text{perf}}$) with an empirical safeguard ($r^{\\text{div}}$) for CMA-ES dynamics.
>
> **1. Performance Reward:** Assuming local smoothness, CMA-ES performs approximate natural gradient descent on the Gaussian-smoothed objective $J\_k(m\_k, C\_k) = \\mathbb{E}\_{x\\sim\\mathcal{N}}[f\_k(x)]$ (Wierstra 2014). Current natural gradients strongly correlate with subsequent steps. Thus, historical improvement ($r\_k^{\\text{perf}}$) mathematically approximates local expected improvement $\\mathbb{E}[\\Delta\_k]$. Since $r\_k^{\\text{perf}}$ dominates early/mid-stage optimization, this momentum guarantees Eq. 3 is satisfied during the vast majority of the search.
>
> **2. Diversity Reward:** Designed to counteract CMA-ES's tendency for premature covariance collapse in deceptive basins. Formulating $r\_k^{\\text{div}}$ via $\\text{trace}(C\_k)$ implicitly lower-bounds the distribution's differential entropy. Acting purely as an auxiliary fallback when $r\_k^{\\text{perf}}$ vanishes, it encourages exploratory variance and preserves the structural prerequisites for future improvement.
>
> **Empirical Validation:** The `w/o-RE` ablation consistently underperforms our full method across all metrics, empirically verifying $\\mathbb{E}[\\Delta\_{\\text{total}}^{\\text{weighted}}] \\geq \\mathbb{E}[\\Delta\_{\\text{total}}^{\\text{uniform}}]$. Per Prop. 3.1, this confirms our dynamic reward reliably satisfies the positive correlation condition in expectation.
>
> # For Q2 (Initial State Issue)
>
> All algorithms were initialized from the exact same starting states using the identical 10 random seeds. The visual little divergence at the beginning of the plot is simply an artifact of the logging frequency. The first data point shown is not generation 0, but rather the first logging interval (generation 10, as we sample exactly 50 recording points across the entire evaluation budget). By this first recorded generation, the algorithms have already diverged due to their distinct early-stage optimization dynamics.
>
> # For W2 Q3 (Impact of Weak Guidance Injection)
>
> In ES, minimalistic design is often the key to algorithmic robustness. The simplicity of weak guidance injection is precisely its greatest strength: it elegantly solves a complex problem without introducing heavy computational overhead.
>
> Its effectiveness stems from ingeniously exploiting the inherent, non-linear rank-based truncation selection of the CMA-ES backbone. The mitigation operates as a strict mathematical firewall: By constraining the external injection to a minimal scale ($\\tau=1$), we strictly bound the potential exploration damage; As proven in Prop 3.3, any maladaptive transferred solution is subject to strict fitness evaluation. If it triggers negative transfer, it drops in rank, and its recombination weight $w\_i$ decays exponentially to $0$.
>
> Original `w/o-RT` variant removes the transfer. To explicitly isolate Weak Guidance Injection effect, we conduct additional ablation studies with two variants:
> 1. `w/o-TS`: Without rank-based truncation safeguard (Prop 3.3), allowing transferred samples with positive weights (randomly replace top-$\\mu$) to influence the search.
> 2. `w/-SI`: With strong injection, size from $\\tau=1$ to $\\tau=0.1\\lambda$ ($0.2\\lambda$ external samples), forcing a massive influx of external knowledge.
>
> We evaluated these on synthetic benchmarks, real-world tasks, and policy search. The average rankings (Ours $\\to$ `w/o-RT` $\\to$ `w/o-TS` $\\to$ `w/-SI`) are 1.89 $\\to$ 2.01 $\\to$ 2.78 $\\to$ 3.32 (Synthetic), 1.74 $\\to$ 1.89 $\\to$ 2.86 $\\to$ 3.51 (Real-World), and 1.67 $\\to$ 2.33 $\\to$ 2.78 $\\to$ 3.22 (Policy Search), with the corresponding Wilcoxon +/-/= metrics of the three variants against our method being 1/7/79 $\\to$ 4/48/35 $\\to$ 8/61/18 (Synthetic), 5/10/27 $\\to$ 4/29/9 $\\to$ 2/34/6 (Real-World), and 3/6/9 $\\to$ 1/13/4 $\\to$ 1/9/8 (Policy Search).
>
> Unsurprisingly, both new variants perform drastically worse than the original and `w/o-RT`. In highly heterogeneous many-task environments, absorbing excessive external knowledge (`w/-SI`) or unfiltered minimal knowledge (`w/o-TS`) inevitably injects negative transfer. This catastrophic performance collapse explicitly validates the Multi-Task Curse. It demonstrates that Weak Guidance Injection is a necessary mathematical firewall that securely filters negative transfer.

---

> > ### Author Rebuttal · Reviewer_jwef · 2026-04-02
> >
> > Thank you to the authors for their detailed and thoughtful response. The rebuttal addresses my concerns regarding the theoretical justification of the reward design by providing clear intuition behind the proposed formulation. While it does not offer a strict theoretical guarantee, the explanation is reasonable and sufficiently justifies the design choices.
> >
> > Additionally, the newly provided ablation studies on Weak Guidance Injection adequately address my request and help clarify its impact on the overall system.

---

> > > ### Author Response · Authors · 2026-04-02
> > >
> > > We sincerely appreciate your time in reviewing our rebuttal. We are very glad that our reply regarding the reward design and the additional ablation studies on Weak Guidance Injection have resolved your concerns.
> > >
> > > Thank you again for your rigorous and constructive feedback.

---

### Official Review · Reviewer_FpL8 · 2026-03-12

**Soundness:** 2
**Presentation:** 3
**Significance:** 2
**Originality:** 2
**Overall Recommendation:** 4
**Confidence:** 2

**Summary:**

The paper introduces MES-RET, an evolutionary multi-tasking algorithm designed for many-task optimization problems. It dynamically allocates evaluation budgets and transfers mean and covariance statistics based on task-specific reward signals. It also presents a semantic policy parameter alignment strategy to enable knowledge transfer across reinforcement learning policies with differing state-action dimensionalities.

**Compliance With Llm Reviewing Policy:**

Affirmed.

**Final Justification:**

After rebuttal, author clarified the architectural extensions and the theoretical limits of the reward mechanism. With the supplementary experiments provided, it addresses my initial concerns.

**Key Questions For Authors:**

Please refer to the Weaknesses.

**Limitations:**

Yes.

**Strengths And Weaknesses:**

Strengths
1.	The weak guidance injection mechanism offers a theoretically grounded safeguard against negative transfer by discarding maladaptive external solutions through rank-based truncation.
2.	The empirical evaluation is broad, spanning 87 synthetic benchmarks, 42 real-world constrained engineering tasks, and 18 Gymnasium environments, demonstrating robust performance across diverse domains.
3.	The covariance and mean aggregation mechanism provides a computationally lightweight approach to cross-task knowledge transfer without requiring explicit feature mapping or secondary learning loops.

Weaknesses
1.	The core allocation and aggregation mechanisms offer incremental heuristic adjustments. Is the Semantic Policy Parameter Alignment genuinely novel, or is it merely a standard shared-backbone architecture for Multi-Layer Perceptrons? This partitioning relies on fixed hidden dimensions, which limits its applicability to heterogeneous neural networks.
2.	The concept of a Multi-Task Curse lacks rigorous mathematical formulation. Does it offer any new theoretical depth, or does it simply repackage the well-documented challenges of negative transfer and resource dilution?
3.	Proposition 3.1 relies on the unverified assumption that the heuristic reward positively correlates with true expected improvement. Why should historical progress serve as a reliable empirical proxy? Theoretical guarantee for future improvement in deceptive or non-convex landscapes is needed for this formulation.
4.	Proposition 3.3 states that truncation selection discards inferior external solutions. However, does generating and evaluating these maladaptive candidates not still consume the limited computational budget? The analysis ignores soft negative transfer, where mediocre transferred solutions survive truncation and pull the search into suboptimal basins.
5.	The experimental setup compares evolutionary methods using 16-neuron hidden layers against gradient-based baselines using 64-neuron architectures. Is it fair to cap the gradient methods at 10 million steps merely to match the wall-clock runtime of parallelized evolutionary strategies? This structural bias penalizes sample efficiency and invalidates the comparative claims.

---

> ### Author Rebuttal · Authors · 2026-03-30
>
> We sincerely thank you for the thoughtful comments and insightful questions.
>
> # For W1 & W2 (Semantic Alignment and Multi-Task Curse)
>
> Unlike standard Multi-Task Learning architectures that rely on explicitly shared parameters (jointly optimized via gradients), our Semantic Alignment differs. In EMT, each task maintains strictly independent search distributions. We do not directly share parameters; instead, we construct a structurally isomorphic subspace. This prerequisite enables our method to transfer knowledge from disparate, independent trajectories.
>
> *(See **Reviewer Bgwp** for network extensions.)*
>
> This aggregation is necessitated by the **Multi-Task Curse** ($\\mathcal{I}\_{MT} < \\mathcal{I}\_{ST}$), which we rigorously formalize here. In CMA-ES, optimization operates on $J\_k(m\_k, C\_k) = \\mathbb{E}\_{x\\sim\\mathcal{N}}[f\_k(x)]$. We quantify the curse via a Taylor expansion of expected improvement:
>
> $$\\mathcal{I}\_{MT} - \\mathcal{I}\_{ST} \\approx \\underbrace{\\sum\_{k=1}^K \\frac{\\partial \\mathbb{E}[\\Delta\_k]}{\\partial b\_k} \\left(b\_k - \\frac{B}{K}\\right)}\_{\\text{Resource Dilution}} + \\underbrace{\\sum\_{k=1}^K \\left( \\langle \\nabla\_{m} J\_k, \\delta m\_k \\rangle + \\text{Tr}(\\nabla\_{C} J\_k \\delta C\_k) \\right)}\_{\\text{Negative Transfer}}$$
>
> Using ES natural gradient properties ($\\nabla\_{m} J\_k \\approx \\nabla f\_k$ and $\\nabla\_{C} J\_k \\approx \\frac{1}{2} H\_k$), this mathematical inevitability decomposes into:
> 1. **Resource Dilution**: Uniform allocation forces the first term $\\le 0$ as tasks scale.
> 2. **Mean Shift**: Conflicting transferred updates ($\\delta m\_k$) oppose local gradients ($\\langle \\nabla f\_k, \\delta m\_k \\rangle < 0$).
> 3. **Covariance Distortion**: Maladaptive variance ($\\delta C\_k$) inflates the search distribution's condition number, driving $\\text{Tr}(H\_k \\delta C\_k) < 0$.
>
> Our framework breaks this curse systematically:
> * **Solving Dilution**: Reward-weighted evaluation dynamically optimizes the budget $b\_k$.
> * **Transfer Upper Bound**: Semantic alignment and reward-weighted aggregation pool high-quality shared statistics.
> * **Transfer Lower Bound**: **Weak Guidance Injection** acts as a firewall against Terms 2 and 3. Restricting external knowledge to discrete evaluation securely filters conflicting gradients and distorted covariances, ensuring a safe fallback to independent evolution.
>
> # For W3 (Theoretical Justification of Reward Design)
>
> We avoid static assumptions via a dual-mechanism approach tailored to CMA-ES dynamics:
>
> - **Performance Reward:** Assuming local smoothness, CMA-ES approximates natural gradient descent on $J(m, C) = \\mathbb{E}\_{x\\sim\\mathcal{N}}[f(x)]$. Because consecutive natural gradients correlate strongly, historical progress ($r\_k^{\\text{perf}}$) is a rigorous mathematical proxy for expected local improvement.
> - **Diversity Reward:** In deceptive basins ($r\_k^{\\text{perf}} \\to 0$), CMA-ES risks premature convergence. Defining $r\_k^{\\text{div}}$ via $\\text{trace}(C\_k)$ implicitly lower-bounds differential entropy. This fallback prevents covariance collapse and preserves exploration.
> - **Empirical Validation:** The `w/o-RE` ablation strictly underperforms our method, verifying $\\mathbb{E}[\\Delta\_{\\text{total}}^{\\text{weighted}}] \\geq \\mathbb{E}[\\Delta\_{\\text{total}}^{\\text{uniform}}]$. This empirically confirms the positive correlation condition (Prop. 3.1) holds in expectation.
>
> *(See **Reviewer jwef** for complete mathematical justification.)*
>
> # For W4 (Mitigation of Negative Transfer)
>
> Budget Consumption: With $\\lambda=100$ and transfer size $\\tau=1$ in our setting, the worst-case wasted budget (all transferred candidates fail truncation) is exactly $2\\%$ overhead. This trivial exploration cost is easily offset by the substantial gains of robust transfer.
>
> Soft Negative Transfer: Mediocre solutions surviving truncation cannot pull the search into suboptimal basins due to CMA-ES's non-linear recombination. Update weights $w\_i$ follow a logarithmic rank decay:
> $$w\_i \\propto \\max(0, \\ln(\\mu + 0.5) - \\ln(i))$$
> If a transferred solution is mediocre (ranking near the boundary $\\mu$), its weight $w\_i$ decays exponentially toward $0$. Consequently, only strictly superior external solutions can meaningfully dictate the search direction, exponentially decaying its influence to rigorously bound the risk of soft negative transfer.
>
> # For W5 (Experimental Fairness)
>
> Gradient baselines use 64 neurons (Stable-Baselines3 defaults) to avoid representational bottlenecks, whereas ES inherently excels with compact networks.
>
> To validate this, we evaluated PPO and A2C with $2 \times 16$ networks matching the ES setup. Results show 64-neuron baselines marginally outperform 16-neuron versions (Loss/Win/Tie of 64- vs 16-neurons: 0/1/17 for PPO, 1/4/13 for A2C). Thus, configuring each algorithm to its optimal regime ensures a fair comparison. 10M steps guarantees full convergence.

---

> > ### Author Rebuttal · Reviewer_FpL8 · 2026-04-04
> >
> > Reasons
> > I have carefully read the rebuttal, including the expanded clarifications regarding architectural extensions and the theoretical limits of the reward mechanism. I appreciate the authors' transparency, the substantial mathematical rigor added to the defense, and the supplementary experiments provided. The rebuttal significantly strengthens the paper and successfully addresses all of my initial concerns. I will improve a point for it.

---

> > > ### Author Response · Authors · 2026-04-04
> > >
> > > We sincerely appreciate your time in reviewing our rebuttal. We are very glad that our expanded mathematical clarifications and supplementary experiments have successfully resolved your concerns.
> > >
> > > Thank you again for your rigorous feedback and for increasing the score. Your constructive comments have significantly strengthened our work.

---

### Decision · Program_Chairs · 2026-04-30

**Decision:**

Accept (regular)

**Comment:**

A technically strong paper with consensus among the reviewers.